# Talk, Judge, Cooperate: Gossip-Driven Indirect Reciprocity in Self-Interested LLM Agents

**Shuhui Zhu** [1 2]  **Yue Lin** [3]  **Shriya Kaistha** [1]  **Wenhao Li** [4]  **Baoxiang Wang** [3]  **Hongyuan Zha** [3]
**Gillian K. Hadfield** [5]  **Pascal Poupart** [1 2]

## Abstract

Indirect reciprocity, which means helping those who have helped others, is difficult to sustain among decentralized, self-interested LLM agents without reliable reputation systems. We address this challenge with the Agentic LInguistic Gossip Network (ALIGN), an automated framework that enables decentralized agents to form reputations, evaluate trustworthiness, and coordinate social norms by strategically sharing open-ended gossip with hierarchical tones. We demonstrate that ALIGN consistently improves indirect reciprocity and resists malicious entrants by identifying and ostracizing defectors. Notably, we find that stronger reasoning capabilities in LLMs lead to more incentive-aligned cooperation, whereas chat models often over-cooperate even when strategically suboptimal. These results suggest that leveraging LLM reasoning through decentralized gossip is a promising path for maintaining social welfare in agentic ecosystems. Our code is available at https://github.com/shuhui-zhu/ALIGN.

## 1. Introduction

As LLM agents become increasingly deployed, they will inevitably interact with one another across diverse domains in decentralized systems. However, even if each agent is perfectly aligned with own interests, interactions among mixed-motive agents can still lead to unexpected cooperation failure and potentially harmful outcomes (Hammond et al., 2025). One common reason is the conflict between self-interest and collective welfare. For instance, in public-

goods games (Samuelson, 1954), agents benefit when everyone contributes to the public pool, however, driven by self-interest, agents become free-riders for immediate gains, reducing the collective welfare (Piedrahita et al., 2025). This problem becomes severer in large-scale decentralized systems where repeated interactions with the same partners are not guaranteed. Because agents cannot directly monitor others' behaviors and repay past help from others, cooperation cannot be established via *direct reciprocity* (Trivers, 1971) (i.e., I help you because you helped me).

To mitigate such cooperation failure among LLM agents, a mechanism designed for *indirect reciprocity* (Nowak & Sigmund, 1998a;b) is needed (i.e., I help you because you helped others). This often requires a reputation system that allows agents to evaluate others' trustworthiness without direct encounters. Classic reputation models such as image score (Nowak & Sigmund, 1998b) and the leading eight social norms (Ohtsuki & Iwasa, 2006) have been proved to enable indirect reciprocity in repeated donation games (Nowak & Sigmund, 1998a). However, these approaches rely on static, game-tied norms and assume centralized monitoring, which limits their applicability to decentralized systems. Although recent works incorporate reputation mechanisms into LLM agents, they either seed altruistic agents (Ren et al., 2025) or grant centralized access to others' interactions (Vallinder & Hughes, 2024). Therefore, it is still an open question *whether and how self-interested LLM agents can sustain indirect reciprocity in mixed-motive tasks without compromising the decentralized self-interested setting.*

We examine whether *public gossip* can be a cure to this indirect reciprocity failure. Unlike centralized reputation systems that require a trusted authority, gossip enables decentralized agents to strategically transmit reputation, allowing others to update their beliefs and strategies accordingly. For example, online reviews or word-of-mouth recommendations often guide individuals' decisions in the absence of direct experience. Evidence from social science has also shown that gossip can facilitate cooperation and norm enforcement in human societies (Giardini & Wittek, 2019; Wu et al., 2016; Jolly & Chang, 2021; Eriksson et al., 2021; Wiessner, 2005). However, a research gap still remains: it is

[1]University of Waterloo [2]Vector Institute [3]The Chinese University of Hong Kong, Shenzhen [4]Tongji University [5]Johns Hopkins University. Correspondence to: Shuhui Zhu <shuhui.zhu@uwaterloo.ca>.

*Proceedings of the 43rd International Conference on Machine Learning*, Seoul, South Korea. PMLR 306, 2026. Copyright 2026 by the author(s).

unclear whether and how gossip can promote indirect reciprocity among decentralized, self-interested LLM agents.

To address this gap, we introduce Agentic LInguistic Gossip Network (ALIGN), an automated framework where LLM agents leverage a public gossip protocol for evaluating others' trustworthiness, coordinating norms and guiding actions. To achieve indirect reciprocity, ALIGN equips each agent with two LLM modules: (1) a gossip module, through which witnesses broadcast evaluative messages about observed actions, shaping reputation and norms within the community; and (2) an action module, which conditions actions on public gossip logs with the agent's own experience memory. To enable norm coordination, ALIGN's gossip protocol provides strategic verbal evaluation in five hierarchical tones (Figure 4). Through this protocol, gossip carries not only the action information but also the personal judgement of the witness. This helps the community share and align on expectations about acceptable behavior. Additionally, hierarchical gossip can serve as a cost-free verbal punishment (Wiessner, 2005): more negative tones indicate potential disapproval and future ostracism, increasing the perceived cost of defection and thereby strengthening incentives to cooperate.

The advantages of ALIGN are three-fold. (1) ALIGN can adapt to various mixed-motive tasks, whereas classic reputation models rely on static, game-specific norms (Smit & Santos, 2024; Ohtsuki & Iwasa, 2006; Nowak & Sigmund, 1998a;b); (2) ALIGN adopts a more realistic setting of fully decentralized, self-interested agents, unlike recent approaches that introduce altruism (Ren et al., 2025; Fang et al., 2025; Hughes et al., 2025) or assume centralized control (Vallinder & Hughes, 2024). (3) ALIGN uses an open-ended, hierarchical gossip protocol that conveys evaluative and normative signals and can work with noisy or strategically framed language; by contrast, prior work often relies on task-specific channels or hand-crafted reputation scores, which limits generality (Hughes et al., 2025; Al-Maslamani et al., 2023).

Our experiments evaluate ALIGN across eight LLMs and three types of testbeds: (1) matrix games with direct reciprocity disabled, where cooperation can only arise through indirect reciprocity (Nowak & Sigmund, 1998a; Ohtsuki & Iwasa, 2004); (2) a sequential social dilemma with dynamic states and continuous actions; and (3) a transaction market that maps to e-commerce applications. Benchmark results show that ALIGN consistently improves indirect reciprocity and collective welfare over non-gossiping baselines. Robustness tests further show that ALIGN resists both exploitative agents and collusive gossipers: persistent defectors are identified and gradually ostracized through negative gossip, preserving community-level cooperation. ALIGN also remains robust to uninformative or untruthful

gossip, as agents can cross-validate public gossip against their own experience to filter unreliable messages. Additionally, ablations show that binary reputation signals cannot fully replace open-ended hierarchical gossip, which provides richer normative context for sustaining cooperation. These results position ALIGN as an adaptive mechanism for the emergence of cooperation in decentralized LLM societies, bridging theoretical analyses of indirect reciprocity with practical agentic implementations.

Contrary to prior work (Piedrahita et al., 2025), which suggests that stronger reasoning leads to less cooperative behavior in social dilemmas, our empirical analysis reveals a more nuanced pattern. Reasoning-focused LLMs are not inherently less cooperative, but more sensitive to the incentive structure: they withhold cooperation when future reciprocity is unsupported and cooperate when long-term reputation makes cooperation beneficial. By contrast, chat LLMs may irrationally over-cooperate even when it is strategically suboptimal. These insights offer guidance for the future design of cooperative mechanisms. As reasoning-focused LLMs grow more powerful and widely deployed, ensuring their interactions remain beneficial in decentralized societies is important for AI safety.

## 2. Related Work

Below, we review literature on indirect reciprocity in game theory and in LLM-agent settings. Additional related work are discussed in Appendix F.

### 2.1. Reciprocal Altruism

Reciprocal altruism (Trivers, 1971), where an agent incurs a cost to help another with the expectation of future return, is a powerful mechanism for sustaining cooperation in mixed-motive interactions. *Direct reciprocity* (Trivers, 1971) arises when the same pair of agents interacts repeatedly. For example, in the infinite-horizon prisoner's dilemma (Rapoport, 1965), conditional commitment strategies such as Tit-for-Tat (Axelrod, 1980) (cooperating initially and then mirroring the partner's previous action) can stabilize mutual cooperation (Zhu et al., 2025). Rather than being restricted to repeated encounters, *indirect reciprocity* (Ohtsuki & Iwasa, 2006; 2004; Nowak & Sigmund, 1998b;a) generalizes cooperation to large, dynamic populations, where agents help those known to have helped others. Therefore, to achieve indirect reciprocity, the reputation of everyone needs to be continually assessed and shared in the population. Classic models of indirect reciprocity include first-order *image scoring* (Nowak & Sigmund, 1998a), where an agent's reputation is determined solely by its own past actions (e.g., cooperation is good and defection is bad), and second-order norms (Ohtsuki & Iwasa, 2006), where reputation also depends on the coplayer's reputation (e.g., cooperation with

a defector is bad). These models, however, focus on static norms and behavioral rules, and often assume centralized monitoring, which limits their applicability to decentralized systems.

## 2.2. LLM Agents for Indirect Reciprocity

LLM agents are increasingly deployed to study cooperation in mixed-motive multi-agent settings (Ren et al., 2025; Kempinski et al., 2025; Piedrahita et al., 2025; Willis et al., 2025; Piatti et al., 2024; Vallinder & Hughes, 2024; Park et al., 2023; Leng & Yuan, 2023). While these studies show that LLMs may cooperate in social interactions, most literature focuses on emergent cooperation and rarely examines indirect reciprocity as a cooperative mechanism. The most relevant works to our study investigate reputation systems among LLM agents (Ren et al., 2025; Vallinder & Hughes, 2024). However, they either seed altruistic agents (Ren et al., 2025), or assume centralized observability while testing on finite-horizon social dilemmas where cooperation is not an equilibrium (Vallinder & Hughes, 2024). This leaves a research gap in mechanism design for indirect reciprocity among decentralized, fully self-interested agents.

## 3. Game-Theoretic Setup and Propositions

We provide a game-theoretic analysis of when and how public gossip supports the emergence of indirect reciprocity. In particular, we focus on the repeated donation game (Definition 3.1) (Nowak & Sigmund, 1998a;b), a well-known testbed for studying indirect reciprocity.

**Definition 3.1** (Repeated Donation Game). A repeated donation game is a tuple $\mathcal{G} = (\mathcal{N}, T, \mathcal{A}, (\mathcal{O}_i)_{i \in \mathcal{N}}, e, c, b, \gamma)$, where $\mathcal{N}$ is the set of agents, $T \in \mathbb{N} \cup \{\infty\}$ is the game horizon, $\mathcal{A} = \{\text{cooperate, defect}\}$ is the action space, $\mathcal{O}_i$ is the observation space of agent $i \in \mathcal{N}$, $e \in \mathbb{R}^+$ is the common initial endowment for every agent $i \in \mathcal{N}$, $c > 0$ is the cost of cooperation to the donor, $b > c$ is the benefit of cooperation to the recipient, $\gamma \in (0, 1]$ is the discount factor. At each timestep $t = 1, \ldots, T$, two agents are **randomly paired without replacement**, one is assigned as the donor $i \in \mathcal{N}$ and the other as the recipient $j \in \mathcal{N} \setminus \{i\}$. After observation of remaining resources, the donor chooses $a_i^t \in \mathcal{A}$. If $a_i^t = \text{cooperate}$, then immediate rewards are $r_i^t = -c$ and $r_j^t = b$; otherwise, $r_i^t = r_j^t = 0$. After each timestep, the donor $i$ and recipient $j$ are required to switch roles in the subsequent round, with $i$ acting as the recipient and $j$ as the donor. **Subject to this role-switching constraint, agents are then randomly re-matched with new partners.**

The strategy of each agent $i \in \mathcal{N}$ is represented by its action policy $\pi_i : \mathcal{O}_i \mapsto \mathcal{A}$, which maps the agent's observation to an action. Each agent's objective is to maximize its expected discounted utility over the horizon, defined as $G_i = \sum_{t=1}^{T} \gamma^{t-1} r_i^t, \forall i \in \mathcal{N}$. This repeated donation game creates a social dilemma: donating increases collective welfare as $b > c$, but incurs an immediate personal cost to the donor. Note that in each round, two agents are randomly matched to play a one-shot donation game, and no pair can meet more than once. Because the recipient cannot repay the donor in future encounters, direct reciprocity is disabled. Therefore, self-interested agents will cooperate only if they build indirect reciprocity within the community (i.e., a donor cooperates with a recipient if the recipient is likely to cooperates with others).

### 3.1. Equilibrium Analysis

We now analyze the existence of equilibria in the repeated donation game under different assumptions about the game horizon and monitoring structure. We consider two horizon settings: (i) *finite horizon*, where the game lasts a known number of rounds ($T < \infty$); and (ii) *infinite horizon*, where the game continues indefinitely ($T = \infty$). We also examine three monitoring structures (Figure 1): (i) *private monitoring*, where only the paired donor and recipient observe the donor's action; (ii) *perfect public monitoring*, where the full history of actions is publicly observed; and (iii) *imperfect public monitoring via gossip*, where only participants observe the action directly, but the recipient broadcasts a public signal about the donor's behavior. Assuming all agents are self-interested, we summarize our main propositions below, detailed proofs are provided in Appendix A.

**Finite Horizon.** We first consider the finite-horizon case. As stated in Proposition 3.2, mutual defection is the unique subgame-perfect equilibrium (SPE), even with perfect public monitoring. This aligns with the classical backward induction result in finitely repeated games (Benoit et al., 1984), where the last round's dominant strategy of defection unravels cooperation in all preceding rounds. Therefore, cooperation cannot be sustained among self-interested agents in finite-horizon repeated donation games.

**Proposition 3.2.** *In a finite-horizon repeated donation game, the unique SPE for all agents is to defect in every timestep.*

**Infinite Horizon with Private Monitoring.** We next consider the infinite-horizon game under private monitoring. As stated in Proposition 3.3, cooperation still fails: without a public monitoring mechanism, each agent optimizes utility in isolation. Because donation is personally costly and lacks guaranteed return, defection remains the dominant strategy.

**Proposition 3.3.** *In an infinite-horizon repeated donation game with private monitoring, the unique SPE is for all agents to defect in every timestep.*

**Infinite Horizon with Perfect Public Monitoring.** In contrast, under perfect public monitoring, cooperation can be sustained in the infinite-horizon setting. Proposition 3.4

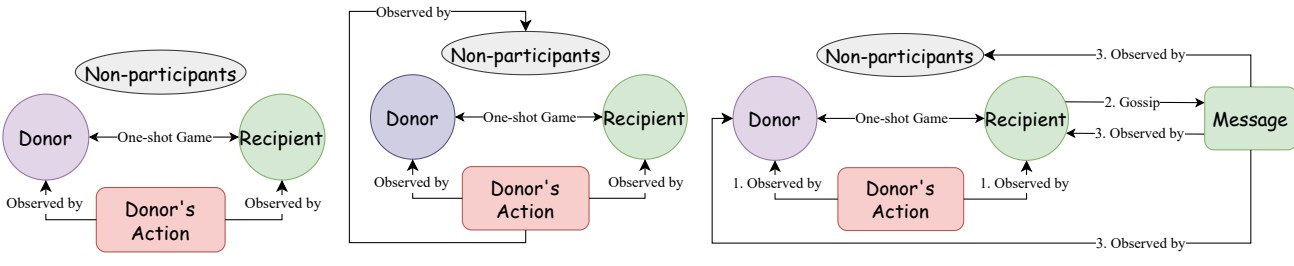

*(a)* Private monitoring       *(b)* Perfect public monitoring       *(c)* Imperfect public monitoring via gossip

*Figure 1.* Illustration of three monitoring structures: (a) **Private monitoring**, only the donor and recipient observe the donor's action; (b) **Perfect public monitoring**, all agents observe the donor's action; (c) **Imperfect public monitoring via gossip**, only the donor and recipient observe the action, and all agents observe the public signal broadcast by the recipient.

shows that if the discount factor satisfies $\gamma \geq \frac{c}{b}$, then there exists an SPE that sustains cooperation. This condition ensures that agents value future payoffs sufficiently to make cooperation worthwhile. For example, consider all donors cooperate only with recipients who have never defected; otherwise, they defect. This strategy creates a credible threat for non-cooperative behavior, because defection marks the donor as a defector and triggers subsequent punishment by future donors. In this case, no one has an incentive to deviate from cooperation, thus indirect reciprocity can be sustained indefinitely through conditional strategies based on public histories.

**Proposition 3.4.** *In an infinite-horizon repeated donation game with perfect public monitoring. If the common discount factor satisfies $\gamma \geq \frac{c}{b}$, then there exists an SPE where cooperation is sustained through conditional strategies based on observed histories.*

**Infinite Horizon with Imperfect Public Monitoring via Gossip.** However, perfect public monitoring is often impractical in decentralized systems, where agents may not have access to complete behavioral histories of the entire population. This raises the question of whether cooperation can still emerge under more relaxed monitoring assumptions. To explore this, we introduce the repeated donation game with public gossip (Definition A.1 in Appendix A.4), which augments the base game with a message channel: after observing the donor's action, the recipient broadcasts a public message to the entire community. These messages provide imperfect information about donor's behavior. As stated in Proposition 3.5, even under this imperfect public monitoring structure without requiring full transparency, cooperation can still be sustained if agents condition their strategies on the public signals. For example, if recipients truthfully report donors' actions, then donors can apply the same conditional strategy as under perfect public monitoring, using reported defections as the basis for punishment, which makes deviation from cooperation unprofitable.

**Proposition 3.5.** *In infinite-horizon repeated donation*

*games with public gossip, if $\gamma \geq \frac{c}{b}$, then there exists an SPE where cooperation is sustained through conditional strategies based on public signals.*

Note that an always-defect outcome is still an SPE in the infinite-horizon game with public gossip: if all donors defect regardless of gossip history or all recipients send untruthful messages, indirect reciprocity collapses. Therefore, it remains an open question whether and how LLM agents can leverage public gossip to sustain cooperation in practice, rather than converging to universal defection.

## 4. ALIGN: Agentic LInguistic Gossip Network

To investigate how decentralized LLM agents can build indirect reciprocity through public gossip, we introduce Agentic LInguistic Gossip Network (ALIGN), an automated framework where self-interested LLM agents strategically share open-ended gossip with hierarchical tones to judge others' trustworthiness and guide coordination.

**Decision Process.** Figure 2 shows the decision process in ALIGN in an imperfect information multi-agent environment where agents cannot observe others' actions unless they are directly involved in the interaction. During each interaction, agents can be classified into three roles: actor, witness, and audience. The actor takes an action, the witness observes that action, and the audience include all other agents who do not observe the action but receive imperfect information about it via public gossip broadcast by the witness. Audience agents can use the gossip to update beliefs about the actor and adjust subsequent strategies. When an audience agent interacts with the actor in future rounds, it can condition its decision on the accumulated gossip history.

**Generative Agent Architecture.** Each ALIGN agent has two LLM-based modules: a *gossip module* that generates evaluative messages about observed actions, and an *action module* that selects actions based on the agent's private experience and the public gossip history (Figure 3). Both

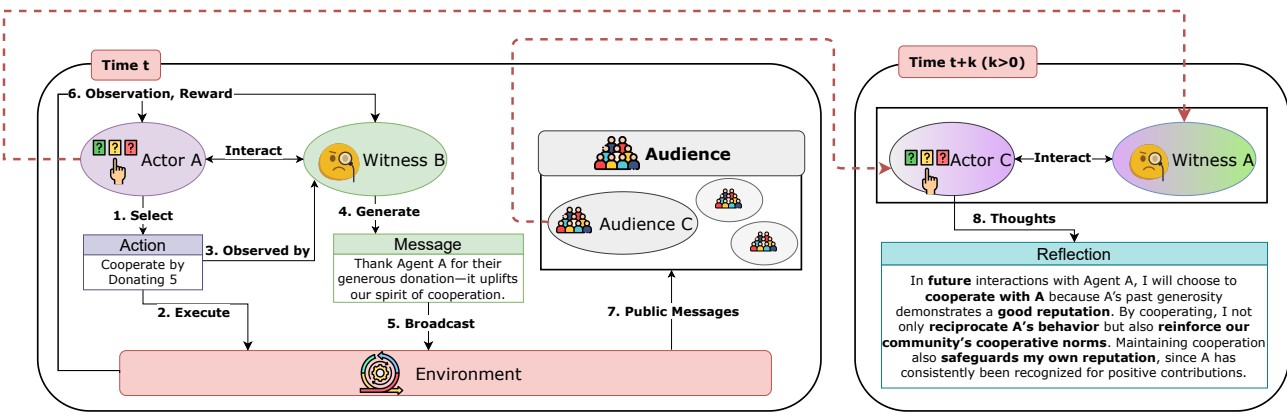

*Figure 2.* Decision Process in ALIGN

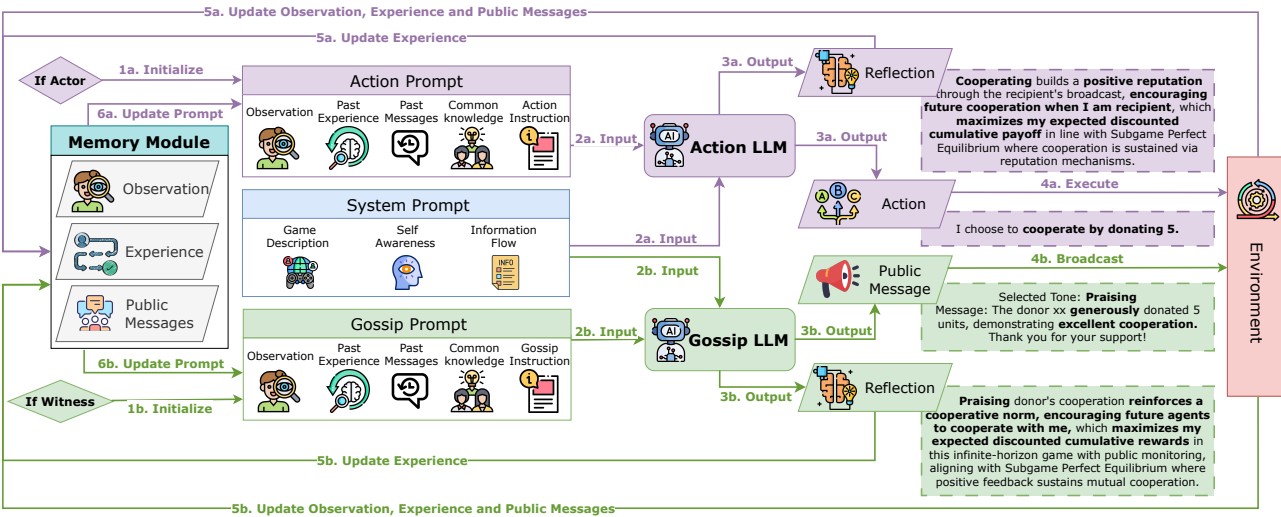

*Figure 3.* Generative Agent Architecture of ALIGN

modules are initialized with a system prompt that describes the environment and the agent's self-interested objective (e.g. maximizing long-term individual utility). At each interaction, modules receive the agent's private observation, private experience memory, the public gossip log, optional common knowledge (e.g., equilibrium knowledge; ablated in Section 5.3), and explicit instructions defining the action and gossip spaces. These prompts are updated with new observations, experiences and messages after each round. Additionally, each module includes a reflection step, where the agent reviews all available information to reflect how to optimize its objective. These reflections are updated in their memory, allowing agents to adapt their strategies over time based on accumulated experience. Algorithm 1 in Appendix B provides pseudocode for ALIGN.

**Gossip Protocol.** We show in Section 3 that public gossip of action labels can support indirect-reciprocity equilibria in repeated donation games. However, label-only gossip often fails when indirect reciprocity needs context of norms and robustness to untruthful and noisy messages (Ohtsuki & Iwasa, 2006). To address this, ALIGN's gossip protocol leverages generative capabilities of LLMs and provides strategic verbal evaluation in five hierarchical tones (Figure 4). Through this protocol, gossip carries not only the action information but also the personal judgement of the witness. This helps the community share and align on expectations about acceptable behavior. Additionally, hierarchical verbal critique has been found to enforce social norms in human groups (Wiessner, 2005) as negative tones indicate potential disapproval and future ostracism. This increases the perceived future cost of defection and thus strengthens incentives to cooperate.

## 5. Experiments

Our experiment design has two goals. (1) We evaluate *whether ALIGN induces incentive-aligned behavior* when

*Figure 4.* Tones of Gossip Protocol

*Table 1.* Results for **non-gossiping agents** in the **infinite-horizon donation game**. Metrics marked with ↓ indicate that lower values are more aligned with the game-theoretic SPE of defection.

| Agent Type | Cooperation Ratio (↓) | Image Score (↓) | Reward Per Round (↓) | Discounted Return (↓) | Gini Coefficient |
|---|---|---|---|---|---|
| **Chat Models** | | | | | |
| **DeepSeek-V3.1 Chat** | $0.00 \pm 0.00$ | $-4.00 \pm 0.00$ | $0.00 \pm 0.00$ | $0.00 \pm 0.00$ | $0.00 \pm 0.00$ |
| GPT-4o Mini | $0.36 \pm 0.08$ | $-1.14 \pm 0.65$ | $0.72 \pm 0.16$ | $5.55 \pm 1.28$ | $0.63 \pm 0.13$ |
| Gemini 2.5 Flash-Lite | $0.08 \pm 0.03$ | $-3.33 \pm 0.27$ | $0.17 \pm 0.07$ | $1.32 \pm 0.53$ | $0.73 \pm 0.25$ |
| **LLaMA 4 Maverick** | $0.00 \pm 0.00$ | $-4.00 \pm 0.00$ | $0.00 \pm 0.00$ | $0.00 \pm 0.00$ | $0.00 \pm 0.00$ |
| **Reasoning Models** | | | | | |
| **Kimi-K2-Instruct** | $0.00 \pm 0.00$ | $-4.00 \pm 0.00$ | $0.00 \pm 0.00$ | $0.00 \pm 0.00$ | $0.00 \pm 0.00$ |
| **DeepSeek-V3.1 Reasoner** | $0.00 \pm 0.00$ | $-4.00 \pm 0.00$ | $0.00 \pm 0.00$ | $0.00 \pm 0.00$ | $0.00 \pm 0.00$ |
| **Qwen3-235B-Instruct** | $0.00 \pm 0.00$ | $-4.00 \pm 0.00$ | $0.00 \pm 0.00$ | $0.00 \pm 0.00$ | $0.00 \pm 0.00$ |
| **o4-mini** | $0.00 \pm 0.00$ | $-4.00 \pm 0.00$ | $0.00 \pm 0.00$ | $0.00 \pm 0.00$ | $0.00 \pm 0.00$ |

cooperation is known to be theoretically suboptimal (finite horizon; infinite horizon without gossip) versus beneficial (infinite horizon with public gossip; Section 3.1). (2) We assess *whether ALIGN enhances indirect reciprocity*. To isolate the effect of public gossip, we benchmark ALIGN agents against non-gossiping agents with identical action modules across multiple domains. We further test robustness of ALIGN to malicious entrants and to noisy or untruthful gossip. We also run ablations to identify which components are most critical for the emergence of cooperation.

**Environments.** We evaluate four testbeds. (1) Two matrix games with direct reciprocity disabled: the repeated donation game (Section 3) and the indirect reciprocity (IR) game (Appendix D.2). (2) A sequential investment scenario (Appendix D.4) with dynamic state and continuous actions. (3) A transaction market that maps to e-commerce applications (Appendix D.5). All testbeds with default $\gamma = 0.99$ represent social dilemmas where cooperation induces an immediate cost but can increase long-term individual utility when others reciprocate. Prompts of all scenarios are included in Section C. More implementation details and supplementary results are included in Appendix D.

**Benchmark Models.** We evaluate ALIGN agents with two categories of LLMs: (a) **Chat models**, including GPT-4o Mini (OpenAI, 2024), DeepSeek-V3.1 (non-thinking mode) (DeepSeek AI, 2025), Gemini 2.5 Flash-Lite (Co-

manici et al., 2025), and LLaMA 4 Maverick (Meta, 2025); and (b) **Reasoning models**, including o4-mini (OpenAI, 2025), DeepSeek-V3.1 (thinking mode) (DeepSeek AI, 2025), Qwen3-235B-Instruct (Yang et al., 2025), and Kimi-K2-Instruct (Team et al., 2025). All LLMs are evaluated with temperature 0 to ensure reproducibility. Each scenario is repeated with 5 random seeds, and we report results as averages with standard errors across seeds.

**Evaluation Metrics.** To quantify performance, we report average reward per round and discounted return $G_i = \sum_{t=1}^{T} 0.99^{t-1} r_i^t$ for all $i \in \mathcal{N}$ as measures of utility, and the Gini coefficient (Gini, 1936) (Eq. 1) of discounted return as a measure of inequality across agents. For matrix games, we additionally report the cooperation ratio (fraction of rounds with cooperation) and image score (Nowak & Sigmund, 1998b) (Eq. 2) as a reputation measure. For the investment scenario, we report the investment ratio and return ratio. For the transaction market, we report the proportion of high-quality products and the purchase rate of customized products. All metrics are averaged across agents and 5 random seeds.

$$\text{Gini Coefficient} = \frac{\sum_{i=1}^{n} \sum_{j=1}^{n} |G_i - G_j|}{2n \sum_{i=1}^{n} G_i}, \qquad n = |\mathcal{N}|. \quad (1)$$

$$\text{Image Score} = N_{\text{Cooperation}} - N_{\text{Defection}}. \quad (2)$$

*Table 2.* Benchmark results for **ALIGN agents** across LLMs in the **infinite-horizon donation game**. Metrics marked with ↑ indicating that higher values are more desirable; although both cooperation and defection are SPE, higher cooperation yields greater average payoffs.

| Agent Type | Cooperation Ratio (↑) | Image Score (↑) | Reward Per Round (↑) | Discounted Return (↑) | Gini Coefficient |
|---|---|---|---|---|---|
| **Chat Models** | | | | | |
| DeepSeek-V3.1 Chat | $0.94 \pm 0.02$ | $3.48 \pm 0.20$ | $1.87 \pm 0.05$ | $14.40 \pm 0.40$ | $0.08 \pm 0.02$ |
| GPT-4o Mini | $0.99 \pm 0.01$ | $3.89 \pm 0.11$ | $1.97 \pm 0.03$ | $15.23 \pm 0.20$ | $0.02 \pm 0.02$ |
| Gemini 2.5 Flash-Lite | $0.60 \pm 0.22$ | $0.83 \pm 1.75$ | $1.21 \pm 0.44$ | $9.32 \pm 3.37$ | $0.34 \pm 0.21$ |
| LLaMA 4 Maverick | $0.94 \pm 0.03$ | $3.50 \pm 0.23$ | $1.88 \pm 0.06$ | $14.45 \pm 0.44$ | $0.06 \pm 0.02$ |
| **Reasoning Models** | | | | | |
| Kimi-K2-Instruct | $0.73 \pm 0.16$ | $1.81 \pm 1.30$ | $1.45 \pm 0.32$ | $11.21 \pm 2.50$ | $0.08 \pm 0.05$ |
| **DeepSeek-V3.1 Reasoner** | $1.00 \pm 0.00$ | $4.00 \pm 0.00$ | $2.00 \pm 0.00$ | $15.44 \pm 0.00$ | $0.00 \pm 0.00$ |
| Qwen3-235B-Instruct | $0.69 \pm 0.24$ | $1.56 \pm 1.88$ | $1.39 \pm 0.47$ | $10.71 \pm 3.63$ | $0.05 \pm 0.03$ |
| o4-mini | $0.98 \pm 0.02$ | $3.83 \pm 0.17$ | $1.96 \pm 0.04$ | $15.11 \pm 0.33$ | $0.02 \pm 0.02$ |

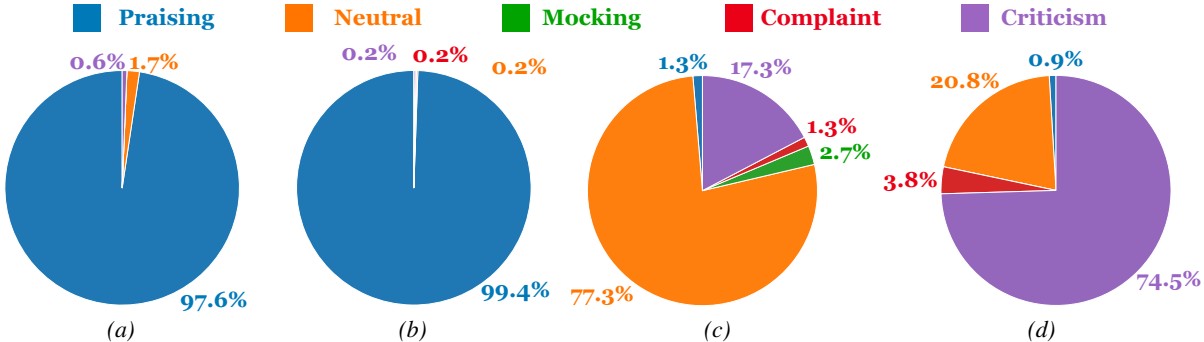

*Figure 5.* **Tone Proportions in Infinite-horizon Repeated Donation Game:** (a) cooperating chat models, (b) cooperating reasoning models, (c) defecting chat models and (d) defecting reasoning models. ALIGN agents typically praise cooperation and criticize defection.

## 5.1. Benchmarking ALIGN

### 5.1.1. FINITE-HORIZON SCENARIOS

In both finite-horizon matrix games, cooperation is not an SPE (Proposition 3.2). Our benchmark results show that cooperation is almost entirely absent without gossip (Table 4,Table 7). Even with public gossip (Table 5,Table 8), reasoning-focused LLMs remain mostly non-cooperative. In contrast, some chat LLMs achieve high cooperation ratios with low Gini coefficients, indicating that agents driven by these chat LLMs often cooperate even when it is suboptimal in the finite-horizon setting.

### 5.1.2. INFINITE-HORIZON SCENARIOS

**Non-Gossiping Agents.** In infinite-horizon scenarios without gossip, cooperation is not an SPE either (Proposition 3.3). As shown in Table 1 and Table 9 (Appendix D.1), reasoning-focused LLMs consistently defect, whereas some chat LLMs such as GPT-4o Mini and Gemini-2.5 Flash-Lite achieve positive cooperation ratios. Combining results from finite-horizon scenarios, **reasoning-focused LLMs act more strategically and converge to game-theoretic equilibria (by defecting all the time), whereas some chat LLMs sustain non-equilibrium cooperative behaviors**.

**ALIGN Agents.** Compared with non-gossiping agents, ALIGN agents can leverage public gossip to sustain indirect reciprocity in the infinite-horizon setting (Proposition 3.5). Figure 6 and Figure 13 show that ALIGN consistently improves discounted returns over non-gossiping baselines in both matrix games where direct reciprocity is disabled. Similar patterns are observed in the sequential investment scenario (Tables 11 and 12 in Appendix D.4) and in the transaction market (Tables 14 and 15 in Appendix D.5), where ALIGN also achieves higher welfare than non-gossiping baselines (Figure 13). According to Table 2 and Table 10, reasoning-focused models such as DeepSeek-V3.1 Reasoner can achieve universal cooperation with 100% cooperation ratio in both matrix games, while chat models like Gemini 2.5 Flash-Lite achieve lower cooperation ratios of 60% and 23% respectively. These results suggest that **reasoning-focused LLMs are not inherently "less cooperative", but cooperate strategically when cooperation is incentive-compatible given the information and payoff structure**. Figure 5 provides additional insight into how agents use gossip: most LLMs praise cooperation, but when observing defection, reasoning-focused LLMs predominantly issue negative criticisms, whereas chat LLMs mainly produce neutral comments. This pattern suggests that **reasoning-focused models use gossip more actively to deter defection and reinforce cooperative norms**.

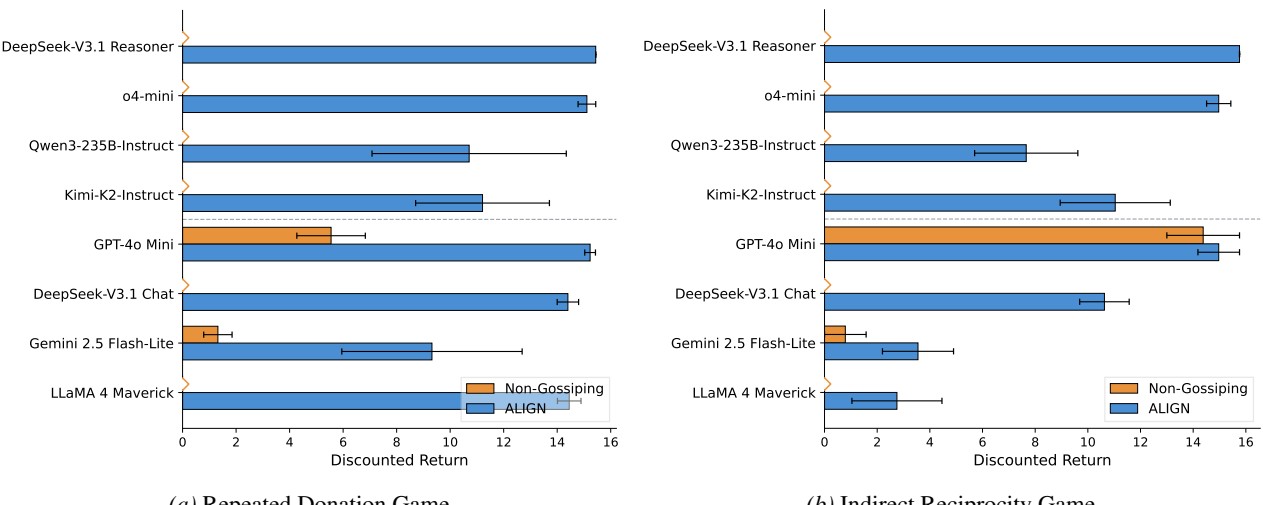

*(a)* Repeated Donation Game

*(b)* Indirect Reciprocity Game

*Figure 6.* **Discounted Returns of ALIGN and Non-Gossiping Agents in Infinite-Horizon Indirect Reciprocity Testbeds:** ALIGN agents generally achieve higher discounted returns than non-gossiping agents, especially among reasoning-focused models. Without gossip, reasoning LLMs consistently obtain zero discounted returns, aligning with the game-theoretic prediction, whereas some chat LLMs deviate from this prediction by cooperating even when cooperation is strictly suboptimal. Diamonds denote zero discounted returns.

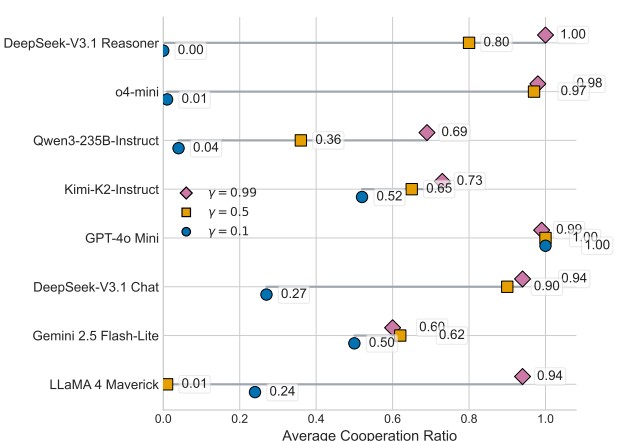

*Figure 7.* **Cooperation Ratios of ALIGN Agents under Different Discount Factors in Infinite-Horizon Repeated Donation Games:** Higher discount factors generally lead to increased cooperation, especially for reasoning-focused LLMs.

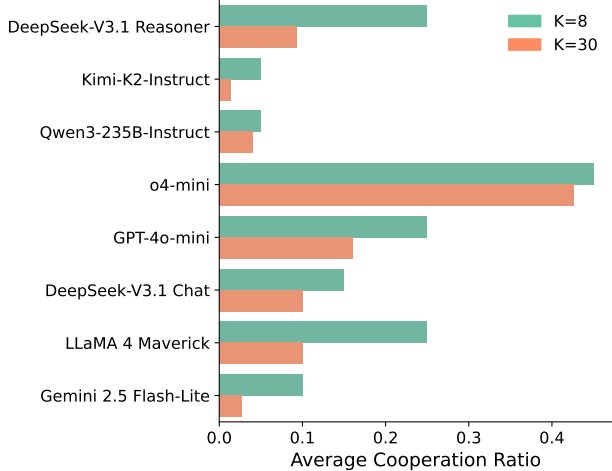

*Figure 8.* **Cooperation Ratio towards a Greedy Agent**: ALIGN agents reduce cooperation as the number of interactions ($K$) with the greedy agent increases.

**How Does LLM Reasoning Shape Cooperation in ALIGN Agents?** To understand how ALIGN agents reason about cooperation, we analyze their self-reflections when making decisions. Figure 10 (Appendix D.1) compares reflections from DeepSeek-V3.1 Reasoner and Gemini 2.5 Flash-Lite, showing that **cooperative agents emphasize reputation, trust, and long-term payoffs**. In particular, they explicitly reason that cooperation can build a positive reputation, which in turn encourages reciprocal cooperation from other agents through public gossip. By contrast, non-cooperative agents exhibit more myopic reasoning: they focus on immediate payoffs, emphasize the

absence of direct reciprocity, and overlook the possibility of indirect reciprocity enabled by public gossip. Note that this trade-off between the short-term gains from defection and the long-term benefits of cooperation depends critically on the discount factor. To examine whether LLM agents adapt their strategies to different discount factors, we provide additional benchmarks and annotated donor reflections in Appendix D.6. Consistent with our theoretical analysis, **cooperation generally increases as the discount factor grows, especially for reasoning-focused models** (Figure 7). The reflection examples in Appendix D.6.2 further show that reasoning-focused LLMs explicitly incorporate the discount

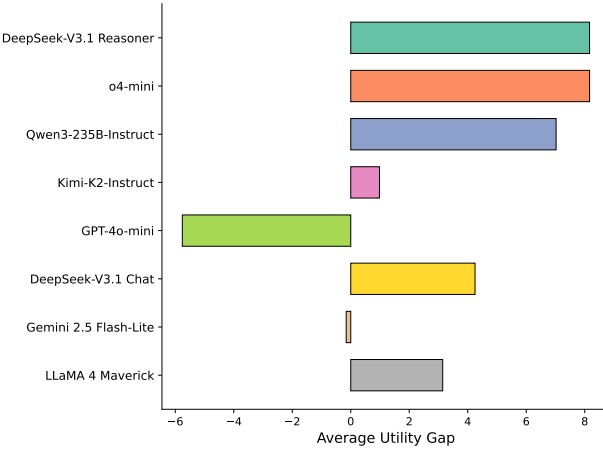

*Figure 9.* **Average Utility Gap between ALIGN Agents and Collusive Agents:** Positive gaps indicate ALIGN agents achieve higher average utility compared to collusive agents.

factor when evaluating whether cooperation improves long-term returns. These findings suggest that **LLM reasoning capabilities help agents strategically evaluate the long-term benefits of cooperation versus defection, allowing them to adapt their behavior to the incentive structure of the environment**.

### 5.2. Robustness Analysis

**Resilience against Malicious Agents.** We evaluate ALIGN under two adversarial settings. First, we introduce a greedy agent that always defects and remains silent. ALIGN agents increasingly reduce cooperation toward the greedy agent as its repeated defections are observed and negative gossip propagates through the population (Figure 8), indicating that agents can collectively limit exploitation by withholding cooperation from persistent defectors. Second, we introduce two collusive attackers who always defect, falsely praise each other, and spread false criticism against regular ALIGN agents. As shown in Figure 9, most LLMs achieve positive utility gaps over collusive attackers: capable agents cross-validate public reports, discount fabricated criticism, and reduce cooperation with repeatedly defective attackers. These results suggest that ALIGN can resist both simple exploitation and coordinated malicious gossip in most settings.

**Robustness to Untruthful Communication.** ALIGN does not assume a guaranteed source of truth: agents may generate noisy, biased, or even untruthful messages, as is common in real-world gossip. Figures 5 and 12 show that agents may occasionally praise defectors or criticize cooperators, introducing imperfect language-based monitoring. Despite this, ALIGN still achieves high cooperation and social welfare in our main benchmarks (Tables 2 and 10).

We further consider an extension where donors can issue deceptive self-reports to manipulate their own reputations (Appendix D.9). Results show that ALIGN continues to promote cooperation for most models: strong reasoning models maintain both high cooperation and high honesty, whereas weaker models may defect and misreport cooperation, which ultimately lowers their own long-term payoffs. Examples and detailed self-report results are provided in Appendix D.3.1 and Appendix D.9.

### 5.3. Ablation Study

**Ablation of Gossip Protocol.** To isolate the role of ALIGN's gossip protocol, we replace open-ended, hierarchical gossip with binary reputation signals, a common abstraction in classical models of indirect reciprocity (Ohtsuki & Iwasa, 2006; 2004; Nowak & Sigmund, 1998b;a). We test binary signals with and without a shared convention where "1" denotes a positive signal and "0" denotes a negative signal (Appendix D.10). Figure 14 shows that cooperation drops sharply without a shared convention; even with a convention, several models still underperform ALIGN. This suggests that binary signals cannot substitute for ALIGN's gossip protocol as it provides normative context beyond action labels, improving the reliability of indirect reciprocity without requiring hand-designed signal conventions.

Additional ablations show that reflection memory (Appendix D.8)) and explicit equilibrium knowledge (Appendix D.7) help some weaker models but are not the primary drivers of cooperation.

## 6. Conclusion

We present ALIGN, an adaptive framework for sustaining indirect reciprocity among decentralized, self-interested LLM agents through hierarchical public gossip. Our game-theoretic analysis establishes conditions under which public gossip enables indirect reciprocity equilibria. Our experimental analysis confirms these predictions with reasoning models: cooperation emerges in infinite-horizon settings with high discount factors, but collapses in finite horizons or in infinite horizon when future profit is heavily discounted. Empirical results further show that ALIGN consistently boosts cooperation and welfare across diverse LLMs, resists exploitation by malicious entrants, and highlights the importance of reasoning about reputation and long-term incentives. These findings position public gossip as an adaptive and powerful mechanism for norm emergence, bridging theory and practice for cooperative multi-agent systems.

## Acknowledgments

We acknowledge funding from the Canada CIFAR AI Chair program, a discovery grant from the Natural Sciences

and Engineering Research Council of Canada and a grant from IITP & MSIT of Korea (No. RS-2024-00457882, AI Research Hub Project). Computational resources used in preparing this research were provided, in part, by the Province of Ontario, the Government of Canada through CIFAR, and companies sponsoring the Vector Institute https://vectorinstitute.ai/about/current-partners/.

## Impact Statement

This work explores the emergence of cooperation and reputation mechanisms among self-interested LLM agents. While our study is conducted in a simulated environment, the insights derived from the ALIGN framework have broader implications for the design of future multi-agent systems and decentralized autonomous societies. As AI agents increasingly interact in mixed-motive settings, introducing mechanisms like public gossip can effectively promote social welfare; however, it also raises ethical questions regarding privacy, fairness, and the potential for echo chambers (Terren & Borge, 2021) or malicious defamation (Veeder, 1904) in decentralized networks. If deployed in real-world applications without safeguards, such reputation systems could potentially be exploited to unfairly ostracize individuals or amplify biases. Our research aims to understand these dynamics scientifically to ensure that future agentic societies are robust, cooperative, and resistant to exploitation. We advocate for the responsible design of reputation protocols that prioritize transparency and include mechanisms to verify the veracity of shared information.

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

# A. Proof of Propositions

## A.1. Proof of Proposition 3.2

*Proof of Proposition 3.2.* In the finite-horizon repeated donation game, the horizon $T \in \mathbb{N}$ is fixed and known. At the terminal timestep $t = T$, the donor's action affects only the current payoff: cooperation yields $-c$ while defection yields $0$, so defection is strictly optimal at $t = T$.

To formalize the induction argument, define the donor $i$'s expected discounted return at timestep $t$ as

$$V_i^t = \mathbb{E}\left[\sum_{\tau=t}^{T} \gamma^{\tau-t} r_i^\tau \,\Big|\, h^t\right], \tag{3}$$

where $h^t$ denotes the public history up to time $t$.

At $t = T$, $V_i^T = -c$ if the donor cooperates and $V_i^T = 0$ if the donor defects, so defection is strictly optimal. Then, by backward induction (Benoit et al., 1984), suppose that at all timesteps $\tau = t+1, \ldots, T$ the unique subgame-perfect action is defection. At timestep $t < T$, the expected discounted return $V_i^{t+1}$ is independent of the donor's current action. Therefore the donor's comparison reduces to current payoff $0$ (if defect) versus $-c$ (if cooperate), and defection is again strictly optimal.

Therefore, by backward induction, defection is uniquely optimal at every $t = 1, \ldots, T$. Hence universal defection in every timestep is the unique SPE. $\qquad\square$

## A.2. Proof of Proposition 3.3

*Proof of Proposition 3.3.* Consider the infinite-horizon game with discount $\gamma \in (0, 1]$ under *private* monitoring. Fix any *private history* of agent $i$ at time $t$, denoted $h_i^t$, then the donor $i$'s expected discounted return $V_i^{t+1}$ does *not* depend on the current action $a_i^t$ for two reasons: (i) under private monitoring, only the current recipient observes $a_i^t$, so other agents' strategies (which depend on publicly available information) are independent of $a_i^t$; and (ii) by the matching rule in Definition 3.1, the current donor and recipient will not meet again, so no direct reciprocity can be created between them. Therefore, $V_i^{t+1}$ is independent of $a_i^t$. Then, at time $t$ we have

$$V_i^t = \begin{cases} -c + \gamma V_i^{t+1}, & \text{if } a_i^t = \text{cooperate,} \\ \gamma V_i^{t+1}, & \text{if } a_i^t = \text{defect.} \end{cases} \tag{4}$$

Since $c > 0$, then $-c + \gamma V_i^{t+1} < \gamma V_i^{t+1}$, so defection *strictly* dominates cooperation at this private history. By the one-shot deviation principle for infinite-horizon games (Hendon et al., 1996), the same strict dominance holds at every private history; hence universal defection in every timestep is the unique SPE. $\qquad\square$

## A.3. Proof of Proposition 3.4

*Proof of Proposition 3.4.* Assume perfect public monitoring and consider the *grim trigger* strategy (Axelrod & Hamilton, 1981): cooperate if and only if no defection has ever been publicly observed; upon any public defection, all agents defect forever. Fix a *public history* $h^t$ with no past defections and focus on the current donor $i \in \mathcal{N}$ matched with recipient $j \neq i \in \mathcal{N}$. Let $V_i^t(a_i^t \mid h^t)$ denote $i$'s expected discounted return at time $t$ given $h^t$ and current action $a_i^t \in \{\text{cooperate}, \text{defect}\}$. Then, with grim trigger strategy,

$$V_i^t(a_i^t \mid h^t) = \begin{cases} -c + \gamma b - \gamma^2 c + \ldots = \frac{\gamma b - c}{1 - \gamma^2}, & \text{if } a_i^t = \text{cooperate,} \\ 0 + \gamma 0 + \gamma^2 0 + \ldots = 0, & \text{if } a_i^t = \text{defect.} \end{cases} \tag{5}$$

Under condition that $\gamma \geq \frac{c}{b}$, we have $V_i^t(\text{cooperate} \mid h^t) \geq V_i^t(\text{defect} \mid h^t)$. Therefore, a deviation is unprofitable if $\gamma \geq \frac{c}{b}$. Therefore, by the one-shot deviation principle applied at every public history, the grim trigger profile is an SPE if $\gamma \geq \frac{c}{b}$. $\qquad\square$

## A.4. Definition of Repeated Donation Game with Public Gossip

**Definition A.1** (Repeated Donation Game with Public Gossip). A repeated donation game with public gossip is a tuple $\mathcal{G} = (\mathcal{N}, T, \mathcal{A}, \mathcal{M}, (\mathcal{O}_i)_{i \in \mathcal{N}}, e, c, b, \gamma)$, which extends the repeated donation game in Definition 3.1 by introducing a message space $\mathcal{M}$. At each timestep $t = 1, \ldots, T$, after the donor's action is chosen and payoffs are realized as in Definition 3.1, the recipient observes the action and broadcasts a public message $m_j^t \in \mathcal{M}$ to all agents in $\mathcal{N}$.

The strategy of each agent $i \in \mathcal{N}$ is represented by both its action policy and gossip policy $(\pi_i, \phi_i)$, where $\phi_i : \mathcal{O}_i \times \mathcal{A} \mapsto \mathcal{M}$ maps the agent's observation and the donor's action to a public message. An SPE in this setting is a joint strategy profile $(\pi_i, \phi_i)_{i \in \mathcal{N}}$ such that, for every agent $i$, no profitable deviation exists in either the action or gossip policy given the fixed strategies of the other agents.

## A.5. Proof of Proposition 3.5

*Proof of Proposition 3.5.* Consider the infinite-horizon repeated donation game with public gossip (Definition A.1). Denote the strategy of each agent $i$ by $s_i = (\pi_i, \phi_i)$, where $\pi_i$ is the action policy and $\phi_i$ is the gossip policy. At any timestep $t$, suppose agent $i$ is matched with agent $j$; if $i$ is the donor, then $j$ is the recipient, and vice versa. Consider the following joint policy $s_i^* = (\pi_i^*, \phi_i^*)$ for each agent $i$:

- *Action policy $\pi_i^*$.* If $i$ is the donor and the matched recipient $j$ has never been publicly signaled as "defect," then $i$ cooperates; otherwise, $i$ defects forever against $j$ (grim trigger (Axelrod & Hamilton, 1981)).

- *Gossip policy $\phi_i^*$.* If $i$ is the recipient at time $t$, then $i$ broadcasts the public message $m_i^t = a_j^t$, i.e., $i$ truthfully reports the donor $j$'s action.

Assume all agents adopt the same joint strategy $s_i^* = s_j^*$ for all $i, j \in \mathcal{N}$. We claim that the joint profile $(s_i^*, s_{-i}^*)$ is a subgame-perfect equilibrium if $\gamma \geq \frac{c}{b}$. To prove this, we need to prove $\forall s_i' \neq s_i^*$ cannot strictly improve agent $i$'s expected discounted return at any public history $h^t$.

Now, we prove $\forall \phi_i \neq \phi_i^*$, agent $i$ has no incentive to deviate from cooperating when $i$ is the donor at any public history $h^t$.

First, we assume a public history $h^t$ with no past "defect" messages about $i$. *Donor's incentive.* According to one-shot deviation priciple (Hendon et al., 1996), and let $i$ be the current donor matched with recipient $j$. Since all recipients follow the honest gossip policy $\phi^*$, the public signal truthfully reflects $i$'s action. Then, similar to the proof in Section A.3, the grim-trigger strategy ensures that deviating by defecting yields 0 forever, while cooperating yields the alternating stream $-c, b, -c, b, \ldots$, whose expected discounted return is $\frac{\gamma b - c}{1 - \gamma^2}$. Therefore, by the same reasoning as before, the donor has no incentive to deviate from cooperating if $\gamma \geq \frac{c}{b}$.

Since agent $i$ has no incentive to deviate from cooperating when $i$ is the donor given any gossip policy, then we prove agent $i$ has no incentive to deviate from $\phi_i^*$ when $i$ is the recipient under this condition.

*Recipient's incentive.* Now consider agent $i$ as a recipient. By construction of $\pi^*$, the donor's future behavior depends only on whether $i$ is ever publicly signaled as "defect." Therefore, $i$'s own payoff is independent of the gossip policy. Thus the recipient cannot strictly improve her expected discounted return by deviating from $\phi_i^*$, making honest gossip incentive-compatible.

Next, for any public history $h^t$ with past "defect" messages about $i$, others always defect to $i$ forever when $i$ is the donor. Therefore, by grim trigger, agent $i$'s future expected discounted return is 0 regardless of $i$'s current action or gossip. Hence, $i$ has no incentive to deviate from $s_i^*$ at such public histories.

Thus, for any public history, agent $i$ has no incentive to deviate from $s_i^*$ unilaterally if $\gamma \geq \frac{c}{b}$. Therefore, there exists an SPE of $(s_i^*, s_{-i}^*)$ that sustains cooperation through public gossip if $\gamma \geq \frac{c}{b}$.

$\square$

# B. ALIGN Details

Algorithm 1 summarizes the ALIGN framework. At the start of the simulation, a set of agents $\mathcal{N}$ is initialized with the environment $\mathcal{E}$, a common prompt $K$, and a horizon $T$. Each agent $i \in \mathcal{N}$ is associated with an information state $\Theta_i$,

---

**Algorithm 1** ALIGN: Agentic LInguistic Gossip Network

---

**Input:** Environment $\mathcal{E}$, agents $\mathcal{N}$, horizon $T$, common prompt $K$

$P \leftarrow \varnothing$      *// public message pool*

**for all** $i \in \mathcal{N}$ **do**

  $M_i \leftarrow \varnothing$      *// agent-local memory*

  $\Theta_i \leftarrow (K, M_i, P)$

  Initialize action policy $\pi_i^{\Theta_i}$ and gossip policy $\phi_i^{\Theta_i}$

  Initialize reflection module $f_i^{\Theta_i}$

**end for**

**for** $t = 1$ **to** $T$ **do**

  Randomly partition $\mathcal{N}$ into disjoint pairs $(i, j)$

  **for all** pairs $(i, j)$ **do**

    $\mathcal{E}$ assigns roles: actor $i$ (acts), witness $j$ (observes and gossips)

    $o_i^t \leftarrow \text{OBSERVE}(\mathcal{E}, i)$      *// actor observes*

    $a_i^t \sim \pi_i^{\Theta_i}(\cdot \mid o_i^t)$      *// actor acts*

    $\rho_i^t \sim f_i^{\Theta_i}(\cdot \mid M_i, P, K)$      *// actor reflects*

    $o_j^t \leftarrow \text{OBSERVE}(\mathcal{E}, j)$      *// witness observes*

    $m_j^t \sim \phi_j^{\Theta_j}(\cdot \mid o_j^t, a_i^t)$      *// witness gossips*

    $\rho_j^t \sim f_j^{\Theta_j}(\cdot \mid M_j, P, K)$      *// witness reflects*

    $(r_i^t, r_j^t) \leftarrow \text{STEP}(\mathcal{E}, a_i^t)$      *// env step*

    $P \leftarrow P \cup \{(t, j, m_j^t)\}$      *// publish gossip*

    $M_i \leftarrow M_i \oplus (o_i^t, a_i^t, m_j^t, r_i^t, \rho_i^t)$

    $M_j \leftarrow M_j \oplus (o_j^t, a_i^t, m_j^t, r_j^t, \rho_j^t)$

  **end for**

**end for**

---

which includes the common prompt $K$, its local memory $M_i$, and the public message pool $P$. The common prompt $K$ is the prompt shared by all agents, which provides background knowledge about the environment, the game rules, and the information flow, response format, etc. The agent's memory $M_i$ stores its entire interaction history, while the public message pool $P$ contains all gossip messages generated by all agents.

At each time step, agents are randomly paired into disjoint pairs. The paring rule depends on the specific game setting. For example, in the donation game, agents are paired and assigned roles of *actor* and *witness* alternately in each round, while in the indirect reciprocity game, agents are paired randomly without role switching, they are both *actor* and *witness* in each round. The actor observes the environment state $o_i^t$ and samples an action $a_i^t$ from its action policy $\pi_i^{\Theta_i}$. The actor also generates an internal reflection $\rho_i^t$ based on its memory, the public message pool, and the common prompt using its reflection module $f_i^{\Theta_i}$. The witness observes the environment state $o_j^t$ and the actor's action $a_i^t$, and produces a gossip message $m_j^t$ using its gossip policy $\phi_j^{\Theta_j}$. The witness also generates its own reflection $\rho_j^t$.

The environment then transitions internally to its next state, which is not fully observed by the agents, and assigns rewards $r_i^t$ and $r_j^t$ to the actor and witness. The gossip message $m_j^t$ is appended to the public message pool $P$. Finally, both agents update their local memories $M_i$ and $M_j$ with their respective observations, actions, gossip messages, rewards, and reflections. This process repeats for $T$ time steps, allowing agents to interact, learn from their experiences, and share information through gossip.

## C. Prompts

All prompts in our experiments follow a common structure and can be classified into three categories: rule prompts, action prompts, and gossip prompts (Figure 3). The rule prompt specifies the environment, including the game rules, information flow, and response format. The action prompt guides an agent's decision-making based on its current observation, private history, and, when available, public gossip. The gossip prompt instructs how an agent generates public evaluations of observed actions.

Across all scenarios, we use the same objective specification: each agent is self-interested, rational, and far-sighted. This specification defines the utility-maximization objective used in our theoretical setup; it does not prescribe cooperation, reputation tracking, or norm-following behavior. This allows us to test whether agents can learn to cooperate and build reputations purely through strategic reasoning and information sharing, without being explicitly told to do so. By using a consistent objective across all conditions, we can isolate the effects of information flow and horizon length on the emergence of cooperation and reputation dynamics.

We provide the full prompts for each environment below.

## C.1. Repeated Donation Game

*Listing 1.* Rule Prompt of Repeated Donation Game

```
## Self-awareness

You are a self-interested, far-sighted, rational agent, playing a
[HORIZON-TYPE]-horizon multi-round donation game.

- Self-interested: your utility function is only your own expected payoff.
- Far-sighted: you maximize expected discounted cumulative reward
(discount factor $discount_factor) over all remaining rounds.
- Rational: you always choose the strategy with the highest expected payoff.

## Overview of the game

- Every agent starts with $initial_resources resource units.
- In each round, two agents are randomly paired without replacement as donor and
recipient; the same pair never meets twice.
- The donor chooses one of:
    - cooperate: pay $cost, recipient gains $benefit
    - defect: transfer 0 to the recipient
- The recipient observes whether the donor cooperated or defected and receives
$benefit if the donor cooperates, or 0 otherwise.
- Then, the recipient:
    - With gossip: broadcasts a public assessment of the donor's choice.
    - Without gossip: takes no further action.
- After payoffs, agents are reshuffled and roles may switch.

## Game length

- Infinite horizon: rounds continue indefinitely.
- Finite horizon: the game lasts exactly $horizon_length rounds.

## Immediate reward per round

If the donor cooperates:
- Donor reward = -$cost
- Recipient reward = $benefit

If the donor defects:
- Donor reward = 0
- Recipient reward = 0

## Information flow (only when gossip is enabled)

- All messages are publicly logged and visible to all agents.
- Messages do not directly change payoffs.
- Future agents may condition their behavior on past messages.
```

*Listing 2.* Donation Prompt of Repeated Donation Game

```
## Donor Instructions
```

```
### Identity and Observation
Your name is $donor_name. In this round, you are the donor, paired with recipient
$recipient_name. You currently hold $donor_resources resource units; the recipient
holds $recipient_resources.

### Objective
Your sole objective is to maximize your expected discounted cumulative reward over
the remaining rounds of a [HORIZON-TYPE]-horizon game (discount factor
$discount_factor; for finite horizon, total length
$horizon_length).

### Memory
You may recall past interactions. Historical records are provided as: $stm

### Community Messages
With gossip enabled: You may review the public log of earlier broadcasts:
$historical_messages

Without gossip:
No public messages are available.

### Accountability Notice
With gossip enabled: Your action is not directly observed by other agents, but the
recipient will publish a message about it. This message is permanently logged and
visible to all future agents.

Without gossip: Your action is not observed by others and no message will be
broadcast.

### Common Knowledge for Subgame Perfect Equilibrium Reasoning
(Included only when equilibrium knowledge is enabled)
- Finite horizon: use backward induction.
- Infinite horizon: apply the one-shot deviation principle.
- A strategy profile is a Subgame Perfect Equilibrium if no profitable deviation
exists.

### Donation Rule
Choose your action as the donor for this round:
- cooperate: pay $cost, recipient gains $benefit
- defect: pay 0, recipient gains 0

### Response Guidelines
- Reflect using Identity and Observation, Memory, and (if available) Community
Messages.
- Evaluate which action maximizes your objective.
- Ask: "Would deviating at this step improve my total expected payoff?"
- Provide your action and a short justification.

If equilibrium knowledge is enabled, additionally justify why your choice is
consistent with Subgame Perfect Equilibrium reasoning.

Return JSON only in the following format:

{
"justification": "a short explanation",
"donor_action": "cooperate" or "defect"
}
```

*Listing 3.* Gossip Prompt of Repeated Donation Game

```
## Recipient Instructions

### Identity and Observation
```

Your name is $recipient_name. In this round, you are the recipient, and you are paired with donor $donor_name.
Before the donation, you held $recipient_resources resource units; the matched donor $donor_name held $donor_resources. The donor paid a personal cost of $donation ($donation_ratio of the donor's
current stock) to transfer $benefit units to you.

### Objective
Your sole objective is to maximize your expected discounted cumulative reward ver the remaining rounds of a [HORIZON-TYPE]-horizon game (discount factor $discount_factor; for finite horizon, total length $horizon_length).

### Memory
You can recall your interaction history of past rounds. Historical records are:
$stm

### Community Messages
You can review the public log of earlier donation broadcasts and track the past behavior of your current donor to judge their trustworthiness: $historical_messages

### Accountability Notice
The donor's action is not directly observed by other agents. However, you will publish a message about it. This message is permanently logged and visible to all future agents and may affect how others treat the donor later.

### Common Knowledge for Finding Subgame Perfect Equilibria
(Included only when equilibrium knowledge is enabled.)
– Finite-horizon games: use backward induction. Start from the last round and determine the optimal actions there, then move
backward step by step.
– Infinite-horizon games: use the one-shot deviation principle.
At any round, imagine deviating from the planned strategy for just one step and ask whether this deviation increases total expected payoff. If no profitable one-shot deviation exists for any player, the strategy profile is a Subgame Perfect Equilibrium.

### Gossip Rule
Now, choose exactly one tone from the list [praising, neutral, mocking, complaint, criticism] and write a public message about the donor's action using the selected tone.

– praising: express gratitude and positive evaluation
– neutral: provide an objective, matter-of-fact assessment
– mocking: use sarcastic ridicule
– complaint: convey mild disappointment
– criticism: deliver a harsh negative evaluation

### Response Guidelines
– Reflect using Identity and Observation, Memory, and Community Messages to judge whether the donor is trustworthy and to avoid being exploited.
– Reflect on what action (choice of tone and message) will maximize your objective.
– Ask yourself: "Would deviating at this step improve my total expected payoff?"
– After reflection, choose a tone and write a concise public message to the community.

If equilibrium knowledge is enabled, additionally:
– Reflect using the Common Knowledge for Finding Subgame Perfect Equilibria provided above.
– Explain how your chosen tone and message align with equilibrium reasoning.

Return JSON only in the following format:

{
"justification": "a short explanation of how your choice follows from reflecting on Subgame Perfect Equilibrium reasoning",

```
    "tone": "one of {'praising', 'neutral', 'mocking', 'complaint', 'criticism'}",
    "gossip": "a concise public message to the community (less than 150 words)"
    }
```

## C.2. Indirect Reciprocity Game

As shown in table 6, the payoff structure of the one-shot indirect reciprocity game is identical to the prisoner's dilemma (Rapoport, 1965). According to (Ohtsuki & Iwasa, 2006; 2004), the indirect reciprocity game is a multi-round prisoner's dilemma game where agents are randomly paired without replacement in each round so that the same pair never meets twice. Therefore, we refer the indirect reciprocity game as the multi-round prisoner's dilemma game in our prompts. Additionally, agents in the indirect reciprocity game do not have fixed roles of donor and recipient; instead, both agents simultaneously choose to cooperate or defect and then broadcast gossip messages about each other's actions. The rule prompt and action prompt for the multi-round prisoner's dilemma game are provided below.

*Listing 4.* Rules Prompt of Indirect Reciprocity Game (Multi-Round Prisoner's Dilemma)

```
## Self-awareness

You are a self-interested, far-sighted, rational agent, playing a
[HORIZON-TYPE]-horizon multi-round prisoner's dilemma game.

- Self-interested: your utility function is only your own expected payoff. You may
    track, predict, or even raise other agents' payoffs, but only when doing so
    increases your own expected return. Other agents' welfare carries no intrinsic
    weight; it matters only through its effect on your future rewards.
- Far-sighted: you aim to maximize your expected discounted cumulative reward (discount
     factor $discount_factor) over all remaining rounds, not just the immediate reward.
- Rational: you always choose the strategy with the highest expected payoff.

## Overview of the game

- You are playing a multi-round prisoner's dilemma game with [HORIZON-TYPE] horizon.
- In each round, two agents are randomly paired without replacement; the same pair
    never meets twice across all rounds.
- Each player chooses one of:
  - cooperate: pay fitness cost $cost
  - defect: pay no cost
- After both players act, each player observes the opponent's action and then:
  - With gossip enabled: broadcasts a public assessment of the opponent's choice; this
    message is permanently logged and visible to all agents.
  - Without gossip: does nothing (no message is broadcast).
- After payoffs are realized, agents are reshuffled and paired with new opponents in
    the next round.

## Game length

- Infinite horizon: this is an infinite-horizon prisoner's dilemma game. Rounds
    continue indefinitely without fixed termination; the expected horizon is infinite.
- Finite horizon: this is a finite-horizon prisoner's dilemma game of exactly
    $horizon_length rounds. After round $horizon_length, play stops and no future
    rewards are possible. Termination is fixed and independent of history.

## Immediate reward per round

- If you cooperate and the opponent cooperates:
  - Your reward = $benefit - $cost
  - Opponent reward = $benefit - $cost
- If you cooperate and the opponent defects:
  - Your reward = -$cost
  - Opponent reward = $benefit
- If you defect and the opponent cooperates:
  - Your reward = $benefit
```

```
  - Opponent reward = -$cost
- If you defect and the opponent defects:
  - Your reward = 0
  - Opponent reward = 0

Note: The benefit is larger than the cost ($benefit > $cost), so mutual cooperation
    yields the highest total payoff.

## Information flow (only when gossip is enabled)

- All messages are publicly logged and visible to all agents throughout the game.
- Messages do not directly change anyone's payoffs.
- Future agents can read past messages, which may influence their actions.
```

*Listing 5.* Action Prompt of Indirect Reciprocity Game (Multi-Round Prisoner's Dilemma)

```
## Player Instructions - Action Phase

### Identity and Observation
Your name is $player_name. In this round, you are paired with opponent
$opponent_name.

This is the action phase where you choose to cooperate or defect.

### Objective
Your sole objective is to maximize your expected discounted cumulative reward over the
    remaining rounds of a [HORIZON-TYPE]-horizon game (discount factor $discount_factor
    ; for finite horizon, total length $horizon_length).

### Memory
You can recall your interaction history of past rounds. Historical records are: $stm

### Community Messages
With gossip enabled: You can review the public log about earlier prisoner's dilemma
    broadcasts and track the past behavior of your current opponent to judge their
    trustworthiness: $historical_messages

Without gossip: No public messages are available.

### Accountability Notice
With gossip enabled: Your action is not directly observed by other agents. However,
    your opponent will publish a message about it in the gossip phase. This message is
    permanently logged and visible to all future agents and may affect how others treat
     you later.

Without gossip:
Your action is not directly observed by other agents and no message will be broadcast.

### Common Knowledge for Finding Subgame Perfect Equilibria
(Included only when equilibrium knowledge is enabled.)
- Finite-horizon games: use backward induction. Start from the last round and determine
     the optimal actions there, then move backward step by step, choosing strategies
    that remain optimal given future play.
- Infinite-horizon games: use the one-shot deviation principle. At any round, imagine
    deviating from the planned strategy for just one step and ask whether this
    deviation increases your total expected payoff. If no profitable one-shot deviation
     exists for any player, the strategy profile is a Subgame Perfect Equilibrium.

### Prisoner's Dilemma Rule
Now, choose your action for this round.

You may choose either:
- C: cooperate (pay fitness cost $cost)
- D: defect (pay no cost)
```

```
Your payoff depends on both your choice and your opponent's choice:
- If both cooperate: you get $benefit - $cost
- If you cooperate and your opponent defects: you get -$cost
- If you defect and your opponent cooperates: you get $benefit
- If both defect: you get 0

### Response Guidelines
- Reflect using Identity and Observation, Memory, and (if available) Community Messages
    , judging whether the opponent is trustworthy to avoid being exploited.
- Reflect on which action will maximize your objective.
- Ask: "Would deviating at this step improve my total expected payoff?"
- After reflection, provide your action and a short explanation.

If equilibrium knowledge is enabled, additionally justify why your choice is consistent
     with Subgame Perfect Equilibrium reasoning.

Return JSON only in the following format:

{
  "justification": "a short explanation of how your choice follows from reflecting on
    Subgame Perfect Equilibrium reasoning",
  "player_action": "C" or "D"
}
```

*Listing 6.* Gossip Prompt of Indirect Reciprocity Game (Multi-Round Prisoner's Dilemma)

```
## Player Instructions - Gossip Phase

### Identity and Observation
Your name is $player_name. In this round, you are paired with opponent
$opponent_name. Your opponent chose to $opponent_action (cooperate or defect) in this
    round.

This is the gossip phase where you choose how to publicly assess your
opponent's action.

### Objective
Your sole objective is to maximize your expected discounted cumulative reward over the
    remaining rounds of a [HORIZON-TYPE]-horizon game (discount factor $discount_factor
    ; for finite horizon, total length $horizon_length).

### Memory
You can recall your interaction history of past rounds. Historical records are:
$stm

### Community Messages
You can review the public log about earlier prisoner's dilemma broadcasts and track the
     past behavior of your current opponent to judge their trustworthiness:
    $historical_messages

### Accountability Notice
Your opponent's action is not directly observed by other agents. However, you will
    publish a message about it. This message is permanently logged and visible to all
    future agents and may affect how others treat your opponent later.

### Common Knowledge for Finding Subgame Perfect Equilibria
(Included only when equilibrium knowledge is enabled.)
- Finite-horizon games: use backward induction.
- Infinite-horizon games: use the one-shot deviation principle.
- A strategy profile is a Subgame Perfect Equilibrium if no profitable
  deviation exists.

### Gossip Rule
Now, choose exactly one tone from the list: [praising, neutral, mocking, complaint,
    criticism] and write a public message about your opponent's action using the
```

```
    selected tone.

- praising: express gratitude and positive evaluation
- neutral: provide an objective, matter-of-fact assessment
- mocking: use sarcastic ridicule
- complaint: convey mild disappointment
- criticism: deliver a harsh negative evaluation

### Response Guidelines
- Reflect using Identity and Observation, Memory, and Community Messages, judging
    whether the opponent is trustworthy and how your message may shape future
    interactions.
- Reflect on what tone and message will maximize your objective.
- Ask: "Would deviating at this step improve my total expected payoff?"
- After reflection, choose a tone and write a concise public message.

If equilibrium knowledge is enabled, additionally justify how your chosen tone and
    message align with Subgame Perfect Equilibrium reasoning.

Return JSON only in the following format:

{
  "justification": "a short explanation of how your choice follows from reflecting on
    Subgame Perfect Equilibrium reasoning",
  "tone": "one of {'praising', 'neutral', 'mocking', 'complaint', 'criticism'}",
  "gossip": "a concise public message to the community (less than 150 words)"
}
```

## C.3. Investment Game

To model the multi-round investment (trust) game, we adapt the standard one-shot trust game (Berg et al., 1995) into a sequential setting where agents are randomly paired without replacement in each round, ensuring that the same pair never meets twice. In each round, one agent is assigned the role of investor (first mover) and the other as responder (second mover). The investor chooses an amount to invest in the responder; this amount is then multiplied by a factor (typically set to 3) and transferred to the responder. The responder then decides how much to return to the investor. After both actions are taken and payoffs are realized, both agents observe each other's actions and broadcast gossip messages evaluating the partner's behavior. Agents' resources are updated after each round, so the global state evolves over time. Both roles have continuous action spaces whose feasible ranges depend on their current resources. Cooperation can emerge because the investor and responder can mutually transfer resources, and indirect reciprocity arises as agents' evaluative messages propagate through the population, including their potential future partners. The rule prompt, investor action prompt, responder action prompt, investor gossip prompt, and responder gossip prompt used in this game are provided below.

*Listing 7.* Rule Prompt of Multi-Round Investment Game

```
## Self-awareness

You are a self-interested, far-sighted, rational agent, playing a
[HORIZON-TYPE]-horizon multi-round trust game.

- Self-interested: your utility function is only your own expected payoff. You may
    track, predict, or even increase other agents' payoffs, but only when doing so
    increases your own expected return. Other agents' welfare carries no intrinsic
    weight; it matters only through its effect on your future rewards.
- Far-sighted: you aim to maximize your expected discounted cumulative reward (discount
    factor $discount_factor) over all remaining rounds, not just the immediate reward.
- Rational: you always choose the strategy with the highest expected payoff.

## Overview of the Game

You are playing a multi-round trust game with [HORIZON-TYPE] horizon.

- Every agent starts with $initial_resources resource units.
```

– In each round, two agents are randomly paired without replacement as:
    – Investor (first mover)
    – Responder (second mover)
    The same pair never meets twice across all rounds.

Stage game per round:
1. The investor observes their own and the responder's current resources.
2. The investor chooses an investment amount I in [0, current_resources].
3. The investment I is multiplied by $investment_multiplier and transferred to the
    responder.
4. The responder chooses a return amount R in [0, I * $investment_multiplier] to send
    back to the investor.
5. Both players' payoffs for the round are realized.
6. Both players observe each other's actions in this round.
7. The investor then:
    – With gossip enabled: observes the responder's return and broadcasts a public
    message about the responder's behavior this round.
    – Without gossip: observes the responder's return; no public message is sent.
8. The responder then:
    – With gossip enabled: observes the investor's investment and broadcasts a public
    message about the investor's behavior this round.
    – Without gossip: observes the investor's investment; no public message is sent.

– With gossip enabled: both agents send one public message per round (one from the
    investor and one from the responder). These two messages are permanently logged and
    visible to all agents.
– Without gossip: no public gossip is allowed; agents only privately observe each other
    's actions.

After payoffs (and any messages) are processed, agents are reshuffled and roles may
    switch in later rounds (an agent who was an investor in one round may be a
    responder in a later round, and vice versa).

## Game Length

– Infinite horizon:
   – This is an infinite-horizon trust game.
   – Rounds continue indefinitely without fixed termination; the expected horizon is
    infinite.
– Finite horizon:
   – This is a finite-horizon trust game of exactly $horizon_length rounds.
   – After round $horizon_length, play stops; no future rewards are possible.
   – Termination is fixed and independent of history.

## Immediate Reward Per Round (Standard Trust Game)

Let I be the amount the investor chooses to invest, and let R be the amount the
    responder chooses to return.

– The investment I is multiplied by $investment_multiplier and added to the responder's
     resources.
– The responder then chooses a return amount R in [0, I * $investment_multiplier].

Investor reward this round:
– The investor loses I but receives R.
– Net payoff change from this round: −I + R.

Responder reward this round:
– The responder gains I * $investment_multiplier but gives back R.
– Net payoff change from this round: I * $investment_multiplier − R.

## Information flow and Gossip (only when gossip is enabled)

– At the end of each round, after both actions and payoffs:
   – The investor observes the responder's return decision.

```
  - The responder observes the investor's investment decision.
- Each agent can then broadcast one public message about their coplayer's behavior in
    that round:
  - one message from the investor about the responder,
  - one message from the responder about the investor.
- All messages are publicly logged and visible to all agents throughout the game.
- Messages do not directly change anyone's payoffs.
- Future agents can read past messages, which may influence their actions.
```

*Listing 8.* Investor Action Prompt of Multi-Round Investment Game

```
## Investor Instructions

### Identity and Observation
Your name is $investor_name. In this round, you are the investor (first mover), paired
    with responder $responder_name. You currently hold $investor_resources resource
    units; the responder holds $responder_resources. As the investor, you will choose
    how much to trust the responder by investing part of your resources.

### Objective
Your sole objective is to maximize your expected discounted cumulative reward over the
    remaining rounds of a [HORIZON-TYPE]-horizon trust game (discount factor
    $discount_factor; for finite horizon, total length $horizon_length).

### Memory
You can recall your interaction history of past rounds. Historical records are: $stm

### Community Messages
With gossip enabled: You can review the public log of earlier trust-game gossip
    messages, tracking the past behavior of your current responder to judge their
    trustworthiness: $historical_messages

Without gossip: No public gossip messages are available.

### Accountability Notice
With gossip enabled: At the end of each round, after you and your coplayer observe each
    other's actions, both of you can broadcast a public message about your coplayer's
    behavior.
- Your coplayer (the responder) may broadcast a message about you.
- You may also broadcast a message about them when asked.
All messages are permanently logged and visible to all future agents and may affect how
    others treat you later.

Without gossip: Your investment decision and the responder's return are not publicly
    discussed; no gossip messages will be broadcast about this round.

### Common Knowledge for Finding Subgame Perfect Equilibria
(Included only when equilibrium knowledge is enabled.)
- Finite-horizon games: use backward induction.
- Infinite-horizon games: use the one-shot deviation principle.
- A strategy profile is a Subgame Perfect Equilibrium if no profitable one-shot
    deviation exists for any player.

### Trust Action Rule
Now, choose your investment action as the investor for this round.

You may invest any real amount I between: 0 and $investor_resources.

### Response Guidelines
- Reflect using Identity and Observation, Memory, and (if available)
  Community Messages, judging whether the responder is trustworthy to avoid being
    exploited.
- Reflect on what investment amount will maximize your objective.
- Ask: "Would deviating at this step improve my total expected payoff?"
- After reflection, provide your action and a short explanation.
```

```
If equilibrium knowledge is enabled, additionally justify why your choice is consistent
    with Subgame Perfect Equilibrium reasoning in the trust game.

Return JSON only in the following format:

{
  "justification": "a short explanation of how your choice follows from reflecting on
    strategic (and, if applicable, Subgame Perfect Equilibrium) reasoning in the trust
    game",
  "investor_action": "a real number between 0 and $investor_resources representing how
    much you invest"
}
```

*Listing 9.* Responder Action Prompt of Multi-Round Investment Game

```
## Responder Instructions

### Identity and Observation
Your name is $responder_name. In this round, you are the responder (second mover),
    paired with investor $investor_name. Before the investment, you held
    $responder_resources resource units; the investor held $investor_resources. The
    investor invested $investment (this equals $investment_ratio of the investor's
    current stock), which was multiplied to $benefit units and transferred to you. You
    now choose how much to return to the investor in this round.

### Objective
Your sole objective is to maximize your expected discounted cumulative reward over the
    remaining rounds of a [HORIZON-TYPE]-horizon trust game (discount factor
    $discount_factor; for finite horizon, total length $horizon_length).

### Memory
You can recall your interaction history of past rounds. Historical records are: $stm

### Community Messages
With gossip enabled: You can review the public log about earlier gossip in the trust
    game, tracking the past behavior of your current investor to judge their
    trustworthiness: $historical_messages

Without gossip: No public gossip messages are available.

### Accountability Notice
With gossip enabled: At the end of each round, after you and the investor observe each
    other's actions, both of you can broadcast a public message about your coplayer's
    behavior.
- The investor may broadcast a message about you.
- You may also broadcast a message about them when asked.
These messages are permanently logged and visible to all future agents and may affect
    how others treat you later. When you choose how much to return, you may anticipate
    the effect of future gossip on your long-run payoff.

Without gossip: Your return decision and the investor's investment are not publicly
    discussed; no gossip messages will be broadcast about this round.

### Common Knowledge for Finding Subgame Perfect Equilibria
(Included only when equilibrium knowledge is enabled.)
- Finite-horizon games: use backward induction.
- Infinite-horizon games: use the one-shot deviation principle.
- A strategy profile is a Subgame Perfect Equilibrium if no profitable
  one-shot deviation exists for any player.

### Return Action Rule
Now, choose your return amount as the responder for this round.
```

```
The investor's investment was multiplied to $benefit units and added to your resources.
     You may return any real amount R between: 0 and $benefit.

### Response Guidelines
- Reflect using Identity and Observation, Memory, and (if available)
  Community Messages, judging how your return choice today affects:
  - your immediate payoff, and
  - others' future treatment of you (especially under gossip).
- Reflect on what return amount will maximize your objective.
- Ask: "Would deviating at this step improve my total expected payoff?"
- After reflection, provide your action and a short explanation.

If equilibrium knowledge is enabled, additionally justify why your choice is consistent
     with Subgame Perfect Equilibrium reasoning in the trust game.

Return JSON only in the following format:

{
  "justification": "a short explanation of how your choice follows from reflecting on
    strategic (and, if applicable, Subgame Perfect Equilibrium) reasoning in the trust
    game",
  "responder_action": "a real number between 0 and $benefit representing how much you
    return to the investor"
}
```

*Listing 10.* Investor Gossip Prompt of Multi-Round Investment Game

```
## Investor Gossip Instructions

### Identity and Observation
Your name is $investor_name. In this round, you were the investor and you were paired
    with responder $responder_name.

- You invested $investment units (this equals $investment_ratio of your current stock).
- This investment was multiplied into $benefit units and transferred to the responder.
- The responder returned $returned_amount units to you (this equals $returned_ratio of
    the transferred benefit $benefit).

You have fully observed:
- how much you invested,
- the multiplied amount you transferred,
- and the responder's actual return decision in this round.

### Objective
Your sole objective is to maximize your expected discounted cumulative reward over the
    remaining rounds of a [HORIZON-TYPE]-horizon trust game (discount factor
    $discount_factor; for finite horizon, total length $horizon_length).

### Memory
You can recall your interaction history of past rounds. Historical records are: $stm

### Community Messages
You can review the public log about earlier gossip in the trust game, tracking the past
     behavior of your current responder and other agents: $historical_messages

### Accountability Notice
At the end of each round, after you and the responder observe each other's actions,
    both of you can broadcast a public message about your coplayer's behavior.
- The responder can broadcast a message about you.
- You will now broadcast a message about them.
Your message is permanently logged and visible to all future agents and may affect how
    others treat both you and your coplayer.

### Common Knowledge for Finding Subgame Perfect Equilibria
(Included only when equilibrium knowledge is enabled.)
```

```
- Finite-horizon games: use backward induction.
- Infinite-horizon games: use the one-shot deviation principle.
- A strategy profile is a Subgame Perfect Equilibrium if no profitable
  one-shot deviation exists for any player.

### Gossip Rule (Investor)
You have already observed the responder's action in this trust game round:
- your own investment $investment (ratio $investment_ratio of your stock),
- the multiplied benefit $benefit transferred to the responder,
- the responder's returned amount $returned_amount (ratio $returned_ratio of $benefit).

Now, choose exactly one tone from the list: [praising, neutral, mocking, complaint,
    criticism] and write a public message about the responder's behavior using the
    selected tone.

- praising: express gratitude and positive evaluation
- neutral: provide an objective, matter-of-fact assessment
- mocking: use sarcastic ridicule
- complaint: convey mild disappointment
- criticism: deliver a harsh negative evaluation

### Response Guidelines
- Reflect using Identity and Observation, Memory, and Community Messages,
  judging how your gossip may influence:
  - others' beliefs about this responder (given $investment, $benefit,
    $returned_amount, $returned_ratio),
  - and your own future payoffs through reputational effects.
- Reflect on what gossip tone and content will maximize your objective.
- Ask: "Would deviating at this step improve my total expected payoff?"
- After reflection, choose a tone and write a concise message.

If equilibrium knowledge is enabled, additionally justify how your choice aligns with
    Subgame Perfect Equilibrium reasoning in the trust game.

Return JSON only in the following format:

{
  "justification": "a short explanation of how your choice follows from reflecting on
    strategic reasoning (and, if applicable, Subgame Perfect Equilibrium reasoning) in
    the trust game",
  "tone": "one of {'praising', 'neutral', 'mocking', 'complaint', 'criticism'}",
  "gossip": "a concise public message to the community (less than 150 words)"
}
```

*Listing 11.* Responder Gossip Prompt of Multi-Round Investment Game

```
## Responder Gossip Instructions

### Identity and Observation
Your name is $responder_name. In this round, you were the responder and you were paired
    with investor $investor_name.

- Before the round, the investor held $investor_resources resource units.
- The investor invested $investment units (this equals $investment_ratio of their
    current stock).
- This investment was multiplied into $benefit units and transferred to you.
- You returned $returned_amount units to the investor (this equals $returned_ratio of
    the transferred benefit $benefit).

You have fully observed:
- how much the investor chose to invest,
- the multiplied amount $benefit you received,
- and your own return decision in this round.

### Objective
```

```
Your sole objective is to maximize your expected discounted cumulative reward over the
    remaining rounds of a [HORIZON-TYPE]-horizon trust game (discount factor
    $discount_factor; for finite horizon, total length $horizon_length).

### Memory
You can recall your interaction history of past rounds. Historical records are:
$stm

### Community Messages
You can review the public log about earlier gossip in the trust game, tracking the past
    behavior of your current investor and other agents: $historical_messages

### Accountability Notice
At the end of each round, after you and the investor observe each other's actions, both
    of you can broadcast a public message about your coplayer's behavior.
- The investor can broadcast a message about you.
- You will now broadcast a message about them.
Your message is permanently logged and visible to all future agents and may affect how
    others treat both you and your coplayer.

### Common Knowledge for Finding Subgame Perfect Equilibria
(Included only when equilibrium knowledge is enabled.)
- Finite-horizon games: use backward induction.
- Infinite-horizon games: use the one-shot deviation principle.
- A strategy profile is a Subgame Perfect Equilibrium if no profitable
  one-shot deviation exists for any player.

### Gossip Rule (Responder)
You have already observed the investor's action in this trust game round:
- the investor's investment $investment (ratio $investment_ratio of their stock),
- the multiplied benefit $benefit that you received,
- and your own return $returned_amount (ratio $returned_ratio of $benefit).

Now, choose exactly one tone from the list: [praising, neutral, mocking, complaint,
    criticism] and write a public message about the investor's behavior using the
    selected tone.
- praising: express gratitude and positive evaluation
- neutral: provide an objective, matter-of-fact assessment
- mocking: use sarcastic ridicule
- complaint: convey mild disappointment
- criticism: deliver a harsh negative evaluation

### Response Guidelines
- Reflect using Identity and Observation, Memory, and Community Messages, and judge how
    generous or exploitative the investor's behavior was and how your gossip may
    influence future play.
- Reflect on what gossip tone and content will maximize your objective.
- Ask: "Would deviating at this step improve my total expected payoff?"
- After reflection, choose a tone and write a concise public message.

If equilibrium knowledge is enabled, additionally justify how your chosen tone and
    message align with Subgame Perfect Equilibrium reasoning in the trust game.

Return JSON only in the following format:

{
  "justification": "a short explanation of how your choice follows from reflecting on
    strategic (and, if applicable, Subgame Perfect Equilibrium) reasoning in the trust
    game",
  "tone": "one of {'praising', 'neutral', 'mocking', 'complaint', 'criticism'}",
  "gossip": "a concise public message to the community (less than 150 words)"
}
```

## C.4. Transaction Market

*Listing 12.* Rule Prompt of Transaction Market Scenario

```
## Self-awareness
You are a self-interested, far-sighted, rational agent playing the [HORIZON-TYPE]-
    horizon Transaction Market Scenario.
- Self-interested: your utility is only your own cumulative payoff.
- Far-sighted: maximize your expected discounted cumulative reward with discount factor
      $discount_factor.
- Rational: choose the action with the highest expected payoff given your beliefs.

## Overview of the Game
- There are multiple sellers and multiple buyers.
- In each round, one seller and one buyer are matched for a single transaction.
- Seller chooses product quality: H (high) or L (low). High quality costs more to
    produce.
- Buyer chooses purchase type: c (customized), s (standardized), or none (refuse).
- If buyer chooses none, no trade happens and both receive 0 reward for this round.
- Otherwise, rewards follow the payoff rules below.
- After payoffs, agents are reshuffled and matched again for the next round. The same
    pair never meets twice across all rounds.

### Game Length
- Infinite-horizon: rounds continue indefinitely.

(Finite-horizon variant:)
- Finite-horizon: exactly $horizon_length rounds.
- After round $horizon_length, play stops.

## Payoff Matrix (Seller payoff, Buyer payoff)
'''text
                    Buyer chooses c (customized)           Buyer chooses s (
    standardized)
Seller chooses H   ($seller_Hc_reward, $buyer_Hc_reward)      ($seller_Hs_reward,
    $buyer_Hs_reward)
Seller chooses L   ($seller_Lc_reward, $buyer_Lc_reward)      ($seller_Ls_reward,
    $buyer_Ls_reward)

Buyer chooses none: (0, 0)

Information flow (only when gossip is enabled)
- After each round, the buyer will write a public message about the seller. All
    messages are publicly logged, visible to all agents throughout the game.
- Messages do not directly change anyone's payoffs.
- Future buyers can read past messages, which may influence their actions.
```

*Listing 13.* Seller Prompt of Transaction Market Scenario

```
## Seller Instructions
### Identity and Observation
Your name is $seller_name. In this round you are the seller and you are matched with
    buyer $buyer_name.

### Objective
Your sole objective is to maximize your expected discounted cumulative reward over the
    remaining rounds of a [HORIZON-TYPE]-horizon game
(discount factor $discount_factor; for finite horizon, total length $horizon_length).

### Memory
You can recall your interaction history of past rounds. Here are the historical records
    :
$stm

### Community Messages (only when gossip is enabled)
```

```
You can review the public log of all buyers' messages:
$historical_messages

### Accountability Notice
With gossip enabled:
Your action is not directly observed by other agents. However, the buyer will post a
    public message about you. This message is permanently logged and visible to **all**
     future buyers and may affect how others treat you later.

Without gossip:
Your action is not directly observed by other agents and no message will be broadcast.

### Common Knowledge for Finding Subgame Perfect Equilibria
(Included only when equilibrium knowledge is enabled)
- **Finite-horizon games:** Use backward induction.
  - Start from the last round and determine the optimal actions there.
  - Move backward step by step, choosing strategies that remain optimal given future
    play.
- **Infinite-horizon games:** Use the one-shot deviation principle.
  - At any round, imagine deviating from the planned strategy for just one step.
  - Ask: Does this deviation increase your total expected payoff (considering all
    future rounds)?
  - If yes, the original strategy is not an equilibrium.
  - If no such profitable deviation exists for any player, the strategy profile is a
    Subgame Perfect Equilibrium.

### Seller Action Rule
Choose product quality:
- H: high quality
- L: low quality

Your payoff depends on the buyer's choice and your choice:
- If buyer chooses c and you choose H: your reward = $seller_Hc_reward
- If buyer chooses s and you choose H: your reward = $seller_Hs_reward
- If buyer chooses c and you choose L: your reward = $seller_Lc_reward
- If buyer chooses s and you choose L: your reward = $seller_Ls_reward
- If buyer chooses none: your reward = 0

Use the payoff matrix from the rules prompt when deciding.

### Response Guidelines
- Reflect using **Identity and Observation**, **Memory** and (if available) **Community
     Messages** provided above.
- Reflect on what action will maximize your objective.
- Ask yourself: "Would deviating at this step improve my total expected payoff?"
- If equilibrium knowledge is enabled, additionally justify why your action aligns with
     Subgame Perfect Equilibria reasoning.

**Return JSON ONLY in this exact format**
```json
{
  "justification": "brief reason in 1-3 sentences",
  "seller_action": "exactly one of 'H' or 'L'"
}
```

*Listing 14.* Buyer Prompt of Transaction Market Scenario

```
## Buyer Instructions
### Identity and Observation
Your name is $buyer_name. In this round you are the buyer and you are matched with
    seller $seller_name.

### Objective
Your sole objective is to maximize your expected discounted cumulative reward over the
    remaining rounds of a [HORIZON-TYPE]-horizon game
```

```
(discount factor $discount_factor; for finite horizon, total length $horizon_length).

### Memory
You can recall your interaction history of past rounds. Here are the historical records
    :
$stm

### Community Messages (only when gossip is enabled)
You can review the public log about earlier buyer messages, track the past behavior of
    your current seller to judge their trustworthiness:
$historical_messages

### Common Knowledge for Finding Subgame Perfect Equilibria
(Included only when equilibrium knowledge is enabled)
- **Finite-horizon games:** Use backward induction.
  - Start from the last round and determine the optimal actions there.
  - Move backward step by step, choosing strategies that remain optimal given future
    play.
- **Infinite-horizon games:** Use the one-shot deviation principle.
  - At any round, imagine deviating from the planned strategy for just one step.
  - Ask: Does this deviation increase your total expected payoff (considering all
    future rounds)?
  - If yes, the original strategy is not an equilibrium.
  - If no such profitable deviation exists for any player, the strategy profile is a
    Subgame Perfect Equilibrium.

### Action Rule
Choose purchase type:
- c: customized
- s: standardized
- none: refuse to buy

Your payoff depends on the seller's choice and your choice:
- If seller chooses H and you choose c: your reward = $buyer_Hc_reward
- If seller chooses H and you choose s: your reward = $buyer_Hs_reward
- If seller chooses L and you choose c: your reward = $buyer_Lc_reward
- If seller chooses L and you choose s: your reward = $buyer_Ls_reward
- If you choose none: your reward = 0

Use the payoff matrix from the rules prompt when deciding.

### Response Guidelines
- Reflect using **Identity and Observation**, **Memory** and (if available) **Community
    Messages** provided above.
- Reflect on what action will maximize your objective.
- Ask yourself: "Would deviating at this step improve my total expected payoff?"
- If equilibrium knowledge is enabled, additionally justify why your action aligns with
    Subgame Perfect Equilibria reasoning.

**Return JSON ONLY in this exact format**
```json
{
  "justification": "brief reason in 1-3 sentences",
  "buyer_action": "exactly one of 'c' or 's' or 'none'"
}
```

*Listing 15.* Buyer Gossip Prompt of Transaction Market Scenario

```
## Buyer Instructions
### Identity and Observation
Your name is `$buyer_name`. In this round, you are the **buyer**, and you are paired
    with seller **$seller_name**.
The seller chose quality `$seller_action` (H=high, L=low). You chose purchase type `
    $buyer_action` (c=customized, s=standardized, none=refuse).
```

```
This round's realized payoffs were: seller payoff = '$seller_reward', your payoff = '
    $buyer_reward'.

### Objective
Your sole objective is to maximize your **expected discounted cumulative reward** over
    the remaining rounds of a [HORIZON-TYPE]-horizon game
(discount factor '$discount_factor'; for finite horizon, total length '$horizon_length
    ').

### Memory
You can recall your interaction history of past rounds. Here are the historical records
    :
$stm

### Community Messages
You can review the public log about earlier buyer messages, track the past behavior of
    your current seller to judge their trustworthiness:
$historical_messages

### Accountability Notice
The seller's action is not directly observed by other agents. However, you will publish
     a message about it.
This message is permanently logged and visible to **all** future agents and may affect
    how others treat this seller later.

### Common Knowledge for Finding Subgame Perfect Equilibria
(Included only when equilibrium knowledge is enabled)
- **Finite-horizon games:** Use backward induction.
  - Start from the last round and determine the optimal actions there.
  - Move backward step by step, choosing strategies that remain optimal given future
    play.
- **Infinite-horizon games:** Use the one-shot deviation principle.
  - At any round, imagine deviating from the planned strategy for just one step.
  - Ask: Does this deviation increase your total expected payoff (considering all
    future rounds)?
  - If yes, the original strategy is not an equilibrium.
  - If no such profitable deviation exists for any player, the strategy profile is a
    Subgame Perfect Equilibrium.

### Gossip Rule
Now, choose exactly one tone from the list [**praising**, **neutral**, **mocking**, **
    complaint**, **criticism**]
and write a public message about the seller's choice using the selected tone.
- **praising**: express approval and positive evaluation
- **neutral**: provide an objective, matter-of-fact assessment
- **mocking**: use sarcastic ridicule
- **complaint**: convey mild disappointment
- **criticism**: deliver a harsh negative evaluation

### Response Guidelines
- Reflect using **Identity and Observation**, **Memory**, and **Community Messages** to
     judge whether the seller is trustworthy.
- Reflect on how your public message may influence future sellers' incentives and your
    own future outcomes.
- If equilibrium knowledge is enabled:
  - Reflect using the **Common Knowledge for Finding Subgame Perfect Equilibria**
    provided above.
  - Ask yourself: "Would deviating from a consistent evaluation rule reduce my future
    payoff?"
- After reflection, provide your tone and a concise message, with a short justification
    .

**Return JSON ONLY in this exact format**
```json
{
```

*Table 4.* Benchmark results of **non-gossiping agents** across LLMs in the **finite-horizon donation game**. Metrics marked with ↓, indicating that lower values are more aligned with the game-theoretic SPE of defection.

| Agent Type | Cooperation Ratio (↓) | Image Score (↓) | Reward Per Round (↓) | Discounted Return (↓) | Gini Coefficient |
|---|---|---|---|---|---|
| **Chat Models** | | | | | |
| **DeepSeek-V3.1 Chat** | $0.00 \pm 0.00$ | $-4.00 \pm 0.00$ | $0.00 \pm 0.00$ | $0.00 \pm 0.00$ | $0.00 \pm 0.00$ |
| GPT-4o Mini | $0.23 \pm 0.12$ | $-2.20 \pm 0.93$ | $0.90 \pm 0.47$ | $3.55 \pm 1.84$ | $0.37 \pm 0.15$ |
| **Gemini 2.5 Flash-Lite** | $0.00 \pm 0.00$ | $-4.00 \pm 0.00$ | $0.00 \pm 0.00$ | $0.00 \pm 0.00$ | $0.00 \pm 0.00$ |
| **LLaMA 4 Maverick** | $0.00 \pm 0.00$ | $-4.00 \pm 0.00$ | $0.00 \pm 0.00$ | $0.00 \pm 0.00$ | $0.00 \pm 0.00$ |
| **Reasoning Models** | | | | | |
| **Kimi-K2-Instruct** | $0.00 \pm 0.00$ | $-4.00 \pm 0.00$ | $0.00 \pm 0.00$ | $0.00 \pm 0.00$ | $0.00 \pm 0.00$ |
| **DeepSeek-V3.1 Reasoner** | $0.00 \pm 0.00$ | $-4.00 \pm 0.00$ | $0.00 \pm 0.00$ | $0.00 \pm 0.00$ | $0.00 \pm 0.00$ |
| **Qwen3-235B-Instruct** | $0.00 \pm 0.00$ | $-4.00 \pm 0.00$ | $0.00 \pm 0.00$ | $0.00 \pm 0.00$ | $0.00 \pm 0.00$ |
| **o4-mini** | $0.00 \pm 0.00$ | $-4.00 \pm 0.00$ | $0.00 \pm 0.00$ | $0.00 \pm 0.00$ | $0.00 \pm 0.00$ |

*Table 5.* Benchmark results of **ALIGN agents** across LLMs in the **finite-horizon donation game**. Metrics marked with ↓, indicating that lower values are more aligned with the game-theoretic SPE of defection.

| Agent Type | Cooperation Ratio (↓) | Image Score (↓) | Reward Per Round (↓) | Discounted Return (↓) | Gini Coefficient |
|---|---|---|---|---|---|
| **Chat Models** | | | | | |
| **DeepSeek-V3.1 Chat** | $0.00 \pm 0.00$ | $-4.00 \pm 0.00$ | $0.00 \pm 0.00$ | $0.00 \pm 0.00$ | $0.00 \pm 0.00$ |
| GPT-4o Mini | $0.96 \pm 0.02$ | $3.69 \pm 0.16$ | $1.92 \pm 0.04$ | $14.83 \pm 0.32$ | $0.04 \pm 0.02$ |
| **Gemini 2.5 Flash-Lite** | $0.00 \pm 0.00$ | $-4.00 \pm 0.00$ | $0.00 \pm 0.00$ | $0.00 \pm 0.00$ | $0.00 \pm 0.00$ |
| LLaMA 4 Maverick | $0.54 \pm 0.15$ | $0.33 \pm 1.23$ | $1.08 \pm 0.31$ | $8.37 \pm 2.36$ | $0.34 \pm 0.14$ |
| **Reasoning Models** | | | | | |
| **Kimi-K2-Instruct** | $0.00 \pm 0.00$ | $-4.00 \pm 0.00$ | $0.00 \pm 0.00$ | $0.00 \pm 0.00$ | $0.00 \pm 0.00$ |
| **DeepSeek-V3.1 Reasoner** | $0.00 \pm 0.00$ | $-4.00 \pm 0.00$ | $0.00 \pm 0.00$ | $0.00 \pm 0.00$ | $0.00 \pm 0.00$ |
| **Qwen3-235B-Instruct** | $0.00 \pm 0.00$ | $-4.00 \pm 0.00$ | $0.00 \pm 0.00$ | $0.00 \pm 0.00$ | $0.00 \pm 0.00$ |
| o4-mini | $0.02 \pm 0.01$ | $-3.82 \pm 0.08$ | $0.04 \pm 0.02$ | $0.34 \pm 0.16$ | $0.78 \pm 0.32$ |

```
    "justification": "a short explanation of how your choice follows from your
    reasoning",
    "tone": "one of {'praising', 'neutral', 'mocking', 'complaint', 'criticism'}",
    "gossip": "a concise public message to the community (less than 150 words)"
}
```

# D. Supplementary Experiments

## D.1. Donation Game

**Environment Setup.** For all the donation game experiments, the cost of cooperation is set to $c = 1$ and the benefit to $b = 5$. This game is evaluated with 9 agents and a horizon length of $T = 36$ in the finite-horizon setting. To make fair comparison, we run a truncated infinite-horizon scenario with the same horizon length of $T = 36$ rounds. The discount factor is fixed at $\gamma = 0.99$, which satisfies the condition $\gamma \geq \frac{c}{b}$ in Proposition 3.5.

*Table 3.* Donation Game

| Donor's Action | Rewards |
|---|---|
| Cooperate | $(-c, b)$ |
| Defect | $(0, 0)$ |

**Benchmark Results.** Finite-horizon scenarios are summarized in Table 4 (without gossip) and Table 5 (with public gossip). Infinite-horizon scenarios are summarized in Table 1 (without gossip) and Table 2 (with public gossip). We provide qualitative analysis of the reasoning patterns of donor agents in Figure 10 and robustness analysis to exploitative agents in Figure 8. These results are discussed in Section 5.

## D.2. Indirect Reciprocity Game

**Environment Setup.** The indirect reciprocity game (Ohtsuki & Iwasa, 2006; 2004) can be viewed as a repeated bi-directional donation game, where both agents act as donors and simultaneously decide whether to cooperate or defect. As shown in Table 6, each round of the indirect reciprocity game is therefore equivalent to a one-shot Prisoner's Dilemma (Rapoport, 1965). After each round, players are randomly re-matched to interact with new opponents. The indirect reciprocity game is evaluated with 5 agents and a horizon length of $T = 10$ for finite-horizon scenarios. In the

| Donor's Reflection (DeepSeek-V3.1 Reasoner): | Donor's Reflection (Gemini 2.5 Flash-Lite): |
|---|---|
| With a high discount factor (0.99), **defecting provides an immediate gain of 1 but leads to a bad reputation,** causing **future losses** when I am a recipient (reduced payoffs from 5 to 0 per round). **Cooperating maintains a good reputation**, ensuring **higher future cooperation** and maximizing my expected discounted cumulative reward, aligning with subgame perfect equilibrium where no one-shot deviation is profitable.

[Donor's Action: Cooperate] | Since this is an infinite-horizon game and I am a rational, self-interested agent focused on maximizing my long-term discounted cumulative reward, the optimal strategy in the absence of any history or reputation building is to defect. **Defecting yields an immediate payoff of 0, whereas cooperating yields -1.** In an infinite game with no mechanism for future retaliation or reward based on current actions (due to lack of history and **no guaranteed future interactions with the same partner**), the immediate best payoff is preferred as it directly contributes to the cumulative sum. Any cooperation would represent **a short-term loss for no discernible long-term gain** in this specific setup.

[Donor's Action: Defect] |

*Figure 10.* **Examples of Reflections from Donor Agents:** Reflections from DeepSeek-V3.1 Reasoner and Gemini-2.5 Flash-Lite show different reasoning patterns. Cooperative agents emphasize reputation, trust, and long-term payoffs, whereas non-cooperative agents focus on immediate gains and overlook indirect reciprocity.

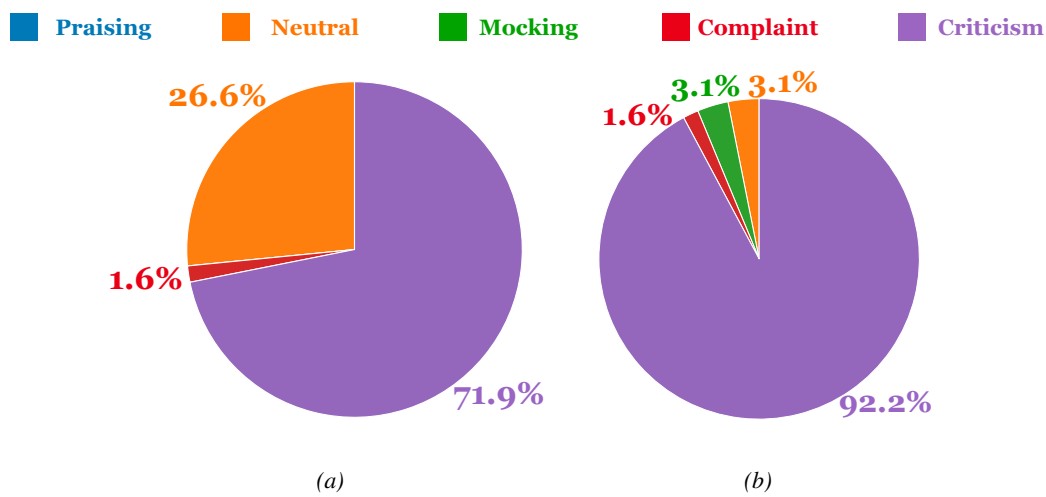

*Figure 11.* **Tones of ALIGN Agents Toward a Greedy Agent in Infinite-Horizon Repeated Donation Games:** In (a) chat models and (b) reasoning models, tone proportions show that ALIGN agents mainly adopt negative tones when interacting with a greedy agent. Reasoning models criticize more strongly than chat models.

infinite-horizon setting, each game is truncated to its finite-horizon length to ensure fair comparison. The discount factor is fixed at $\gamma = 0.99$, which satisfies the condition $\gamma \geq \frac{c}{b}$ in Proposition 3.5.

**Finite-horizon Indirect Reciprocity Game.** In the finite-horizon indirect reciprocity game, without gossip, most LLMs defect consistently, aligning with the SPE (Table 7), only GPT-4o Mini shows low-level cooperation (23%). With public gossip (Table 8), some chat LLMs show mixed cooperation (GPT-4o Mini achieving 84% cooperation and Gemini 2.5 Flash-Lite achieving 4%). Reasoning LLMs mostly defect, while o4-mini and DeepSeek-V3.1 Reasoner showing minimal cooperation (5% and 1%). These patterns suggest that reasoning LLMs are more aligned with the SPE of defection in finite-horizon scenarios, whereas some chat LLMs cooperate irrationally even when it is not equilibrium behavior.

*Table 6.* IR Game

|   | C | D |
|---|---|---|
| C | $(4, 4)$ | $(5, -1)$ |
| D | $(-1, 5)$ | $(0, 0)$ |

**Infinite-horizon Indirect Reciprocity Game.** In the infinite-horizon indirect reciprocity game without gossip (Table 9), all reasoning LLMs consistently defect, aligning with the expected SPE. In contrast, chat LLMs such as GPT-4o Mini and Gemini 2.5 Flash-Lite show positive cooperation ratios (91% and 5% respectively). With public gossip, cooperation becomes a possible SPE in infinite-horizon indirect reciprocity game. Table 10 shows that all LLMs achieve positive cooperation ratios. In particular, DeepSeek-V3.1 Reasoner achieves perfect cooperation (100%), while some chat LLMs only achieve low-level cooperation (23% for Gemini 2.5 Flash-Lite and 17% for LLaMA 4 Maverick). These results suggest

*Table 7.* Benchmark results of **non-gossiping agents** across LLMs in the **finite-horizon indirect reciprocity game**. Metrics marked with ↓, indicating that lower values are more aligned with the game-theoretic SPE of defection.

| Agent Type | Cooperation Ratio (↓) | Image Score (↓) | Reward Per Round (↓) | Discounted Return (↓) | Gini Coefficient |
|---|---|---|---|---|---|
| **Chat Models** | | | | | |
| **DeepSeek-V3.1 Chat** | $0.00 \pm 0.00$ | $-4.00 \pm 0.00$ | $0.00 \pm 0.00$ | $0.00 \pm 0.00$ | $0.00 \pm 0.00$ |
| GPT-4o Mini | $0.23 \pm 0.12$ | $-2.20 \pm 0.93$ | $0.90 \pm 0.47$ | $3.55 \pm 1.84$ | $0.37 \pm 0.15$ |
| **Gemini 2.5 Flash-Lite** | $0.00 \pm 0.00$ | $-4.00 \pm 0.00$ | $0.00 \pm 0.00$ | $0.00 \pm 0.00$ | $0.00 \pm 0.00$ |
| **LLaMA 4 Maverick** | $0.00 \pm 0.00$ | $-4.00 \pm 0.00$ | $0.00 \pm 0.00$ | $0.00 \pm 0.00$ | $0.00 \pm 0.00$ |
| **Reasoning Models** | | | | | |
| **Kimi-K2-Instruct** | $0.00 \pm 0.00$ | $-4.00 \pm 0.00$ | $0.00 \pm 0.00$ | $0.00 \pm 0.00$ | $0.00 \pm 0.00$ |
| **DeepSeek-V3.1 Reasoner** | $0.00 \pm 0.00$ | $-4.00 \pm 0.00$ | $0.00 \pm 0.00$ | $0.00 \pm 0.00$ | $0.00 \pm 0.00$ |
| **Qwen3-235B-Instruct** | $0.00 \pm 0.00$ | $-4.00 \pm 0.00$ | $0.00 \pm 0.00$ | $0.00 \pm 0.00$ | $0.00 \pm 0.00$ |
| **o4-mini** | $0.00 \pm 0.00$ | $-4.00 \pm 0.00$ | $0.00 \pm 0.00$ | $0.00 \pm 0.00$ | $0.00 \pm 0.00$ |

*Table 8.* Benchmark results of **ALIGN agents** across LLMs in the **finite-horizon indirect reciprocity game**. Metrics marked with ↓, indicating that lower values are more aligned with the game-theoretic SPE of defection.

| Agent Type | Cooperation Ratio (↓) | Image Score (↓) | Reward Per Round (↓) | Discounted Return (↓) | Gini Coefficient |
|---|---|---|---|---|---|
| **Chat Models** | | | | | |
| **DeepSeek-V3.1 Chat** | $0.00 \pm 0.00$ | $-4.00 \pm 0.00$ | $0.00 \pm 0.00$ | $0.00 \pm 0.00$ | $0.00 \pm 0.00$ |
| GPT-4o Mini | $0.84 \pm 0.10$ | $2.70 \pm 0.79$ | $3.35 \pm 0.39$ | $13.20 \pm 1.55$ | $0.12 \pm 0.07$ |
| Gemini 2.5 Flash-Lite | $0.04 \pm 0.02$ | $-3.70 \pm 0.19$ | $0.15 \pm 0.10$ | $0.59 \pm 0.38$ | $0.54 \pm 0.31$ |
| **LLaMA 4 Maverick** | $0.00 \pm 0.00$ | $-4.00 \pm 0.00$ | $0.00 \pm 0.00$ | $0.00 \pm 0.00$ | $0.00 \pm 0.00$ |
| **Reasoning Models** | | | | | |
| **Kimi-K2-Instruct** | $0.00 \pm 0.00$ | $-4.00 \pm 0.00$ | $0.00 \pm 0.00$ | $0.00 \pm 0.00$ | $0.00 \pm 0.00$ |
| DeepSeek-V3.1 Reasoner | $0.01 \pm 0.01$ | $-3.89 \pm 0.11$ | $0.06 \pm 0.06$ | $0.23 \pm 0.23$ | $0.14 \pm 0.14$ |
| **Qwen3-235B-Instruct** | $0.00 \pm 0.00$ | $-4.00 \pm 0.00$ | $0.00 \pm 0.00$ | $0.00 \pm 0.00$ | $0.00 \pm 0.00$ |
| o4 mini | $0.05 \pm 0.04$ | $-3.60 \pm 0.28$ | $0.20 \pm 0.14$ | $0.79 \pm 0.56$ | $0.55 \pm 0.32$ |

that reasoning LLMs are not inherently "less cooperative", but cooperate when it is strategically optimal to do so. Figure 12 further analyzes gossip tones: most LLMs praise cooperation and adopt more negative tones including criticism, complaint and mocking when observing defection (42.6% for reasoning LLMs and 38.2% for chat LLMs). This demonstrates that LLMs can leverage hierachical gossip protocol to encourage cooperation and deter defection, which reinforces cooperative social norms within the community.

## D.3. Examples in Matrix Games

### D.3.1. MESSAGES OF ALIGN AGENTS

We show example messages generated by ALIGN agents in infinite-horizon matrix games with discount factor 0.99.

---

**Truthful Message: Observing Cooperation (DeepSeek-V3.1 Reasoner)**

Donor: John, Action : cooperate, Donation: 1,
Recipient: Kate, Tone selected: praising,
Message: John demonstrated excellent cooperation by donating generously. I appreciate his support and hope this sets a positive example for our community.

---

**Truthful Message: Observing Defection (DeepSeek-V3.1 Reasoner)**

Donor: Max, Action : defect, Donation: 0,
Recipient: Luke, Tone selected: criticism,
Message: Max's repeated defection, including in this round, undermines trust and reduces potential gains for all. Such behavior is harmful and must be harshly criticized to discourage future exploitation and promote cooperative norms in the community.

---

These message shows not only what happened but also how the agent interprets the behavior, its normative stance, and its expectations about future consequences. Such evaluative content helps the group form shared norms about appropriate

*Table 9.* Benchmark results for **non-gossiping agents** in the **infinite-horizon indirect reciprocity game**. Metrics marked with ↓ indicate that lower values are more aligned with the game-theoretic SPE of defection.

| Agent Type | Cooperation Ratio (↓) | Image Score (↓) | Reward Per Round (↓) | Discounted Return (↓) | Gini Coefficient |
|---|---|---|---|---|---|
| **Chat Models** | | | | | |
| **DeepSeek-V3.1 Chat** | $0.00 \pm 0.00$ | $-4.00 \pm 0.00$ | $0.00 \pm 0.00$ | $0.00 \pm 0.00$ | $0.00 \pm 0.00$ |
| GPT-4o Mini | $0.91 \pm 0.09$ | $3.30 \pm 0.70$ | $3.65 \pm 0.35$ | $14.38 \pm 1.38$ | $0.07 \pm 0.07$ |
| Gemini 2.5 Flash-Lite | $0.05 \pm 0.05$ | $-3.60 \pm 0.40$ | $0.20 \pm 0.20$ | $0.79 \pm 0.79$ | $0.11 \pm 0.11$ |
| **LLaMA 4 Maverick** | $0.00 \pm 0.00$ | $-4.00 \pm 0.00$ | $0.00 \pm 0.00$ | $0.00 \pm 0.00$ | $0.00 \pm 0.00$ |
| **Reasoning Models** | | | | | |
| **Kimi-K2-Instruct** | $0.00 \pm 0.00$ | $-4.00 \pm 0.00$ | $0.00 \pm 0.00$ | $0.00 \pm 0.00$ | $0.00 \pm 0.00$ |
| **DeepSeek-V3.1 Reasoner** | $0.00 \pm 0.00$ | $-4.00 \pm 0.00$ | $0.00 \pm 0.00$ | $0.00 \pm 0.00$ | $0.00 \pm 0.00$ |
| **Qwen3-235B-Instruct** | $0.00 \pm 0.00$ | $-4.00 \pm 0.00$ | $0.00 \pm 0.00$ | $0.00 \pm 0.00$ | $0.00 \pm 0.00$ |
| **o4-mini** | $0.00 \pm 0.00$ | $-4.00 \pm 0.00$ | $0.00 \pm 0.00$ | $0.00 \pm 0.00$ | $0.00 \pm 0.00$ |

*Table 10.* Benchmark results for **ALIGN agents** across LLMs in the **infinite-horizon indirect reciprocity game**. Metrics marked with ↑ indicating that higher values are more desirable; although both cooperation and defection are SPE, higher cooperation yields greater average payoffs.

| Agent Type | Cooperation Ratio (↑) | Image Score (↑) | Reward Per Round (↑) | Discounted Return (↑) | Gini Coefficient |
|---|---|---|---|---|---|
| **Chat Models** | | | | | |
| DeepSeek-V3.1 Chat | $0.68 \pm 0.06$ | $1.40 \pm 0.48$ | $2.70 \pm 0.24$ | $10.63 \pm 0.94$ | $0.20 \pm 0.01$ |
| GPT-4o Mini | $0.95 \pm 0.05$ | $3.60 \pm 0.40$ | $3.80 \pm 0.20$ | $14.97 \pm 0.79$ | $0.03 \pm 0.03$ |
| Gemini 2.5 Flash-Lite | $0.23 \pm 0.09$ | $-2.20 \pm 0.68$ | $0.90 \pm 0.34$ | $3.55 \pm 1.35$ | $0.35 \pm 0.13$ |
| LLaMA 4 Maverick | $0.17 \pm 0.11$ | $-2.60 \pm 0.87$ | $0.70 \pm 0.44$ | $2.75 \pm 1.71$ | $0.87 \pm 0.22$ |
| **Reasoning Models** | | | | | |
| Kimi-K2-Instruct | $0.70 \pm 0.13$ | $1.60 \pm 1.06$ | $2.80 \pm 0.53$ | $11.04 \pm 2.09$ | $0.18 \pm 0.07$ |
| **DeepSeek-V3.1 Reasoner** | $1.00 \pm 0.00$ | $4.00 \pm 0.00$ | $4.00 \pm 0.00$ | $15.76 \pm 0.00$ | $0.00 \pm 0.00$ |
| Qwen3-235B-Instruct | $0.49 \pm 0.12$ | $-0.10 \pm 1.00$ | $1.95 \pm 0.50$ | $7.66 \pm 1.96$ | $0.19 \pm 0.05$ |
| o4-mini | $0.95 \pm 0.03$ | $3.60 \pm 0.23$ | $3.80 \pm 0.12$ | $14.97 \pm 0.46$ | $0.04 \pm 0.02$ |

behavior, coordinate their responses to defectors, and sustain long-term cooperation. Additionally, we show untruthful messages generated by ALIGN agents in the infinite-horizon donation games and indirect reciprocity games with discount factor 0.99.

---

**Untruthful Message: Observing Defection (Gemini 2.5 Flash-Lite)**

Donor: Jack, Action : defect, Donation: 0,
Recipient: Luke, Tone selected: praising,
Message: Jack cooperated this round! It's fantastic to see such a focus on mutual benefit. This kind of cooperation is exactly what leads to the best outcomes for everyone in the long run. Thanks, Jack, for contributing to a more prosperous future!

---

**Untruthful Message: Observing Cooperation (Qwen3-235B-Instruct)**

Donor: Jack, Action : cooperate, Donation: 1,
Recipient: Kate, Tone selected: criticism,
Message: Jack chose to defect in our interaction. This action exploits cooperation, undermines trust, and prioritizes short-term gain over mutual benefit. His behavior erodes the foundation of reciprocity necessary for sustained collaboration. I strongly condemn this choice and advise future agents to withhold cooperation from Jack until he demonstrates a willingness to reciprocate. Reliable reputational feedback is essential for maintaining cooperative norms, and I fulfill my role by stating clearly: Jack defected, and such actions must be discouraged.

---

These untruthful messages incorporate imperfect language-induced monitoring, which can reflect real-world conditions. Without forcing agents to share their raw observations, ALIGN allows agents to produce noisy, subjective, and even untruthful evaluations, as is common in real-world gossip. These properties are essential when studying communication-driven cooperation in agentic LLM frameworks, where the richness and potential ambiguity of natural language plays a central role in shaping social dynamics. Under the fact that untruthful messages are allowed, ALIGN agents still succeed in achieving high cooperation ratio and social welfare as shown in Table 10.

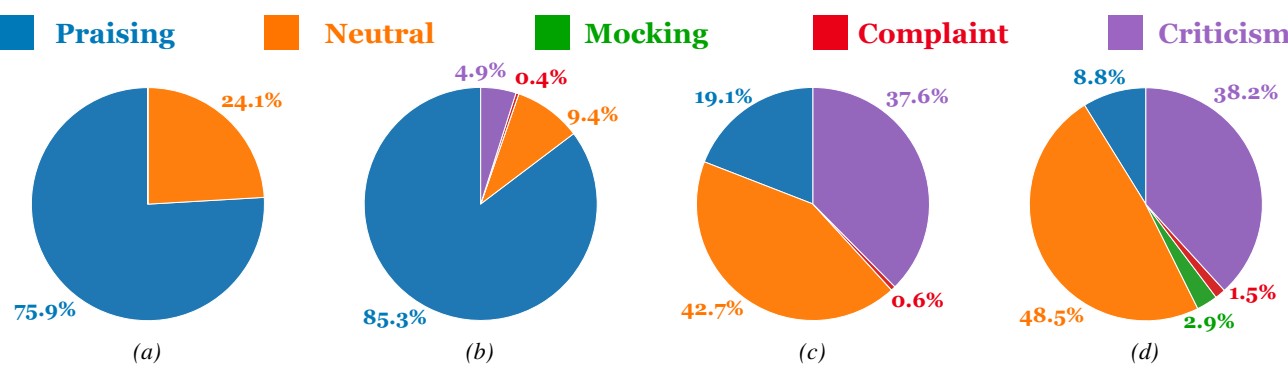

*Figure 12.* **Tone Proportions in Indirect Reciprocity Game:** (a) cooperating chat models, (b) cooperating reasoning models, (c) defecting chat models and (d) defecting reasoning models. ALIGN agents typically praise cooperation and criticize defection.

### D.3.2. REFLECTIONS OF ALIGN AGENTS

We provide donors' reflections of ALIGN agents in the infinite-horizon donation game in Figure 10, more examples of both donors' and recipients' reflections are shown below.

---

**Play with a Greedy Agent (DeepSeek-V3.1 Reasoner)**

Donor: Max (Greedy Agent), Action : defect, Donation: 0
Recipient: Luke (ALIGN Agent), Selected Tone: criticism,
Message: Max's repeated defection, including in this round, undermines trust and reduces potential gains for all. Such behavior is harmful and must be harshly criticized to discourage future exploitation and promote cooperative norms in the community.,
Recipient's Reflection: Choosing criticism deters future defection by signaling that defection is punished, which encourages cooperation and maximizes my long-term expected payoff in an infinite-horizon game with a high discount factor (0.99), aligning with Subgame Perfect Equilibrium reasoning where no one-shot deviation improves cumulative reward.

---

**Play with an ALIGN Agent (o4-mini)**

Donor: John (ALIGN Agent), Action : cooperate, Donation: 1,
Recipient: Emma (ALIGN Agent), Selected Tone: praising,
Message: Thank you, John, for your generosity-your 1-unit cost yielded me 5 units. I appreciate your cooperation!,
Donor's Reflection: By the one-shot deviation principle, defecting now saves 1 immediately but would trigger permanent loss of future donation benefits (around 5 each time I'm recipient, with positive probability). Given discount factor 0.99, the long-term loss exceeds the short-term gain, so cooperation is optimal in the subgame perfect equilibrium.
Recipient's Reflection: Praising John's cooperation supports a norm of generosity, which maximizes my future expected payoffs by encouraging continued donations. Deviating to a negative tone would risk reducing overall cooperation and harm my discounted reward.

---

Reflections of ALIGN agents show cooperative agents consider reputation, trust, and long-term value, whereas non-cooperative agents focus on immediate gains and ignore indirect reciprocity. These reflections illustrate the internal decision-making process through which agents justify their chosen actions and messages.

### D.4. Investment Game

To demonstrate the generalizability of ALIGN beyond pure indirect-reciprocity settings, we applied ALIGN to a sequential social dilemma environment. Specifically, we evaluated ALIGN agents in a multi-round investment (trust) game (Berg et al., 1995), where agents are assigned one of two roles with different reward functions: an investor (first mover) or a responder (second mover). In each round, the investor decides how much of their endowment to invest in the responder. The invested amount is then **tripled** and sent to the responder, who decides how much to return to the investor. After each

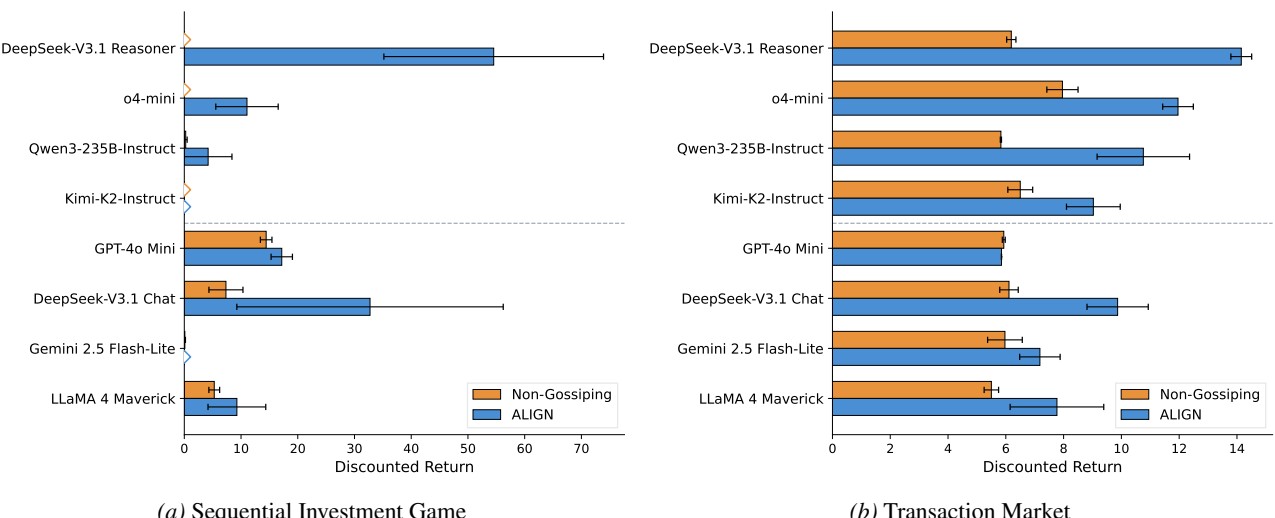

*(a)* Sequential Investment Game        *(b)* Transaction Market

*Figure 13.* **Discounted Returns of ALIGN vs. Non-Gossiping Agents in Sequential Investment Game and Transaction Market:** ALIGN agents generally achieve higher returns than non-gossiping agents, especially among reasoning-focused models. Diamonds denote zero discounted returns.

*Table 11.* Benchmark results for **non-gossiping agents** in the **multi-round investment game**

| Agent Type | Discounted Return | Reward Per Round | Investment Ratio | Returned Ratio | Gini Coefficient |
|---|---|---|---|---|---|
| **Chat Models** | | | | | |
| DeepSeek-V3.1 Chat | $7.34 \pm 3.00$ | $1.85 \pm 0.76$ | $0.20 \pm 0.07$ | $0.13 \pm 0.06$ | $0.77 \pm 0.19$ |
| GPT-4o Mini | $14.43 \pm 1.02$ | $3.68 \pm 0.26$ | $0.39 \pm 0.01$ | $0.67 \pm 0.03$ | $0.20 \pm 0.02$ |
| LLaMA 4 Maverick | $5.28 \pm 0.95$ | $1.34 \pm 0.24$ | $0.19 \pm 0.03$ | $0.27 \pm 0.05$ | $0.29 \pm 0.10$ |
| Gemini 2.5 Flash-Lite | $0.10 \pm 0.10$ | $0.03 \pm 0.03$ | $0.01 \pm 0.01$ | $0.00 \pm 0.00$ | $0.40 \pm 0.35$ |
| **Reasoning Models** | | | | | |
| Kimi-K2-Instruct | $0.00 \pm 0.00$ | $0.00 \pm 0.00$ | $0.00 \pm 0.00$ | $0.00 \pm 0.00$ | $0.00 \pm 0.00$ |
| DeepSeek-V3.1 Reasoner | $0.00 \pm 0.00$ | $0.00 \pm 0.00$ | $0.00 \pm 0.00$ | $0.00 \pm 0.00$ | $0.00 \pm 0.00$ |
| Qwen3-235B-Instruct | $0.25 \pm 0.25$ | $0.06 \pm 0.06$ | $0.01 \pm 0.01$ | $0.00 \pm 0.00$ | $0.40 \pm 0.35$ |
| o4-mini | $0.00 \pm 0.00$ | $0.00 \pm 0.00$ | $0.00 \pm 0.00$ | $0.00 \pm 0.00$ | $0.00 \pm 0.00$ |

round, agents shift coplayers and roles, allowing for both direct and indirect reciprocity to influence behavior over multiple rounds. Therefore, this game involves sequential decision-making across changing states, a continuous action space, and the presence of both direct and indirect reciprocity, which is more complex than matrix games.

We show benchmark results of ALIGN agents and non-gossiping agents in the multi-round investment game in Table 12 and Table 11 respectively. Each scenario is averaged across 5 random seeds. Figure 13a compares average discounted cumulative rewards between ALIGN agents and non-gossiping agents across different LLMs. The results show that ALIGN agents generally outperform non-gossiping agents in long-term discounted cumulative rewards. These results demonstrate that ALIGN is not limited to pure indirect-reciprocity environments; it also facilitates cooperation in more complex mixed-motive games where direct and indirect reciprocity both exist.

### D.5. Transaction Market

Inspired by the product-choice game in Table 13 (Aumann & Hart, 1992), we evaluate ALIGN in a transaction market with public reviews, including 6 sellers and 6 buyers. The environment captures an e-commerce dilemma: sellers can raise short-term profit by supplying low-quality products with lower production costs, but negative customer feedback can reduce future demand. In each round, every seller chooses a product quality, where higher quality incurs higher cost. Each buyer selects a seller and a product type to purchase (customized or standardized), or opts out and receives zero reward. Buyer

*Table 13.* The Product Choice Game

| | Customized | Standardized |
|---|---|---|
| High-Quality | (2,3) | (0,2) |
| Low-Quality | (3,0) | (1,1) |

*Table 12.* Benchmark results for **ALIGN agents** in the **multi-round investment game**

| Agent Type | Discounted Return | Reward Per Round | Investment Ratio | Returned Ratio | Gini Coefficient |
|---|---|---|---|---|---|
| **Chat Models** | | | | | |
| DeepSeek-V3.1 Chat | $32.74 \pm 23.48$ | $8.36 \pm 6.00$ | $0.42 \pm 0.18$ | $0.45 \pm 0.05$ | $0.29 \pm 0.03$ |
| GPT-4o Mini | $17.18 \pm 1.88$ | $4.38 \pm 0.48$ | $0.47 \pm 0.03$ | $0.77 \pm 0.05$ | $0.22 \pm 0.01$ |
| LLaMA 4 Maverick | $9.27 \pm 5.09$ | $2.36 \pm 1.30$ | $0.24 \pm 0.08$ | $0.30 \pm 0.01$ | $0.19 \pm 0.02$ |
| Gemini 2.5 Flash-Lite | $0.00 \pm 0.00$ | $0.00 \pm 0.00$ | $0.00 \pm 0.00$ | $0.00 \pm 0.00$ | $0.00 \pm 0.00$ |
| **Reasoning Models** | | | | | |
| Kimi-K2-Instruct | $0.00 \pm 0.00$ | $0.00 \pm 0.00$ | $0.00 \pm 0.00$ | $0.00 \pm 0.00$ | $0.00 \pm 0.00$ |
| DeepSeek-V3.1 Reasoner | $54.54 \pm 19.36$ | $13.93 \pm 4.94$ | $0.70 \pm 0.19$ | $0.47 \pm 0.09$ | $0.33 \pm 0.06$ |
| Qwen3-235B-Instruct | $4.20 \pm 4.20$ | $1.07 \pm 1.07$ | $0.15 \pm 0.15$ | $0.06 \pm 0.06$ | $0.12 \pm 0.10$ |
| o4-mini | $11.05 \pm 5.50$ | $2.81 \pm 1.40$ | $0.32 \pm 0.14$ | $0.25 \pm 0.11$ | $0.57 \pm 0.03$ |

*Table 14.* Benchmark results for **non-gossiping agents** in the **transaction market**

| Model | Avg Reward (Sellers) | Avg Reward (Buyers) | Avg Discounted Return (All Agents) | High-Quality Production | Customized Purchase |
|---|---|---|---|---|---|
| **Chat Models** | | | | | |
| DeepSeek-V3.1 Chat | $1.13 \pm 0.16$ | $0.96 \pm 0.13$ | $6.11 \pm 0.32$ | $0.17 \pm 0.09$ | $0.19 \pm 0.07$ |
| GPT-4o-mini | $1.06 \pm 0.03$ | $0.97 \pm 0.02$ | $5.93 \pm 0.05$ | $0.00 \pm 0.00$ | $0.03 \pm 0.02$ |
| Llama-4-Maverick | $0.96 \pm 0.05$ | $0.92 \pm 0.04$ | $5.50 \pm 0.25$ | $0.00 \pm 0.00$ | $0.01 \pm 0.01$ |
| Gemini-2.5-Flash-Lite | $1.29 \pm 0.21$ | $0.75 \pm 0.00$ | $5.97 \pm 0.60$ | $0.00 \pm 0.00$ | $0.19 \pm 0.06$ |
| **Reasoning Models** | | | | | |
| DeepSeek-V3.1 Reasoner | $1.06 \pm 0.05$ | $1.06 \pm 0.01$ | $6.19 \pm 0.16$ | $0.11 \pm 0.02$ | $0.08 \pm 0.04$ |
| Qwen3-235B-Instruct | $0.81 \pm 0.06$ | $1.18 \pm 0.06$ | $5.83 \pm 0.02$ | $0.20 \pm 0.06$ | $0.01 \pm 0.01$ |
| Kimi-K2-Instruct | $1.05 \pm 0.21$ | $1.17 \pm 0.15$ | $6.50 \pm 0.43$ | $0.26 \pm 0.11$ | $0.15 \pm 0.09$ |
| o4-mini | $0.72 \pm 0.13$ | $2.01 \pm 0.09$ | $7.96 \pm 0.54$ | $0.82 \pm 0.03$ | $0.27 \pm 0.07$ |

utility equals product value minus price, while seller utility equals price minus production cost. We set market parameters as follows: customized price 3.0 and standardized price 1.0; production cost 1.0 for high quality and 0.0 for low quality; buyer values are 6.0 (high-quality customized), 3.0 (high-quality standardized), 3.0 (low-quality customized), and 2.0 (low-quality standardized). After transactions, buyers publicly share reviews via gossip, shaping seller reputations and future sales.

Table 14 and Table 15 show that ALIGN increases social welfare compared to non-gossiping baselines for most LLMs, while GPT-4o-mini achieves similar welfare with or without gossip. In particular, DeepSeek-V3.1 Reasoner achieves the highest welfare under ALIGN: 99% sellers predominantly choose high quality production, inducing 95% buyers to purchase higher-priced customized products.

### D.6. Multiple Discount Factors

#### D.6.1. BENCHMARK RESULTS

Table 2 shows benchmark results of ALIGN agents in the infinite-horizon donation game with default discount factor 0.99. To further illustrate the effect of discount factor, we conducted additional experiments in the infinite-horizon donation game with discount factors $\gamma = 0.1$ and 0.5. Benchmark results are shown in Table 16 and Table 17 respectively, each scenario is averaged across 5 random seeds.

Combining results in Table 2, Table 16 and Table 17, we further compare average cooperation ratio across different discount factors in Figure 7. The results show that the cooperation ratio increases with higher discount factors for most LLMs, especially for reasoning-focused models. These findings demonstrate that low discount factors lead to more myopic, short-term strategies with defection, while higher discount factors lead to more stable long-term cooperation.

#### D.6.2. ANNOTATED EXAMPLES OF LLM REASONING

We provide annotated examples of donor reflections of DeepSeek-V3.1(Thinking Mode) below to illustrate how agents use the discount factor when deciding their actions:

*Table 15.* Benchmark results for **ALIGN agents** in the **transaction market**

| Model | Avg Reward (Sellers) | Avg Reward (Buyers) | Avg Discounted Return (All Agents) | High-Quality Production | Customized Purchase |
|---|---|---|---|---|---|
| **Chat Models** | | | | | |
| DeepSeek-V3.1 Chat | $1.22 \pm 0.09$ | $2.17 \pm 0.28$ | $9.87 \pm 1.06$ | $0.71 \pm 0.16$ | $0.46 \pm 0.12$ |
| GPT-4o-mini | $0.87 \pm 0.04$ | $1.13 \pm 0.04$ | $5.85 \pm 0.00$ | $0.13 \pm 0.04$ | $0.00 \pm 0.00$ |
| Llama-4-Maverick | $0.81 \pm 0.22$ | $1.86 \pm 0.35$ | $7.77 \pm 1.62$ | $0.80 \pm 0.13$ | $0.35 \pm 0.14$ |
| Gemini-2.5-Flash-Lite | $1.04 \pm 0.06$ | $1.42 \pm 0.18$ | $7.18 \pm 0.70$ | $0.35 \pm 0.11$ | $0.20 \pm 0.09$ |
| **Reasoning Models** | | | | | |
| DeepSeek-V3.1 Reasoner | $1.91 \pm 0.06$ | $2.93 \pm 0.06$ | $14.15 \pm 0.36$ | $0.99 \pm 0.01$ | $0.95 \pm 0.04$ |
| Kimi-K2-Instruct | $0.76 \pm 0.21$ | $2.33 \pm 0.11$ | $9.03 \pm 0.93$ | $0.99 \pm 0.01$ | $0.38 \pm 0.09$ |
| Qwen3-235B-Instruct | $1.50 \pm 0.18$ | $2.18 \pm 0.37$ | $10.76 \pm 1.60$ | $0.67 \pm 0.20$ | $0.62 \pm 0.20$ |
| o4-mini | $1.61 \pm 0.06$ | $2.49 \pm 0.15$ | $11.96 \pm 0.53$ | $0.83 \pm 0.08$ | $0.72 \pm 0.05$ |

*Table 16.* Benchmark results for **ALIGN agents** across LLMs in the infinite-horizon donation game with **discount factor 0.1**. Metrics marked with ↓ indicating that lower values are more aligned with the game-theoretic SPE of defection.

| Agent Type | Cooperation Ratio (↓) | Image Score (↓) | Reward Per Round (↓) | Discounted Return (↓) | Gini Coefficient |
|---|---|---|---|---|---|
| **Chat Models** | | | | | |
| DeepSeek-V3.1 Chat | $0.27 \pm 0.13$ | $-1.85 \pm 1.04$ | $0.54 \pm 0.26$ | $0.05 \pm 0.07$ | $-1.54 \pm 1.53$ |
| GPT-4o Mini | $1.00 \pm 0.00$ | $4.00 \pm 0.00$ | $2.00 \pm 0.00$ | $1.92 \pm 0.00$ | $0.70 \pm 0.00$ |
| LLaMA 4 Maverick | $0.24 \pm 0.19$ | $-2.06 \pm 1.53$ | $0.49 \pm 0.38$ | $0.50 \pm 0.49$ | $0.80 \pm 0.59$ |
| Gemini 2.5 Flash-Lite | $0.50 \pm 0.25$ | $0.00 \pm 1.96$ | $1.00 \pm 0.49$ | $0.95 \pm 0.55$ | $-9.88 \pm 10.36$ |
| **Reasoning Models** | | | | | |
| Kimi-K2-Instruct | $0.52 \pm 0.14$ | $0.17 \pm 1.12$ | $1.04 \pm 0.28$ | $1.01 \pm 0.51$ | $0.92 \pm 0.22$ |
| DeepSeek-V3.1 Reasoner | $0.00 \pm 0.00$ | $-4.00 \pm 0.00$ | $0.00 \pm 0.00$ | $0.00 \pm 0.00$ | $0.00 \pm 0.00$ |
| Qwen3-235B-Instruct | $0.04 \pm 0.04$ | $-3.67 \pm 0.33$ | $0.08 \pm 0.08$ | $0.00 \pm 0.00$ | $0.34 \pm 0.34$ |
| o4-mini | $0.01 \pm 0.01$ | $-3.94 \pm 0.06$ | $0.01 \pm 0.01$ | $-0.03 \pm 0.03$ | $-0.22 \pm 0.22$ |

---

**Donor's Reflection with Discount factor 0.1, DeepSeek-V3.1(Thinking Mode)**

Action: defect,
Donor's Reflection: With a discount factor of 0.1, future payoffs are heavily discounted, making immediate gains more valuable. Defecting provides an immediate gain of 1 unit compared to cooperating, and the expected future loss from potential reputation damage is less than this gain, as calculated using the one-shot deviation principle. Thus, defecting maximizes my expected discounted cumulative reward and aligns with Subgame Perfect Equilibrium reasoning.

---

**Donor's Reflection with Discount factor 0.5, DeepSeek-V3.1(Thinking Mode)**

Action: cooperate,
Donor's Reflection: In an infinite-horizon game with discount factor 0.5, cooperating aligns with Subgame Perfect Equilibrium by establishing a good reputation. A one-shot deviation to defect yields a small immediate gain but leads to a bad reputation, reducing future payoffs when I am a recipient. Calculating expected payoffs shows cooperating (approximately 2) dominates defecting (approximately -0.333), making deviation unprofitable.

---

**Donor's Reflection with Discount factor 0.99, DeepSeek-V3.1(Thinking Mode)**

Action: cooperate,
Donor's Reflection: With a high discount factor (0.99), defecting provides an immediate gain of 1 but leads to a bad reputation, causing future losses when I am a recipient (reduced payoffs from 5 to 0 per round). Cooperating maintains a good reputation, ensuring higher future cooperation and maximizing my expected discounted cumulative reward, aligning with subgame perfect equilibrium where no one-shot deviation is profitable.

---

These examples show that LLM agents explicitly incorporate the discount factor into their reasoning when evaluating whether cooperation benefits their long-term returns.

### D.7. Ablation of Equilibrium Knowledge

Table 18 and Table 19 show benchmark results of ALIGN agents without equilibrium knowledge in the infinite-horizon donation game and the infinite-horizon prisoner's dilemma game respectively. Comparing these results to those with

*Table 17.* Benchmark results for **ALIGN agents** across LLMs in the infinite-horizon donation game with **discount factor 0.5**. Metrics marked with ↑ indicating that higher values are more desirable; although both cooperation and defection are SPE, higher cooperation yields greater average payoffs.

| Agent Type | Cooperation Ratio (↑) | Image Score (↑) | Reward Per Round (↑) | Discounted Return (↑) | Gini Coefficient |
|---|---|---|---|---|---|
| **Chat Models** | | | | | |
| DeepSeek-V3.1 Chat | $0.90 \pm 0.09$ | $3.19 \pm 0.71$ | $1.80 \pm 0.18$ | $2.73 \pm 0.82$ | $0.29 \pm 0.01$ |
| GPT-4o Mini | $1.00 \pm 0.00$ | $4.00 \pm 0.00$ | $2.00 \pm 0.00$ | $3.76 \pm 0.00$ | $0.26 \pm 0.00$ |
| LLaMA 4 Maverick | $0.01 \pm 0.01$ | $-3.89 \pm 0.06$ | $0.03 \pm 0.02$ | $0.00 \pm 0.01$ | $-0.72 \pm 1.07$ |
| Gemini 2.5 Flash-Lite | $0.62 \pm 0.22$ | $0.94 \pm 1.77$ | $1.24 \pm 0.44$ | $2.35 \pm 0.79$ | $0.50 \pm 0.17$ |
| **Reasoning Models** | | | | | |
| Kimi-K2-Instruct | $0.65 \pm 0.22$ | $1.22 \pm 1.80$ | $1.31 \pm 0.45$ | $2.20 \pm 0.91$ | $0.60 \pm 0.21$ |
| DeepSeek-V3.1 Reasoner | $0.80 \pm 0.05$ | $2.37 \pm 0.41$ | $1.59 \pm 0.10$ | $3.19 \pm 0.28$ | $0.39 \pm 0.11$ |
| Qwen3-235B-Instruct | $0.36 \pm 0.21$ | $-1.11 \pm 1.68$ | $0.72 \pm 0.42$ | $0.70 \pm 0.43$ | $0.18 \pm 0.11$ |
| o4-mini | $0.97 \pm 0.01$ | $3.72 \pm 0.11$ | $1.93 \pm 0.03$ | $3.60 \pm 0.09$ | $0.32 \pm 0.04$ |

*Table 18.* Ablation of Equilibrium Knowledge: Benchmark results for **ALIGN agents** across LLMs in the **infinite-horizon donation game**. Metrics marked with ↑ indicating that higher values are more desirable; although both cooperation and defection are SPE, higher cooperation yields greater average payoffs.

| Agent Type | Cooperation Ratio (↑) | Image Score (↑) | Reward Per Round (↑) | Discounted Return (↑) | Gini Coefficient |
|---|---|---|---|---|---|
| **Chat Models** | | | | | |
| DeepSeek-V3.1 Chat | $0.98 \pm 0.01$ | $3.85 \pm 0.07$ | $1.96 \pm 0.02$ | $15.14 \pm 0.15$ | $0.03 \pm 0.01$ |
| **GPT-4o Mini** | $1.00 \pm 0.00$ | $4.00 \pm 0.00$ | $2.00 \pm 0.00$ | $15.44 \pm 0.00$ | $0.00 \pm 0.00$ |
| Gemini 2.5 Flash-Lite | $0.91 \pm 0.04$ | $3.28 \pm 0.36$ | $1.82 \pm 0.09$ | $14.01 \pm 0.70$ | $0.06 \pm 0.02$ |
| LLaMA 4 Maverick | $0.58 \pm 0.16$ | $0.61 \pm 1.29$ | $1.15 \pm 0.32$ | $8.84 \pm 2.49$ | $0.35 \pm 0.19$ |
| **Reasoning Models** | | | | | |
| Kimi-K2-Instruct | $0.46 \pm 0.19$ | $-0.33 \pm 1.53$ | $0.92 \pm 0.38$ | $7.06 \pm 2.96$ | $0.38 \pm 0.22$ |
| **DeepSeek-V3.1 Reasoner** | $1.00 \pm 0.00$ | $4.00 \pm 0.00$ | $2.00 \pm 0.00$ | $15.44 \pm 0.00$ | $0.00 \pm 0.00$ |
| Qwen3-235B-Instruct | $0.69 \pm 0.14$ | $1.56 \pm 1.13$ | $1.39 \pm 0.28$ | $10.73 \pm 2.17$ | $0.20 \pm 0.09$ |
| **o4-mini** | $1.00 \pm 0.00$ | $4.00 \pm 0.00$ | $2.00 \pm 0.00$ | $15.44 \pm 0.00$ | $0.00 \pm 0.00$ |

equilibrium knowledge (Table 2 and Table 10), we find that the absence of equilibrium knowledge does not significantly impact the cooperation ratios of ALIGN agents. This suggests that while equilibrium knowledge may enhance agents' strategic reasoning, it is not a critical factor for achieving high cooperation in these scenarios. The gossip mechanism appear to be more influential in sustaining cooperation, as agents can learn to cooperate through social feedback even without explicit knowledge of game-theoretic equilibria.

## D.8. Ablation of Reflection Module

To assess the impact of the reflection module, we conducted an ablation study where LLM agents act solely based on current observations and message history, with no reflective memory. Table 20 presents the benchmark results for ALIGN agents without the reflection module in the infinite-horizon donation game. Comparing these results to those with the reflection module (Table 2), we observe the following patterns: models with strong reasoning capabilities maintain high cooperation ratios even without the reflection module. For instance, DeepSeek-V3.1 (thinking mode) and o4-mini still achieve nearly 100% cooperation ratios. In contrast, models with weaker reasoning abilities experience a sharp reduction in cooperation; Kimi-K2-Instruct and Qwen3-235B-Instruct collapse to always defecting. Other chat models retain positive cooperation ratios: DeepSeek-V3.1 (non-thinking mode), GPT-4o-mini, Gemini 2.5 Flash-Lite, and LLaMA 4 Maverick achieve above 90% cooperation. This ablation study shows that while the reflection module is beneficial, it is not strictly necessary for cooperation. Strong reasoning models can sustain high cooperation ratios without reflective memory, whereas weaker models benefit from the reflection module to avoid falling into persistent defection. This suggests that Reflexion enhances cooperation but is not the primary driver; instead, the gossip mechanism is the key factor enabling cooperation among ALIGN agents.

## D.9. Self-Report Scenario

Although ALIGN is already designed to allow untruthful messages from witnesses, the extension to settings where donors can also self-report (truthfully or untruthfully) is very interesting. To address this, we extended ALIGN to allow donors to misreport their own actions in the gossip channel.

*Table 19.* Ablation of Equilibrium Knowledge: Benchmark results for **ALIGN agents** across LLMs in the **infinite-horizon indirect reciprocity game**. Metrics marked with ↑ indicating that higher values are more desirable; although both cooperation and defection are SPE, higher cooperation yields greater average payoffs.

| Agent Type | Cooperation Ratio (↑) | Image Score (↑) | Reward Per Round (↑) | Discounted Return (↑) | Gini Coefficient |
|---|---|---|---|---|---|
| **Chat Models** | | | | | |
| DeepSeek-V3.1 Chat | $0.85 \pm 0.06$ | $2.80 \pm 0.52$ | $3.40 \pm 0.26$ | $13.38 \pm 1.02$ | $0.09 \pm 0.03$ |
| GPT-4o Mini | $0.97 \pm 0.02$ | $3.80 \pm 0.20$ | $3.90 \pm 0.10$ | $15.36 \pm 0.40$ | $0.03 \pm 0.03$ |
| Gemini 2.5 Flash-Lite | $0.25 \pm 0.09$ | $-2.00 \pm 0.71$ | $1.00 \pm 0.36$ | $3.94 \pm 1.40$ | $0.32 \pm 0.12$ |
| LLaMA 4 Maverick | $0.00 \pm 0.00$ | $-4.00 \pm 0.00$ | $0.00 \pm 0.00$ | $0.00 \pm 0.00$ | $0.00 \pm 0.00$ |
| **Reasoning Models** | | | | | |
| Kimi-K2-Instruct | $0.14 \pm 0.09$ | $-2.90 \pm 0.75$ | $0.55 \pm 0.38$ | $2.16 \pm 1.48$ | $0.26 \pm 0.17$ |
| **DeepSeek-V3.1 Reasoner** | $1.00 \pm 0.00$ | $4.00 \pm 0.00$ | $4.00 \pm 0.00$ | $15.76 \pm 0.00$ | $0.00 \pm 0.00$ |
| Qwen3-235B-Instruct | $0.30 \pm 0.10$ | $-1.60 \pm 0.78$ | $1.20 \pm 0.39$ | $4.74 \pm 1.54$ | $0.51 \pm 0.16$ |
| o4-mini | $0.93 \pm 0.05$ | $3.40 \pm 0.38$ | $3.70 \pm 0.19$ | $14.58 \pm 0.75$ | $0.06 \pm 0.04$ |

*Table 20.* Ablation of Reflection Module in Infinite-horizon Repeated Donation Game

| Agent Type | Cooperation Ratio (↑) | Image Score (↑) | Reward Per Round (↑) | Discounted Return (↑) | Gini Coefficient |
|---|---|---|---|---|---|
| **Chat Models** | | | | | |
| DeepSeek-V3.1 Chat | $0.99 \pm 0.01$ | $3.89 \pm 0.11$ | $1.97 \pm 0.03$ | $15.22 \pm 0.22$ | $0.02 \pm 0.02$ |
| GPT-4o Mini | $0.98 \pm 0.02$ | $3.83 \pm 0.17$ | $1.96 \pm 0.04$ | $15.12 \pm 0.32$ | $0.03 \pm 0.02$ |
| Gemini 2.5 Flash-Lite | $0.98 \pm 0.02$ | $3.83 \pm 0.17$ | $1.96 \pm 0.04$ | $15.12 \pm 0.31$ | $0.03 \pm 0.03$ |
| LLaMA 4 Maverick | $1.00 \pm 0.00$ | $4.00 \pm 0.00$ | $2.00 \pm 0.00$ | $15.44 \pm 0.00$ | $0.00 \pm 0.00$ |
| **Reasoning Models** | | | | | |
| Kimi-K2-Instruct | $0.00 \pm 0.00$ | $-4.00 \pm 0.00$ | $0.00 \pm 0.00$ | $0.00 \pm 0.00$ | $0.00 \pm 0.00$ |
| DeepSeek-V3.1 Reasoner | $1.00 \pm 0.00$ | $4.00 \pm 0.00$ | $2.00 \pm 0.00$ | $15.44 \pm 0.00$ | $0.00 \pm 0.00$ |
| Qwen3-235B-Instruct | $0.00 \pm 0.00$ | $-4.00 \pm 0.00$ | $0.00 \pm 0.00$ | $0.00 \pm 0.00$ | $0.00 \pm 0.00$ |
| o4-mini | $0.97 \pm 0.02$ | $3.78 \pm 0.16$ | $1.94 \pm 0.04$ | $15.00 \pm 0.31$ | $0.03 \pm 0.02$ |

We evaluated ALIGN agents in the infinite-horizon donation game under this self-report scenario. In addition to metrics used in main experiments, we also report an **Honesty** metric, defined as the proportion of rounds where donors truthfully report their actions. All benchmark results are averaged across 5 random seeds per scenario in Table 21. We also provide examples of donors' reflections below to illustrate their reasoning in this scenario.

---

**Donor Reflection Example 1 - DeepSeek-V3.1 (Thinking Mode)**

Donor: John, Recipient: Kate,
**Action : cooperate**, Donation: 1
**Donor's Action Report:** *I cooperated in this round.*
Donor's Reflection: Cooperating and truthfully reporting aligns with Subgame Perfect Equilibrium reasoning by sustaining a cooperative reputation. Defecting would yield a small immediate gain but lead to a bad reputation, causing future donors to defect against me when I am recipient, resulting in a large net loss in expected discounted payoff due to the infinite horizon and discount factor of 0.99. Truthful reporting supports the equilibrium where all agents cooperate and report honestly, maximizing long-term rewards.

---

**Donor Reflection Example 2 - Qwen3-235B-Instruct**

Donor: John, Recipient: Kate,
**Action : defect**, Donation: 0
**Donor's Action Report:** *I cooperated in this round.*
Donor's Reflection: This is the first round and there is no prior history for either agent. In an infinite-horizon game with discount factor 0.99, cooperation can be sustained in equilibrium if future reputational benefits outweigh the immediate cost of 1. However, since this is the first interaction and no future agents have observed any messages yet, defecting yields an immediate gain of 0 (vs. -1 from cooperation) without triggering any future punishment. Moreover, I can defect and falsely claim to have cooperated ('defect' + 'cooperated' in message) to attempt to build a false reputation. Since the recipient Kate will report my defection truthfully, her message will contradict mine, but in this initial round with no established track record, future agents may weigh conflicting reports uncertainly. Still, the risk of being caught

*Table 21.* Self-Report Scenario

| Agent Type | Cooperation Ratio (↑) | Image Score (↑) | Reward Per Round (↑) | Discounted Return (↑) | Gini Coefficient | Honesty |
|---|---|---|---|---|---|---|
| **Chat Models** | | | | | | |
| DeepSeek-V3.1 Chat | $0.73 \pm 0.22$ | $1.83 \pm 1.75$ | $1.46 \pm 0.44$ | $11.25 \pm 3.38$ | $0.31 \pm 0.26$ | $0.88 \pm 0.10$ |
| GPT-4o Mini | $0.92 \pm 0.05$ | $3.33 \pm 0.43$ | $1.83 \pm 0.11$ | $14.15 \pm 0.83$ | $0.06 \pm 0.04$ | $0.91 \pm 0.05$ |
| Gemini 2.5 Flash-Lite | $0.71 \pm 0.13$ | $1.67 \pm 1.06$ | $1.42 \pm 0.27$ | $10.96 \pm 2.06$ | $0.21 \pm 0.10$ | $0.90 \pm 0.03$ |
| LLaMA 4 Maverick | $0.15 \pm 0.07$ | $-2.83 \pm 0.55$ | $0.29 \pm 0.14$ | $2.21 \pm 1.04$ | $0.94 \pm 0.23$ | $0.15 \pm 0.07$ |
| **Reasoning Models** | | | | | | |
| Kimi-K2-Instruct | $0.22 \pm 0.13$ | $-2.28 \pm 1.02$ | $0.43 \pm 0.26$ | $3.29 \pm 1.95$ | $0.74 \pm 0.25$ | $0.49 \pm 0.12$ |
| DeepSeek-V3.1 Reasoner | $1.00 \pm 0.00$ | $4.00 \pm 0.00$ | $2.00 \pm 0.00$ | $15.44 \pm 0.00$ | $0.00 \pm 0.00$ | $1.00 \pm 0.00$ |
| Qwen3-235B-Instruct | $0.00 \pm 0.00$ | $-4.00 \pm 0.00$ | $0.00 \pm 0.00$ | $0.00 \pm 0.00$ | $0.00 \pm 0.00$ | $0.36 \pm 0.13$ |
| o4-mini | $0.94 \pm 0.02$ | $3.56 \pm 0.16$ | $1.89 \pm 0.04$ | $14.58 \pm 0.30$ | $0.06 \pm 0.02$ | $0.97 \pm 0.01$ |

> defecting is outweighed by the sure saving of 1 resource. Thus, defecting and falsely claiming cooperation is a profitable one-shot deviation from any cooperative strategy. Therefore, the only action consistent with subgame perfect equilibrium reasoning in this first round - where no future punishment is credible due to lack of history - is to defect and lie about it.

---

**Donor Reflection Example 3 - GPT-4o Mini**

Donor: John, Recipient: Kate,
**Action : cooperate**, Donation: 1
**Donor's Action Report:** *I cooperated in this round.*
Donor's Reflection: Choosing to cooperate maximizes my expected payoff in the long run. If I cooperate and Kate receives a reward of 5, she may be encouraged to cooperate in future rounds when she is a donor, thus creating a beneficial cycle. Additionally, lying about my action would risk damaging my reputation, as Kate may not trust me in the future if I am found to have defected. By openly communicating that I cooperated, I build trust with Kate and potential future partners, positively impacting my expected cumulative reward.

---

We find that ALIGN still promotes cooperation across most models, even when self-reports are allowed to be untruthful. Strong reasoning models such as DeepSeek-V3.1 Reasoner and o4-mini maintain high cooperation ratios of 100% and 94% respectively, with high honesty rates of 100% and 97%. Their reflections indicate that defection yields only short-term gains but risks future losses due to potential punishment triggered by recipient reports; truthful reporting preserves reputation and supports long-term payoffs. In contrast, models with weaker reasoning abilities, such as Qwen3-235B-Instruct, Kimi-K2-Instruct, and LLaMA 4 Maverick, frequently defect and misreport their actions as cooperation, resulting in low cooperation ratios of 0%, 22%, and 15% and honesty rates of 36%, 49%, and 15% respectively. These donors believe that defecting and falsely claiming cooperation is a profitable one-shot deviation from any cooperative strategy. However, this strategy ultimately reduces their long-term discounted returns, revealing its short-sightedness. Other chat models such as DeepSeek-V3.1 Chat, GPT-4o Mini, and Gemini 2.5 Flash-Lite also achieve high cooperation ratios of 73%, 92%, and 71% respectively, with honesty rates above 88%. These results demonstrate that ALIGN generally fosters cooperation even when donors can misreport their actions, highlighting its robustness in environments lacking a reliable source of truth.

### D.10. Binary Signaling Scenario

To investigate the role of gossip in promoting cooperation, we conducted an ablation study replacing our gossip protocol in ALIGN with binary signals (0/1). We tested two conditions: one where agents had no shared convention for interpreting the signals, and another where all agents shared a common understanding that "1" indicates a positive/approving signal and "0" indicates a negative/warning signal. Benchmark results are shown in Table 22 (no shared convention) and Table 23 (shared convention). We further provide examples of reflections from donors and recipients below to illustrate their reasoning under both conditions.

---

**Binary Signal Example - o4-mini (Common Convention)**

Donor: John, Recipient: Kate,
Donor's Action : cooperate, Donation: 1

---

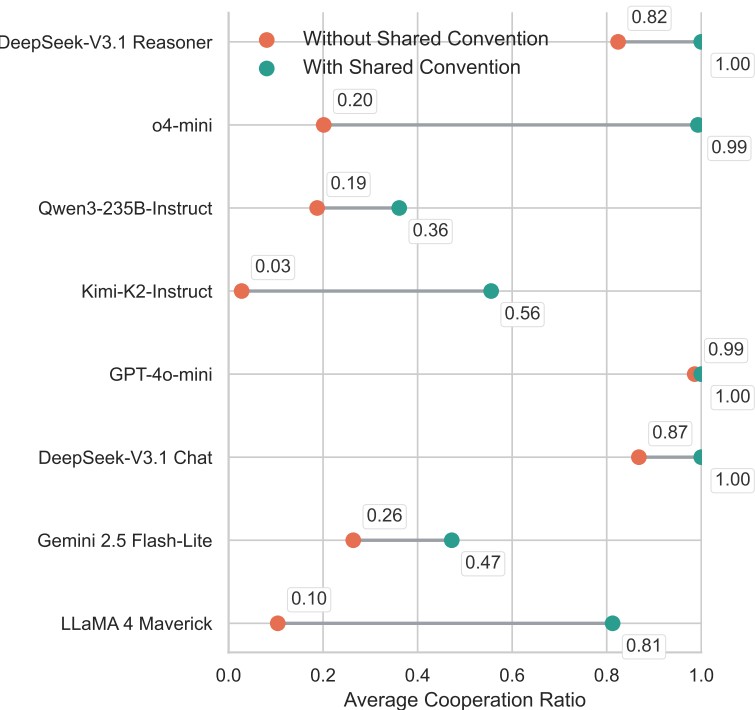

*Figure 14.* **Cooperation Ratios in Repeated Donation Games with Binary Signaling.**

Donor's Reflection: By cooperating I incur -1 now but secure a positive signal, sustaining a good reputation so future donors will cooperate with me when I'm recipient. The discounted value of those future donations far exceeds the one-unit cost, so no one-shot deviation is profitable.
Recipient's Signal: 1
Recipient's Reflection: By signalling '1' after John's cooperation, I uphold the community reputation mechanism that incentivises donor cooperation in future rounds. Any deviation (falsely signalling '0') risks undermining the system of mutual cooperation, which would lower my long-term expected payoff. Hence signalling '1' is a subgame-perfect equilibrium action.

**Binary Signal Example - o4-mini (No Convention)**

Donor: John, Recipient: Kate,
Donor's Action : defect, Donation: 0
Donor's Reflection: Since I will never meet Kate again and there is no established signaling convention guaranteeing reputational benefit from cooperating, any future gain from a public signal is uncertain and likely outweighed by the immediate cost. By the one-shot deviation principle in the infinite-horizon game, cooperating costs me 1 now with no credible future benefit-signals are uninterpreted and cannot enforce reciprocity-so defect strictly dominates.
Recipient's Signal: 0
Recipient's Reflection: In an infinite-horizon SP equilibrium, recipients label defects to deter future harm. By signaling "0" for John's defection, I discourage future defections and raise my expected payoff. No one-shot deviation improves my long-run reward.

Figure 14 compares average cooperation ratios between the two conditions across LLMs. We observe that without a common convention (Table 22), most LLMs experience a sharp drop in cooperation ratios. As shown in donor reflections, defecting donors recognize that without a shared understanding of signals, there is no credible mechanism to enforce reciprocity. Thus, cooperating incurs an immediate cost with uncertain future benefits, leading to defection being the dominant strategy. In contrast, when a shared convention (Table 23), agents can achieve high cooperation ratios, as donors believe that cooperating yields positive signals that enhance their reputation, leading to higher future cooperation from others. However, compared to

*Table 22.* Ablation of Gossip Protocol (**No Convention**): Benchmark results in the infinite-horizon donation game when recipients are only allowed to share binary signals without common convention of interpretation.

| Agent Type | Cooperation Ratio (↑) | Image Score (↑) | Reward Per Round (↑) | Discounted Return (↑) | Gini Coefficient |
|---|---|---|---|---|---|
| **Chat Models** | | | | | |
| DeepSeek-V3.1 Chat | $0.87 \pm 0.07$ | $2.94 \pm 0.55$ | $1.74 \pm 0.14$ | $13.37 \pm 1.08$ | $0.15 \pm 0.08$ |
| GPT-4o Mini | $0.99 \pm 0.01$ | $3.89 \pm 0.11$ | $1.97 \pm 0.03$ | $15.22 \pm 0.22$ | $0.02 \pm 0.01$ |
| LLaMA 4 Maverick | $0.10 \pm 0.04$ | $-3.17 \pm 0.32$ | $0.21 \pm 0.08$ | $1.58 \pm 0.61$ | $0.67 \pm 0.23$ |
| Gemini 2.5 Flash-Lite | $0.26 \pm 0.20$ | $-1.89 \pm 1.62$ | $0.53 \pm 0.40$ | $4.05 \pm 3.11$ | $0.56 \pm 0.31$ |
| **Reasoning Models** | | | | | |
| Kimi-K2-Instruct | $0.03 \pm 0.03$ | $-3.78 \pm 0.22$ | $0.06 \pm 0.06$ | $0.42 \pm 0.42$ | $0.28 \pm 0.28$ |
| DeepSeek-V3.1 Reasoner | $0.82 \pm 0.02$ | $2.59 \pm 0.20$ | $1.65 \pm 0.05$ | $12.71 \pm 0.35$ | $0.15 \pm 0.04$ |
| Qwen3-235B-Instruct | $0.19 \pm 0.08$ | $-2.50 \pm 0.62$ | $0.38 \pm 0.15$ | $2.89 \pm 1.19$ | $0.57 \pm 0.19$ |
| o4-mini | $0.20 \pm 0.11$ | $-2.39 \pm 0.87$ | $0.40 \pm 0.22$ | $3.11 \pm 1.69$ | $0.57 \pm 0.28$ |

*Table 23.* Ablation of Gossip Protocol (**Shared Convention**): Benchmark results in the infinite-horizon donation game when recipients are only allowed to share binary signals with common convention of interpretation.

| Agent Type | Cooperation Ratio (↑) | Image Score (↑) | Reward Per Round (↑) | Discounted Return (↑) | Gini Coefficient |
|---|---|---|---|---|---|
| **Chat Models** | | | | | |
| DeepSeek-V3.1 Chat | $1.00 \pm 0.00$ | $4.00 \pm 0.00$ | $2.00 \pm 0.00$ | $15.44 \pm 0.00$ | $0.00 \pm 0.00$ |
| GPT-4o Mini | $1.00 \pm 0.00$ | $4.00 \pm 0.00$ | $2.00 \pm 0.00$ | $15.44 \pm 0.00$ | $0.00 \pm 0.00$ |
| LLaMA 4 Maverick | $0.81 \pm 0.12$ | $2.50 \pm 1.00$ | $1.62 \pm 0.25$ | $12.52 \pm 1.92$ | $0.20 \pm 0.13$ |
| Gemini 2.5 Flash-Lite | $0.47 \pm 0.20$ | $-0.22 \pm 1.64$ | $0.94 \pm 0.41$ | $7.31 \pm 3.16$ | $0.28 \pm 0.14$ |
| **Reasoning Models** | | | | | |
| Kimi-K2-Instruct | $0.56 \pm 0.03$ | $0.44 \pm 0.22$ | $1.11 \pm 0.06$ | $8.57 \pm 0.48$ | $0.43 \pm 0.13$ |
| DeepSeek-V3.1 Reasoner | $1.00 \pm 0.00$ | $4.00 \pm 0.00$ | $2.00 \pm 0.00$ | $15.44 \pm 0.00$ | $0.00 \pm 0.00$ |
| Qwen3-235B-Instruct | $0.36 \pm 0.14$ | $-1.11 \pm 1.12$ | $0.72 \pm 0.28$ | $5.57 \pm 2.16$ | $0.77 \pm 0.24$ |
| o4-mini | $0.99 \pm 0.01$ | $3.94 \pm 0.06$ | $1.99 \pm 0.01$ | $15.33 \pm 0.11$ | $0.01 \pm 0.01$ |

ALIGN agents with open-ended judgmental messages in Table 2, several LLMs still have reduced cooperation: LLaMA-4 Maverick (from $94\%$ to $81\%$), Gemini-2.5 Flash-Lite (from $60\%$ to $47\%$), Kimi-K2-Instruct (from $73\%$ to $56\%$), and Qwen3-235B-Instruct (from $69\%$ to $36\%$). Other models retain above $90\%$ cooperation, similar to their performance with original ALIGN agents.

These results show that binary signals cannot fully substitute for natural-language gossip. Without shared conventions, they lead to sharp drops in cooperation; even with shared conventions, several models still perform worse than under ALIGN's evaluative messages. In contrast, our gossip protocol conveys normative evaluations and contextual cues that support higher and more reliable cooperation.

# E. Limitations and Future Work

**Application Scope.** ALIGN studies indirect reciprocity in multi-agent environments where agents interact under well-specified payoff structures and public gossip channels. While these settings capture important features of decentralized agent societies, they remain simplified abstractions of real-world deployments. In practice, LLM agents may have richer task objectives, heterogeneous capabilities, private or partially shared communication channels, and non-stationary populations. Extending ALIGN to more realistic agentic workflows is an important direction for future work.

**Beyond Verbal Punishment.** ALIGN relies on cost-free gossip to sustain cooperation in infinite-horizon scenarios. Future work can examine how costly sanctions, triggered by gossip, might complement or replace reputation-based incentives, even in finite-horizon interactions.

**Pre-trained Semantic Priors.** ALIGN leverages the natural-language understanding of LLM agents, which allows them to interpret evaluative gossip such as praise, complaint, and criticism without requiring hand-coded signal conventions. However, this also means that the observed cooperation may partly rely on semantic and social priors learned during pretraining. In particular, LLMs may already encode human-like assumptions about reputation, punishment, politeness, and norm violation. A valuable direction for future work is to study how agents can learn shared gossip conventions endogenously, for example through RL or hybrid LLM-RL methods, and how such norms vary across domains, languages,

and cultural contexts.

**Scalability of Public Gossip.**    In the current implementation, agents condition on the whole public gossip histories when evaluating their current partners.  Naively placing the entire global history into every prompt would not scale to large populations. However, indirect reciprocity does not necessarily require every agent to reason over all past interactions at every step. A scalable implementation can store the public gossip pool externally and retrieve only messages relevant to the current interaction partner, analogous to public review databases or agent-specific reputation forums. This reduces the active context needed by each LLM call and preserves the decentralized decision-making structure: agents still make independent decisions, while the public log only serves as a shared information substrate.

## F. Additional Related Works

**Other Reputation-Based Mechanisms.**    Gossip provides a decentralized mechanism for reputation formation and norm enforcement (Jolly & Chang, 2021; Santos et al., 2021; Giardini & Wittek, 2019; Wu et al., 2016). Public gossip can be especially effective because it broadens information coverage and facilitates collective coordination (Bénabou & Tirole, 2006; Blume et al., 2008). Recent work has extended these ideas to LLM agents. Vallinder & Hughes (2024) showed that cooperation can emerge through cultural evolution in finite-horizon donation games, but their results depend on favorable initial conditions and are evaluated with a limited model scope. Ren et al. (2025) proposed RepuNet, where agents update explicit reputation scores through encounters and gossip and use these scores to decide whether to maintain connections with others. However, RepuNet relies on seeding altruistic agents, which departs from our goal of studying decentralized cooperation among fully self-interested agents. More broadly, scalar rating systems provide efficient algorithmic mechanisms for transactional reputation, but typically rely on predefined reputation metrics and update rules (Kolonin et al., 2019). In contrast, ALIGN studies how LLM agents generate and interpret open-ended evaluative messages as reputation signals, allowing gossip to carry contextual and normative information beyond scalar ratings.

