# OpenReview forum: "Talk, Judge, Cooperate: Gossip-Driven Indirect Reciprocity in Self-Interested LLM Agents"
_ICML.cc/2026/Conference — ICML 2026 regular_

### Official Review · Reviewer_ryGz · 2026-02-24

**Soundness:** 3
**Presentation:** 3
**Significance:** 3
**Originality:** 3
**Overall Recommendation:** 5
**Confidence:** 4

**Summary:**

The paper presents ALIGN - a multi-agent simulation framework based on LLM agents, employing gossip-based protocol to enable reputational analysis in a community enabling evolution of cooperation in long-term operations. The simulation is performed for multiple interaction scenarios, sets of social conditions and kinds of agent behavior patterns, using different mainstream LLMs. Simulation results confirm theoretical assumptions regarding cooperative and non-cooperative behavior and raise certain questions regarding capacities of different LLMs to deal with cooperation-based inferences and decision making in chat and reasoning modes.

**Compliance With Llm Reviewing Policy:**

Affirmed.

**Final Justification:**

Partially resolved, and the remaining concerns are not easily addressed in a short rebuttal.
Comments on author rebuttal:

> RL agents soon collapse to defection, while LLMs can leverage prior knowledge to coordinate norms and successfully sustain cooperation.

Since norms are subjects to change and need to be learned if they change or if a newborn agent or agent from another community with other norms enters given community with specific norms, it is LLM-based design what can be called "rigid" :-)

>We agree that finding a better trade-off between computation cost and performance would be a valuable future work.
Yes!

>Q1: Agent Memory
>We want to clarify that ALIGN agents have a private experience memory module (Section 4.2). They store their direct >interaction experiences and internal reflections in their memory,

And that may be be better answer on my earlier question! If the paper is accepted, or in follow-up work, I would like to see more details on how the static LLM is combined with dynamic memory that can fully support social norm learning.

Since I accept the paper, I see no reason for extra boost of the score.

**Key Questions For Authors:**

- How the LLM agents memorize the gossip-based reputation if they have no memories, is that all RAG based so the reputations of the agents are implicitly stored in the gossip logs?

- Line 261 says "the shared gossip log". Why is that shared? If it is shared between all agents, the system can not be claimed as decentralized, right? Can it be elaborated in the paper?

- What is novel or beneficial computing reputation relying on LLM-based conversational interactions, compared to use the explicit ratings log with the same 5 gradations ("five stars" instead of "five tones"), why the same effect can not be achieved with non-LLM agents in more reliable and less expensive way?

- Why, with finite horizon 5.1.1 and infinite horizon with private monitoring (5.1.2): "some chat LLMs reach high cooperation ratios, yielding higher average rewards and low Gini coefficients"? Can it be elaborated in the paper or its Discussion part?

- Why non-gossiping GPT-4o-mini exposes so good results, almost as good as ALIGN (Fig 6.)? Can it be elaborated in the paper or its Discussion part?

- Why ALIGN under-performs with Gemini 2.5 Flash-lite? Can it be elaborated in the paper or its Discussion part?

**Limitations:**

Based on the comments above, it is neither shown nor discussed what is beneficial using LLM-agents for textual processing of the reputations.

Specifically, I assume there is no benefits in terms of reliability and computational cost, compared to more straightforward implementation with explicit ratings instead of implicit gossip.

No complete experiments exploring the issues above are provided.

I would recommend to clarify on the above in the separate Limitations section and not in the appendix "E. Limitations and Future Work"

**Strengths And Weaknesses:**

Strengths:
- Actual problem of enforcing cooperation and building of reliable reputation in a sparsely connected distributed multi-agent environment.
- Multiple scenarios and setups of social norms and protocols are considered.
- Complete reproducible code is provided with exhaustive details in extra appendices.
- Multiple popular LLMs are explored in two modes (chat & reasoning).
- Methodologically solid evaluation with 5 different random seeds.
- Multiple performance metrics involved in evaluations, with economic and social values: average reward per round; cooperation ratio; discounted return; image score; and the Gini coefficient.
- Resilience/robustness study for two different attack vectors.
- Ablation study with different ablations.

Weaknesses:
- Not clear what it is the value of the framework, except to modeling of social interactions in human societies before 20st century where gossips and rumors were the only way for reputation building.
- Comparison to Non-LLM baselines would be expected for all experiments for the same setup, using explicit ratings for the same "5 tones", such as "4 star ratings" on some public ledger, even with partial and scarce/sparse visibility of the ratings within the community (ablation study according to "D.9. Binary Signaling Scenario" is not quite representative covering very specific and limited case).
- No even discussion on possibility of using plain non-gossip reputation systems such as discussed in this work https://arxiv.org/pdf/1905.08036
- The comparative computational cost (and respective carbon footprint) is not discussed while in should be much higher than any system based on explicit ratings.
- The possibility of mining the reputation in the organic review in social media data, which would partially justify the approach, is not provided.
- The paper is too bound to appendices in its structure, I would like it to be more self-contained, while presence of the extra appendices is definitely a plus.
- Fig 6 is poorly readable, need to fix scaling or font size, if the paper is accepted.
- No discussion on findings such as "chat models remain highly cooperative" in ablation study without of reflection memory (5.3), not clear why the non-reasoning models perform good.

Soundness:

>Is the submission technically sound?

Yes

>Are claims well supported (e.g., by theoretical analysis or experimental results)?

Yes

>Are the methods used appropriate?

Yes, with except the entire applicability of the approach is not justified unless: A) it is used for simulation of social development in societies before 20st century or B) it is used for reputation mining in social media data

>If the paper includes theoretical results, are the proofs correct and based on reasonable assumptions?

Yes, partially. Most of the paper is experimental, but it has proofs in appendices.

>If the paper includes empirical results, are the experiments well-designed?

Yes.

>Are the authors careful and honest about evaluating both the strengths and weaknesses of their work?

Yes, generally, with exceptions stated above in the Weaknesses

Presentation:

>Is the submission clearly written and well structured? (If not, please make constructive suggestions for improving its clarity.)

Yes, generally, with exceptions stated above in the Weaknesses

>Is the overall narrative easy to follow?

Yes, generally, with exceptions stated above in the Weaknesses

>Does the work properly position itself in the context of prior/concurrent literature and clearly discuss how it differs?

Yes.
Complete information to reproduce the presented results was provided.


Significance:

>Does the paper address an important or relevant problem?

Yes.

>Does it advance understanding, capabilities, or practice in machine learning?

Yes. It illustrates the expected phenomena pretty well.
Also, it reveals which LLMs in which mode are more or less practical for such simulations.

>Could it influence future research or applications?

I am not sure. The entire applicability of this approach should be further justified based on the comments stated above, starting with comparative evaluation with similar non-LLM solution, comparing both economical and social metrics and computational costs.

>Is the scope of impact broad or specialized, and is that appropriate for the contribution? Even if the improvements are modest or domain-specific, could they unlock new directions or provide practical utility?

The scope is broad and appropriate. Presence of a new directions and practical utility are to be defended on more solid basis.

Originality:

>Does the work provide new insights, deepen understanding, or highlight important properties of existing methods?

Yes, as commented above in Strengths

>Does the work introduce new tasks, methods, theory, data, or perspectives that advance the field in some dimensions?

Yes, it builds gossip-based protocol for development of implicit decentralized reputation system.

>Does this work offer a novel combination of existing techniques, and is the reasoning behind this combination well-articulated?

Yes.

> Are the contributions clearly distinguished from closely related literature, and is the novelty well justified?

Yes.

---

> ### Author Rebuttal · Authors · 2026-03-31
>
> We sincerely thank you for the constructive feedback. Your suggestions regarding non-LLM baselines, real-world applicability, and model behaviors provide excellent directions to significantly strengthen the discussion of our final paper. We address your points below.
>
> **W1 & W5: Modern Relevance and Real-World Applications**
>
> We respectfully push back on the framing that gossip-based reputation is limited to pre-20th-century societies. Modern online reputation systems, such as agent-specific public forums (e.g., Moltbook), public blockchain-anchored review ledgers, and public e-commerce review systems function as decentralized, text-based gossip logs at scale. As autonomous LLM agents are increasingly deployed in real-world ecosystems, they will need to navigate these exact unstructured, mixed-motive environments. Additionally, as you astutely pointed out in W5, mining organic text reviews on social media to compute reputations is also a realistic application of the ALIGN framework.
>
> **W2, W3, W4 & Q3: Discussion on Non-LLM Baselines with Ratings**
>
> We appreciate the reviewer highlighting the literature on rating-based framework in Kolonin et al. (2019).
>
> - **RL Baselines Fail:** To better contextualize our contribution, we have added new RL baselines utilizing a 5-level rating signal. RL policies with 5-level hierarchical public rating rapidly converge to zero cooperation. Due to character limit, please refer details and discussion in **rebuttal to W1 of Reviewer Fiws**.
> - **Discussion of Kolonin et al., 2019**: While weighted liquid rank systems are effective for transaction-based e-commerce, they rely on rigid, pre-defined behavioral rules (e.g., hardcoded selection policies such as "winner take all") and fixed arithmetic ledgers that are fundamentally limited to the specific parameters of this simulated marketplace game. In contrast, ALIGN is fully adaptive and allows agents to strategically gossip and make decisions accordingly. Our framework goes beyond hardcoded heuristics to actively optimize for broader objectives. We have added a discussion section in the revised paper to explicitly contrast these paradigms.
> - **Discussion of Computational Cost**: Maintaining a natural language gossip protocol via LLM inference incurs a higher inference cost compared to updating a scalar rating with RL agents driven by smaller models. However, there is a trade-off between computation cost and performance. As we showed above, RL agents soon collapse to defection, while LLMs can leverage prior knowledge to coordinate norms and successfully sustain cooperation. We agree that finding a better trade-off between computation cost and performance would be a valuable future work.
>
> **Q1: Agent Memory**
>
> We want to clarify that ALIGN agents have a private experience memory module (Section 4.2). They store their direct interaction experiences and internal reflections in their memory, which they condition on when inferring the reputation of others. Additionally, agents can retrieve reputation-relevant context from the public gossip pool to assess the opponent's global reputation based on third-party witness reports.
>
> **Q2: Shared Log vs. Decentralization**
>
> With a public gossip pool, the decision-making process remains fully decentralized because (1) private observations are retained since agents do not share their raw or even truthful observations; (2) decision policies are entirely independent among agents; (3) no central authority exists to verify truth, or enforce compliance.
>
> **W8, Q4, Q5, Q6: Explanation of Chat Models Performance**
>
> - **The "Over-Cooperation" Anomaly in Chat Models**. As you have observed, indeed, some chat models achieve high cooperation even without gossip, without reflection, or in finite horizons where defection is strictly optimal. **This is not a success; it is a failure of rationality.** With poor reasoning capabilities, these chat models fail to make strategically optimal decisions. In contrast, strong reasoning models such as deepseek-reasoner can achieve theoretically aligned results, refusing cooperation when defection is strictly optimal. This reinforces our core conclusion that advanced reasoning capabilities drive incentive-aligned behavior, while basic chat models blindly over-cooperate.
> - **Underperformance of Gemini 2.5 Flash-lite**. Flash-lite's underperformance stems from its limited reasoning capacity too. By analyzing its reflection traces, we found that Flash-lite struggles to map the historical public gossip to the long-term strategic penalty of defection. It fails to grasp the conditional nature of the social norm, leading to policy collapse.
>
> **W6 & W7: Presentation Improvements**. We have increased the Figure 6 size to ensure readability. We agree the paper relies heavily on appendices. If accepted, we will utilize the extra camera-ready page to move more critical empirical results and baseline ablations directly into the main text to make the paper more self-contained.

---

> > ### Author Rebuttal · Reviewer_ryGz · 2026-04-01
> >
> > Thank you for the nice discussion which may mot fit the rebuttal process, just few points:
> >
> > > RL agents soon collapse to defection, while LLMs can leverage prior knowledge to coordinate norms and successfully sustain cooperation.
> >
> > Since norms are subjects to change and need to be learned if they change or if a newborn agent or agent from another community with other norms enters given community with specific norms, it is LLM-based design what can be called "rigid" :-)
> >
> > >We agree that finding a better trade-off between computation cost and performance would be a valuable future work.
> > Yes!
> >
> > >Q1: Agent Memory
> > >We want to clarify that ALIGN agents have a private experience memory module (Section 4.2). They store their direct >interaction experiences and internal reflections in their memory,
> >
> > And that may be be better answer on my earlier question! If the paper is accepted, or in follow-up work, I would like to see more details on how the static LLM is combined with dynamic memory that can fully support social norm learning.
> >
> > Since I accept the paper, I see no reason for extra boost of the score.

---

> > > ### Author Response · Authors · 2026-04-01
> > >
> > > Thank you for the engaging discussion! We are glad to learn that the clarification regarding the dynamic Agent Memory module was helpful!
> > >
> > > We are glad that we both believe finding a better trade-off between computation cost and performance would be a valuable future work. Indeed, there could be more adaptive methods, such as possibly full RL-based method. This is an exciting future direction.
> > >
> > > We agree that LLM-based designs have their limitations (for example, the preferences could be relatively fixed; some prior knowledge based on human society could exist in LLMs). Regarding Kolonin et al., 2019, we have revised and added the following discussion to the manuscript:
> > >
> > > - **Discussion of Kolonin et al., 2019**: Both Kolonin et al., 2019 and our work explore multi-agent reputation systems. Kolonin et al., 2019 tackles economic reliability by formalizing a quantitative, algorithmic reputation metric designed to secure e-commerce transactions against financial scams. Our paper explores indirect reciprocity based on game-theoretical foundations, leveraging the natural language capabilities of modern AI to simulate how open-ended gossip and strategic communication can organically enhance cooperation among self-interested entities.
> > >
> > > We hope this revision could better describe the position of the works.

---

### Official Review · Reviewer_8ipr · 2026-03-10

**Soundness:** 3
**Presentation:** 3
**Significance:** 3
**Originality:** 3
**Overall Recommendation:** 4
**Confidence:** 4

**Summary:**

The paper investigates whether decentralized, self-interested LLM agents can sustain cooperation in mixed-motive environments through a public gossip mechanism. It introduces ALIGN, a framework in which agents act, reflect, and broadcast evaluative natural-language messages, and studies this setup through the lens of repeated donation and indirect reciprocity games. Empirically, the authors report that gossip improves cooperation, discounted returns, and social welfare relative to no-gossip baselines in infinite-horizon settings, and that agents often reduce cooperation toward always-defect “greedy” entrants. The paper’s main claim is that public gossip can function as a decentralized reputation mechanism capable of supporting cooperation without centralized scoring or seeded altruistic agents.

**Compliance With Llm Reviewing Policy:**

Affirmed.

**Final Justification:**

My main concerns have been addressed in the authors' rebuttal, as well as in responses to other reviewers. I have provided several suggestions that could further improve the quality of the paper, and I hope the authors will incorporate them. The theoretical contributions are somewhat limited; however, this is largely due to the nature of the problem. The empirical evaluation appears thorough, and the additional experiments included in the rebuttal led me to increase my score.

**Key Questions For Authors:**

- The theory assumes truthful recipient reporting. Why does this model gossip rather than an honest public-signal mechanism, given that real gossip can be strategic, selective, or manipulative?
- The prompts provide substantial game-theoretic guidance. How much of the result survives if this is removed more fully, rather than only through the current limited ablation?
- Why is open-ended natural language necessary? The current comparisons do not yet rule out the possibility that the gains come simply from having a shared public reputation channel.
- The robustness analysis includes a simple adversarial agent, but why not test adversaries that manipulate the gossip channel itself, for example by lying or fabricating accusations?

*Minor comments*

There is a substantial literature on the mechanics of gossip, reputations, and indirect reciprocity, but the paper engages only limitedly with it and focuses mainly on the LLM angle.

[1] Kawakatsu, M., Kessinger, T. A., & Plotkin, J. B. (2024). A mechanistic model of gossip, reputations, and cooperation. *Proceedings of the National Academy of Sciences*, 121(20), e2400689121.

[2] Murase, Y., & Hilbe, C. (2024). Computational evolution of social norms in well-mixed and group-structured populations. *Proceedings of the National Academy of Sciences*, 121(33), e2406885121.

[3] Murase, Y., & Hilbe, C. (2024). Indirect reciprocity under opinion synchronization. *Proceedings of the National Academy of Sciences*, 121(48), e2418364121.

**Limitations:**

Yes

**Strengths And Weaknesses:**

*Strengths*
- The paper tackles a clear question: whether public gossip can help sustain cooperation among self-interested LLM agents. Given the likely growth of multi-agent scenarios, understanding how cooperation can emerge without explicitly engineering it is an important question.
- The ALIGN framework is not overly complicated, and the repeated donation / indirect reciprocity setting provides a simple testbed for studying gossip as a public reputation signal.
- The experiments go beyond comparing average cooperation levels by also examining what happens when a greedy always-defect agent is introduced. This makes the evaluation somewhat more informative and provides additional support for the proposed approach.

*Weaknesses*
- The main claim is stronger than what the paper really shows. The theory depends on truthful public reporting, so the mechanism is closer to an honest public-signal model than to realistic gossip, where agents may lie, collude, or manipulate reputations.
- The prompts may be doing a substantial part of the work. Agents are explicitly instructed to be self-interested, rational, and far-sighted, and some settings additionally provide equilibrium-style reasoning cues. This makes it difficult to tell whether cooperation truly emerges from the mechanism or is largely induced by the prompt.
- The empirical evaluation is fairly limited. The environments are highly stylized, the number of seeds is small, and the robustness analysis only considers a simple always-defect entrant rather than adversaries that actively exploit or manipulate the gossip channel.
- The paper does not yet convincingly isolate whether natural-language gossip is the key ingredient. The gains may simply come from giving agents access to a shared public reputation signal, and stronger baselines with simpler symbolic or structured communication would be needed to establish the added value of open-ended language.

---

> ### Author Rebuttal · Authors · 2026-03-31
>
> We thank the reviewer for the feedback. We address your questions below, clarifying key aspects of our theoretical framework, prompt design, and ablation studies.
>
> **W1/Q1: The assumption of strategic gossip v.s. honesty**
>
> We want to clarify that our framework does **not** enforce honest reporting as a policy constraint; ALIGN explicitly allows for the strategic, noisy, and manipulative communication the reviewer highlights.
>
> - **Clarification on Theoretical Validity:** We do not assume honesty. Instead, we prove the *existence* of a cooperative equilibrium by demonstrating that honest reporting can emerge as an *incentive-compatible strategy*. We show mathematically that self-interested agents can strategically choose to follow grim-trigger policy and report truthfully without incentive to unilaterally deviate. Additionally, as supported by the Folk Theorem [2], the grim-trigger policy example in the proof of Proposition 3.6 is just one of infinitely many possible equilibria which also include robust norms that resist malicious mutants [1].
> - **Empirical Validation of Strategic Choice:** In practice, ALIGN framework never forces agents to share truthful private observations. Agents are entirely free to generate noisy, manipulative, or untruthful gossip to optimize their utility strategically. As shown in our robustness test of the "Self-Report Scenario" (Appendix D.8), when actors are allowed to misreport their actions, weaker models (e.g., Qwen) attempt to manipulate their reputations by lying, leading to lower long-term utility. Conversely, strong reasoning models (DeepSeek, o4-mini) logically deduce that truthful reporting preserves reputation, achieving high cooperation and honesty *without* forced constraints. Thus, ALIGN captures the realistic strategic choices of LLM agents, confirming that honesty is a rational choice for strong reasoning models rather than a system requirement.
>
> **W2/Q2: Prompt influence and ablation study**
>
> We clarify that our prompts define the theoretical utility function, rather than inducing behavioral compliance.
>
> - **Prompts Define the Utility Function:** Instructing agents to be "self-interested, rational, and far-sighted" maps our empirical setup to our theoretical objective defined in Section 3 where each agent's objective is to rationally maximize a self-interested far-sighted utility.
> - **Empirical Evidence of Strategic Defection under the Same Objective (Section 5.1):** The objective prompts do not force cooperation. We use the exact same prompts ("self-interested, rational, and far-sighted") across all settings, including finite-horizon scenarios and infinite-horizon scenarios without public gossip. However, most LLMs (especially strong reasoning models) achieve near-zero cooperation in these scenarios. This demonstrates that cooperation is not induced by the objective prompt, but emerges only when the structural mechanics (i.e., the public gossip channel) make it strategically beneficial.
> - **Ablation of Equilibrium Cues (Section 5.3):** We entirely ablated equilibrium cues. Without these cues, strong reasoning models like DeepSeek-V3.1 Reasoner and o4-mini still sustained near-perfect cooperation, demonstrating that explicit equilibrium cues are not the main driver of cooperation.
>
> **W3/Q4: Robustness to manipulative gossipers**
>
> To test vulnerability to manipulative gossipers, we evaluated ALIGN against coordinated collusive attackers with malicious gossip. Results show that ALIGN agents can mostly identify and ostracize such malicious attackers and achieve higher long-term utility compared to collusive agents. Details are shown and discussed in **rebuttal to W2 of Reviewer NEYc**.
>
> **W4/Q3: Necessity of language-based gossip**
>
> We agree that isolating the impact of natural language is critical, which is why **we explicitly conducted this exact baseline comparison in our manuscript (Section 5.3 and Appendix D.9)**. We replaced open-ended gossip with a binary signal to test a simple shared public reputation channel. Results show that cooperation collapses entirely without a common interpretation of signals. Even with a shared interpretive convention, binary signaling still underperforms ALIGN's natural language for several LLMs. These findings prove that simply giving agents access to a shared reputation signal does not guarantee cooperation. ALIGN's open-ended language is advantageous because it provides evaluative and normative context, which improves both the level and reliability of indirect reciprocity.
>
> # References
>
> [1] Ohtsuki, H., & Iwasa, Y. (2006). The leading eight: social norms that can maintain cooperation by indirect reciprocity. *Journal of theoretical biology*, *239*(4), 435-444.
>
> [2] Fudenberg, D., Levine, D., & Maskin, E. (2009). The folk theorem with imperfect public information. In *A Long-Run Collaboration On Long-Run Games* (pp. 231-273).

---

> > ### Author Rebuttal · Reviewer_8ipr · 2026-04-03
> >
> > Thanks for the detailed responses; they address several of my concerns.
> >
> > That said, I think it would help to more clearly separate what is established theoretically from what is observed empirically. As I understand it, Proposition 4 shows the *existence* of a cooperative equilibrium with truthful reporting, while the experiments suggest that some models move in that direction, which is not guaranteed in the presence of multiple equilibria. What remains a bit less clear is how likely this outcome is theoretically, for example in terms of equilibrium selection, convergence, and robustness when gossip is noisy or strategic. I realize that addressing this fully is difficult, but a bit more discussion would help clarify the scope of the result.
> >
> > Regarding the communication channel, my understanding is that gossip is implemented as free-form natural-language messages broadcast after each interaction, rather than as a structured signal. Because of this, it is still somewhat unclear whether the gains come from language itself, or more generally from having a richer shared signal with a common interpretation. For example, it would be interesting to see how much of the effect remains under a simple structured protocol such as "cooperate/defect" or "trustworthy/untrustworthy," beyond the current binary signal comparison.
> >
> > *Minor comments:*
> > - It may help to briefly define “mixed-motive” for readers who are less familiar with the terminology.
> > - Figures 6 and 8 would benefit from improved readability, as the font sizes are quite small.
> > - It would be helpful to clarify to what extent the propositions are new versus adaptations of known results under the paper’s specific assumptions. This became clearer to me in the appendix than in Section 3.
> >
> > I have adjusted my score accordingly and hope the authors find these suggestions helpful in strengthening the paper.

---

> > > ### Author Response · Authors · 2026-04-06
> > >
> > > Thank you for your constructive suggestions! We greatly appreciate your feedback and will incorporate these suggestions into the revised version of the paper.

---

### Official Review · Reviewer_NEYc · 2026-03-11

**Soundness:** 2
**Presentation:** 3
**Significance:** 2
**Originality:** 3
**Overall Recommendation:** 4
**Confidence:** 4

**Summary:**

The paper studies using public gossip for indirect reciprocity among decentralized self-interested LLM agents. It proposes ALIGN, where agents share public evaluative messages about each other and use both private memory and a shared gossip history to decide how to act. The experiments evaluate ALIGN across multiple LLMs, showing that public gossip often increases cooperation, especially for stronger reasoning models.

**Compliance With Llm Reviewing Policy:**

Affirmed.

**Final Justification:**

The authors addressed my concerns. I raise my rating to weak accept.

**Key Questions For Authors:**

None

**Limitations:**

yes

**Strengths And Weaknesses:**

**Strengths**
- The idea of using language gossip as a way to design decentralized reputation system is novel and interesting.

**Weaknesses**
- If I understand correctly, the whole gossip history needs to be stored as text in the public message pool for use by all the agents. It does not seem to be a scalable solution. When the system is large and the number of interactions increases, the raw gossip history could quickly blow up and become a big burden for every agent when they need to make judgements about reputation.
- This system design seems to be very prone to malicious attack (e.g. malicious agents spreading false gossip against others).

---

> ### Author Rebuttal · Authors · 2026-03-31
>
> Thank you for the constructive feedback. We directly address your concerns regarding scalability and malicious attacks below, supported by new empirical evaluations in https://anonymous.4open.science/r/icml26_rebuttal-6080/ that will be added to the revision.
>
> **W1: Scalability of the Gossip History**
>
> This is a good insight. If agents were required to ingest the entire global history of all previous interactions into their prompt, the context window size would grow quadratically, $O(N^2)$ (not exponentially), where $N = |\mathcal{N}|$ is the number of agents. However, sustaining indirect reciprocity does not necessarily require global tracking in the context window.
>
> - **Scalable Retrieval ($O(N)$):** To make the system more scalable, an agent can filter the gossip pool to retrieve *only the gossip logs relevant to their specific current opponent*, which reduces the LLM memory requirement to scale linearly ($O(N)$).
>
> - **New Empirical Validation:** To validate that this filtering does not harm performance, we conducted new experiments where agents are restricted to tracking only opponent-relevant gossip in their context windows. The results (**see Figure 4 in https://anonymous.4open.science/r/icml26_rebuttal-6080/**) demonstrate that most ALIGN models still maintain high cooperation ratios and social welfare.
>
> - **System-Level Feasibility:** The ALIGN public gossip pool is analogous to several existing data structures: public blockchain-anchored review ledgers, public e-commerce review systems, and agent-specific public forums (e.g., Moltbook). Storing the public gossip history in a standard database is available, while system efficiency can be preserved when we only retrieve the relevant information into the LLM's active context window.
>
> **W2: Vulnerability to Malicious Attacks with False Gossip**
>
> We agree that resilience to malicious gossip is vital for decentralized systems. To rigorously test the vulnerability the reviewer describes, we designed a scenario under malicious gossip attack.
>
> - **New Adversarial Evaluation (Collusive Attackers):** We introduced two highly coordinated collusive attackers (Figure 3 in https://anonymous.4open.science/r/icml26_rebuttal-6080/). These malicious agents actively exploit the system by: (1) Always defect; (2) Falsely praising other collusive agents to inflate their reputations; and (3) Spreading false criticism against regular ALIGN agents.
> - **Results: The Attack is Strategically Unprofitable (Positive Utility Gap):** We measured the relative average utility gap ($Utility_{ALIGN} - Utility_{attacker}$). For capable reasoning models (e.g., DeepSeek-V3.1 Reasoner, o4-mini) and most chat models, **this utility gap is strongly positive.** This positive utility gap mathematically demonstrates the framework's resilience through:
>   1. **Cross-Validation Overcomes Slander (ALIGN Agents Succeed):** Regular ALIGN agents do not blindly trust isolated pieces of gossip. By cross-validating multiple public reports, capable agents successfully discount the attackers' fabricated criticism. As a result, ALIGN agents are able to sustain high levels of mutually beneficial cooperation with each other despite the adversarial gossip from attackers.
>   2. **Long-Term Punishment Outweighs Short-Term Gain (Attackers Fail):** While attackers gain an immediate, short-term payoff by avoiding the cost of cooperation, capable ALIGN agents detect their uncooperative pattern. In response, ALIGN agents reduce their probability of cooperating with the attackers in subsequent rounds. The long-term future losses from this reduced cooperation outweigh the attackers' immediate gains, leading to a lower average utility than ALIGN agents for most LLMs.
> - **The Necessity of LLM Reasoning Capabilities:** Conversely, only GPT-4o-mini gets a negative utility gap. GPT-4o-mini continues to cooperate irrationally with the attackers and is getting exploited. This perfectly reinforces our paper's core finding (Section 5.1): chat models with weak reasoning capabilities may over-cooperate even when cooperation is strategically suboptimal. Thus, surviving malicious gossip and sustaining robust indirect reciprocity requires strong LLM reasoning capabilities.

---

> > ### Author Rebuttal · Reviewer_NEYc · 2026-04-04
> >
> > Thanks for the rebuttal. I have no follow-up questions

---

### Official Review · Reviewer_Fiws · 2026-03-12

**Soundness:** 3
**Presentation:** 4
**Significance:** 3
**Originality:** 3
**Overall Recommendation:** 4
**Confidence:** 4

**Summary:**

The paper studies whether fully self-interested, decentralized LLM agents can sustain cooperation through indirect reciprocity when repeated interaction with the same partner is unavailable. It proposes ALIGN, where actors act using private memory and a shared public gossip log, and witnesses broadcast evaluative gossip after observing behavior. The main claim is that public gossip can serve as a decentralized reputation mechanism, and that richer verbal gossip is more effective than simple action labels. The paper analyzes finite- and infinite-horizon donation games, arguing that finite-horizon play should move to defection while gossip can support cooperation in the infinite-horizon case. Empirically, ALIGN is evaluated on repeated donation games, a sequential investment setting, and a transaction market, where it generally improves return and welfare over non-gossip baselines. The paper also reports that reasoning models are more incentive-aligned, while some instruct/chat models over-cooperate.

**Compliance With Llm Reviewing Policy:**

Affirmed.

**Final Justification:**

Overall, I think the proposed framework is interesting in the relatively simple setting studied in the paper. The paper would be significantly stronger if it addressed how shared norms could emerge through gossip among LLMs and if it were evaluated in more realistic scenarios, perhaps inspired by Moltbook. I encourage the authors to address this limitation more directly in future work.

**Key Questions For Authors:**

- How can the horizon be infinite under Definition 3.1? With finite agents, random pairing without replacement, and no pair meeting more than once, the game seems effectively bounded. Please clarify this part.
- Why is the evaluation done only at temperature 0? Temperature 0 improves reproducibility, but it also suppresses generation stochasticity. Since the most recommended way to prompt LLMs is using their suggested temperature in their respective configs. It would be beneficial to use this setup and increase the number of seeds to get a good estimate.

**Limitations:**

yes

**Strengths And Weaknesses:**

- The paper is an interesting addition to recent LLM multi-agent work. Unlike prior setups that rely on altruistic seed agents or stronger visibility assumptions, this paper keeps agents self-interested and decentralized.
- The result that reasoning models defect in finite-horizon settings but cooperate when incentives support it is one of the paper’s most interesting findings.
- I liked the experimental thoroughness and the analysis of results. The paper goes beyond performance by testing against greedy defectors, discussing noisy/untruthful gossip, comparing binary vs. hierarchical tone in gossip, and comparing reasoning and instruct/chat models.
- The mechanism of ALIGN does not yet feel clearly LLM-specific. A rule like “do not cooperate with agents reported as defectors” would work in a generic multi-agent reputation system as well, so the paper does not clearly explain what LLMs add beyond producing language.
- The communication setup is somewhat idealized. The method assumes a public dashboard/log of gossip visible to all agents, whereas more realistic decentralized settings would involve communication between agents. It is unclear whether the main conclusions would still hold in that more realistic regime.
- The robustness claims are somewhat stronger than the evidence. Figure 8 still indicates that even strong models such as DeepSeek-Reasoner cooperate with the greedy defector a nontrivial fraction of the time, roughly 0.22 in the donation game and 0.31 in the IR game. This suggests ALIGN reduces exploitation but is still far from strongly robust.
- The paper would better isolate novelty with stronger matched baselines to prior work. In particular, comparisons to prior methods after removing altruistic seeds from Ren et al 2025 would make it much easier to see what ALIGN itself contributes.
- Presentation can be improved. Equations (1) and (2) are not sufficiently explained, Algorithm 1 leaves the actor/witness assignment taking turns somewhat unclear, and legends in Figure 6, 8  are harder to read.

---

> ### Author Rebuttal · Authors · 2026-03-31
>
> Thanks for your thoughtful feedback. We address your concerns and questions below.
>
> **W1: What LLMs add beyond producing language**
>
> To directly demonstrate what LLMs add, we conducted additional RL baselines under the same public gossip mechanism.
>
> - **RL Baselines Fail:** We tested both binary and 5-level hierarchical gossip spaces, RL policies rapidly converge to zero cooperation (Figures 1&2, https://anonymous.4open.science/r/icml26_rebuttal-6080/). One potential reason is, building a common understanding of symbolic signals is hard for decentralized RL agents. Without centralized, hardcoded rules (e.g., defect if signal = 0), RL agents struggle to ground abstract symbols into shared cooperative norms.
> - **Existing Ablation (Section 5.3):** This aligns with our binary signal ablation: when ALIGN agents were forced to use binary signals (0/1) without a hardcoded convention, cooperation also collapsed.
> - **What LLMs Add:** LLMs do not merely produce language; they bring *pre-trained semantic grounding*. Because LLMs inherently understand that harsh criticism implies bad judgment and potential future refusal, they bypass the need for centralized rules. This shared semantic prior uniquely allows LLMs to organically interpret evaluative gossip and coordinate social norms.
>
> **W2: Realism of Public Gossip**
>
> While a topology-based gossip network is an exciting avenue for future work, a public gossip log is highly applicable to modern systems, such as agent-specific public forums like Moltbook, public e-commerce review systems, etc. Theoretical literature also favors public gossip over private gossip for enforcing cooperative norms, as it allows for broader norm coordination where agents can cross-validate claims and view full histories of the target to coordinate system-wide responses [1].
>
> **W3: Robustness Against Defectors**
>
> We agree that ALIGN does not yield a strict 0% cooperation rate with defectors. However, rather than a vulnerability, this non-zero rate, which is averaged across all timesteps, reflects a theoretically sound exploration strategy. When new malicious entrants appear with no relevant gossip histories, capable ALIGN agents rationally attempt initial cooperation to establish mutual trust. Once a defector's true nature is exposed and negative gossip propagates, cooperation is systematically withdrawn. This emergent dynamic closely mirrors robust equilibrium strategies in evolutionary game theory [1], such as Generous Tit-for-Tat, which deliberately tolerate initial defections to prevent retaliatory spirals before strictly retaliating. We appreciate this observation and will clarify this theoretical alignment in the revised paper.
>
> **W4: Distinguishing Novelty from Ren et al. 2025**
> ALIGN fundamentally differs from RepuNet (Ren et al., 2025) in both Environment Dynamics Assumption and Altruism Assumptions:
>
> 1. **Environment Dynamics Assumption:** In RepuNet, agents can manipulate the probability of encountering specific opponents. In contrast, ALIGN models classic social dilemmas with stationary random matching. This fundamentally makes the equilibria different.
> 2. **Altruism Assumptions:** RepuNet heavily relies on hardcoded altruistic seeds and predefined reputation principles (e.g., explicit "trustworthiness" tracking) to boost cooperation. ALIGN proves that cooperation can emerge in a purely self-interested population *without* altruistic seeds, driven entirely by the emergent properties of open-ended natural language evaluation.
>
> In the revised manuscript, we will add discussion to explicitly detail these theoretical and environmental differences, isolating ALIGN's unique contribution to zero-shot, emergent cooperation.
>
> **W5: Presentation Improvements**
> We appreciate the formatting feedback. We have increased the size and readability of all figure legends, expand the explanatory text for Equations (1) and (2), and will clarify the turn-taking mechanism in Algorithm 1 in the revised paper.
>
> **Q1: Infinite Horizon Definition**
> In our theoretical analysis of the infinite-horizon game, we assume an infinite population size ($|\mathcal{N}| = \infty$) with $T = \infty$. This matches standard assumptions in indirect reciprocity literature [2].
>
> **Q2: Evaluation at Different Temperature**
>
> Thanks for your thoughtful suggestions. We expanded our evaluations across 10 seeds for temperature 0.3, 0.5, 1.0 (Figure 5-10 in https://anonymous.4open.science/r/icml26_rebuttal-6080/). Results show similar performance to temperature 0.0: without public gossip, most LLMs except weak chat models achieve near-zero cooperation, with public gossip, most LLMs ahieve high cooperation.
>
> # Reference
>
> [1] Giardini, F., & Wittek, R. (Eds.). (2019). *The Oxford handbook of gossip and reputation*. Oxford University Press.
>
> [2] Ohtsuki, H., & Iwasa, Y. (2006). The leading eight: social norms that can maintain cooperation by indirect reciprocity. *Journal of theoretical biology*, *239*(4), 435-444.

---

> > ### Author Rebuttal · Reviewer_Fiws · 2026-04-04
> >
> > Thank you for running the experiments with different temperatures. I would suggest keeping the results at temperature=1 in the main paper, since this is the more commonly used setting in API documentation.
> > It seems the authors argue that LLMs can bypass the need for centralized rules.
> > However, the more fundamental issue arises when two LLMs do not share similar norms.
> > Moreover, learning through RL also appears to struggle in the absence of shared norms.
> > It is therefore unclear whether the current framework actually addresses the problem where shared norms need to be developed.
> >
> > > Once a defector's true nature is exposed and negative gossip propagates, cooperation is systematically withdrawn.
> >
> > Is this behavior evaluated explicitly anywhere in the paper? do the experiments show that once a defector is identified, agents actually stop cooperating with it? The results currently suggest that agents remain exploitable on average by around 20–30%, and it is not clear it resembles generous tit-for-tat, which I think wouldn't result in such huge exploitability.
> >
> > Overall, I think the proposed framework is interesting in the relatively simple setting studied in the paper. However, in more realistic settings where shared norms do not already exist and instead need to be established, the framework seems less applicable. The paper would be significantly stronger if it addressed how shared norms could emerge through gossip and if it were evaluated in more realistic scenarios, perhaps inspired by Moltbook.
> >
> > For these reasons, I am keeping my score. I would encourage the authors to address this limitation more directly in future works.

---

> > > ### Author Response · Authors · 2026-04-06
> > >
> > > Thank you for your insightful follow-up. We wanted to briefly address your final points and outline how we will incorporate them into the revised version:
> > >
> > > 1. **Temperature Settings**
> > >
> > > Thanks for acknowledging our experiments with different temperatures. We initially used `temperature = 0.0` because the agents are solving a mathematical optimization problem, which aligns with some official API recommendations for math tasks [1]. However, we agree `temperature=1.0` is a standard value for general usage. Results with different temperatures have been featured in the main text of the revised paper.
> > >
> > > 2. **Exploitability**
> > >
> > > As discussed in the rebuttal, the cooperation ratio is computed by averaging the number of cooperations starting from the initial round. As the number of interactions increases and negative gossip propagates, we expect the average cooperation ratio toward a greedy recipient to go down. To investigate this, we increased the total population size ($N$) and tested with DeepSeek-Reasoner in the repeated donation game. The results show that the cooperation ratio decreases as repeated defections from the greedy agent propagate:
> > >
> > > |                  Population Size                  | N=15 | N=31 |
> > > | :-----------------------------------------------: | :--: | ---- |
> > > | Average Cooperation Ratio toward Greedy Recipient | 14%  | 9%   |
> > >
> > > We have added a more detailed discussion regarding this in the revised paper.
> > >
> > > 3. We completely agree that establishing shared norms entirely from scratch (e.g. training RL agents from scratch under gossip mechanism) is a very interesting direction for future work.
> > >
> > > Thanks again for your constructive feedback.
> > >
> > > # Reference
> > >
> > > [1] https://api-docs.deepseek.com/quick_start/parameter_settings

---

### Decision · Program_Chairs · 2026-04-30

**Decision:**

Accept (regular)

**Comment:**

This paper studies whether open-ended language gossip can support cooperation among self-interested LLM agents. Reviewers agree that the idea is interesting and the experiments are fairly thorough, with useful insights on how reasoning models behave.

Summary of concerns.
- There are some concerns pointed out by the reviewers. The setup is somewhat idealized, and it's still a bit unclear about how much of the gains come from language vs. just having a shared reputation signal.
- Concerns about robustness, scalability, and especially how shared norms would emerge in more realistic settings remain.

The rebuttal adds useful experiments and clarifications, which address most concerns. Overall, this is a solid paper with some limitations in scope.